# Learning Robust Diffusion Models from Imprecise Supervision

## Abstract

Conditional diffusion models have achieved remarkable success in various generative tasks recently, but their training typically relies on large-scale datasets that inevitably contain imprecise information in conditional inputs. Such supervision, often stemming from noisy, ambiguous, or incomplete labels, will cause condition mismatch and degrade generation quality. To address this challenge, we propose *DMIS*, a unified framework for training robust Diffusion Models from Imprecise Supervision, which is the first systematic study within diffusion models. Our framework is derived from likelihood maximization and decomposes the objective into generative and classification components: the generative component models imprecise-label distributions, while the classification component leverages a diffusion classifier to infer class-posterior probabilities, with its efficiency further improved by an optimized timestep sampling strategy. Extensive experiments on diverse forms of imprecise supervision, covering tasks of image generation, weakly supervised learning, and noisy dataset condensation demonstrate that *DMIS* consistently produces high-quality and class-discriminative samples.

## 1 Introduction

Diffusion models (DMs) (Ho et al., 2020; Song et al., 2020; Karras et al., 2022) have emerged as powerful generative frameworks that have unprecedented capabilities in generating realistic data (He et al., 2025; Yang et al., 2024; Ho et al., 2022). With the classifier guidance (Ho & Salimans, 2022; Dhariwal & Nichol, 2021), conditional diffusion models (CDMs) extended the capabilities of DMs by conditioning the generation process on additional information, such as text descriptions or class labels. These models have demonstrated remarkable performance in various tasks, including text-to-image synthesis (Rombach et al., 2022; Saharia et al., 2022), image inpainting (Zhao et al., 2024; Corneanu et al., 2024), and super-resolution (Esser et al., 2024; Xie et al., 2025).

Unfortunately, the conditioning information required by CDMs is often imprecise in real-world scenarios. When sourced from the internet or obtained through crowdsourcing, such information can be affected by factors such as privacy constraints or limited annotator expertise, leading to various imperfections. In particular, the conditioning data may contain noise, exhibit ambiguity, or suffer from missing and incomplete annotations. We refer to such cases collectively as imprecise supervision (Chen et al., 2024a), where the provided conditioning information is not fully aligned with the true underlying labels. This includes scenarios such as noisy-label data (Li et al., 2017; Wei et al., 2021), partial-label data (Wang et al., 2025b;a), and supplementary-unlabeled data (He et al., 2023). These forms of imprecise supervision can introduce incorrect inductive biases during training and severely affect the reliability and generalization of CDMs.

To address this, several recent studies have proposed adaptations of diffusion models to handle imprecise supervision, such as noise-robust diffusion models (Na et al., 2024; Li et al., 2024) and positive-unlabeled diffusion models (Takahashi et al., 2025). However, these approaches often focus on specific types of imprecise supervision. Moreover, many of them rely on strong external priors to guide the learning process. For example, Na et al. (2024) estimated a noise transition matrix using external noisy-label learning methods, and Li et al. (2024) required risk confidence scores associated with noisy samples. These diffusion-based methods not only rely on prior knowledge from data or previous techniques, but are also designed with task-specific architectures for particular types of supervision. Such reliance and structural complexity limit their applicability and efficiency in

practice. There remains a need for a unified framework that can robustly train CDMs under diverse forms of imprecise supervision without requiring strong prior assumptions.

In this paper, to train a robust CDM in a unified manner, we first formulate the overall learning objective as a likelihood maximization problem (Section 4.1). Then we decompose this objective into a generative term that models the imprecise data distribution (Section 4.2) and a classification term that infers posterior label probabilities from imprecise supervision (Section 4.3). During generative modeling, we show that the imprecise-label conditional score can be expressed as a linear combination of clean-label conditional scores, weighted by the corresponding posterior probabilities. Building on this insight, we propose a weighted denoising score matching objective, which enables the model to achieve label-conditioned learning without requiring clean annotations. Finally, to reduce the time complexity of posterior inference, we further introduce an efficient timestep sampling strategy (Section 5). Extensive experiments across multiple tasks, including image generation, weakly supervised learning, and noisy dataset condensation show that CDMs trained with our framework not only achieve strong generative quality but also produce class-discriminative samples. Our contributions are summarized as follows:

- We propose a unified diffusion framework for training CDMs under diverse forms of imprecise supervision, which is the first exploration in the diffusion model field.
- To improve efficiency, we develop an optimized timestep sampling strategy for diffusion classifiers that greatly reduces the computation cost without compromising performance.
- Building on this framework, we pioneer the study of noisy dataset condensation, a practical yet previously unexplored setting, and establish a solid baseline for future research.
- Extensive experiments on image generation, weakly supervised learning, and noisy dataset condensation demonstrate the effectiveness and versatility of our unified framework.

# 2 RELATED WORK

## 2.1 ROBUST DIFFUSION MODELS

Training conditional diffusion models under limited or imperfect supervision is still relatively under-explored. Recent work has begun to address specific forms of weak or noisy information, such as noise-robust diffusion models (Na et al., 2024; Li et al., 2024) that mitigate corrupted labels, and positive-unlabeled diffusion models (Takahashi et al., 2025) that combine positive samples with large unlabeled corpora to approximate conditional distributions. We take a different perspective: rather than tailoring objectives to a single type of imperfect label, we formulate a unified conditional score-learning framework that can be instantiated under multiple imprecise-label regimes.

## 2.2 IMPRECISE LABEL LEARNING

Imprecise label learning studies supervision that is incomplete, ambiguous, or corrupted relative to clean ground-truth labels. Canonical settings include partial-label learning (Feng et al., 2020; Wu et al., 2022; Tian et al., 2023; Lv et al., 2020; Wang et al., 2025b), where each instance is associated with a candidate label set containing the true label; semi-supervised learning (Berthelot et al., 2019b; Zhang et al., 2021a; Yang et al., 2022; Wang et al., 2022c), where only a subset of samples are labeled; and noisy-label learning (Han et al., 2018; Wei et al., 2021; Han et al., 2020), where observed labels are corrupted versions of the true labels. Beyond these settings, mixture imprecise-label learning (Chen et al., 2024a; Zhang et al., 2020; Wei et al., 2023; Shukla et al., 2023; Xie et al., 2024) combines several forms of imprecision in a single framework. Our work can be viewed as lifting these ideas from discriminative prediction to conditional score modeling, providing a generative view of learning with heterogeneous imprecise supervision.

## 2.3 DATASET CONDENSATION

Dataset distillation (DD) (Wang et al., 2018) compresses a large labeled dataset into a compact synthetic set that preserves task-relevant information, thereby reducing training cost while maintaining competitive accuracy. Bi-level optimization methods learn synthetic data whose training signals

match those of the original data via gradient, trajectory, or meta-model matching (Zhao et al., 2021; Kim et al., 2022; Cazenavette et al., 2022a; Cui et al., 2023; Wang et al., 2018; Loo et al., 2022), often achieving high fidelity at nontrivial computational cost. Distribution-matching approaches instead align statistics in pixel, feature, or kernel space (Wang et al., 2022b; Sajedi et al., 2023; Xue et al., 2025; Yin et al., 2024; Sun et al., 2024; Shao et al., 2024; Yin & Shen, 2024), enabling more scalable DD. While DD primarily targets data efficiency under clean labels, our framework instead focuses on robustness to imprecise supervision.

## 3 BACKGROUND

**Diffusion Models.** Let $\mathcal{X} \subseteq \mathbb{R}^d$ denote the $d$-dimensional input space. Given a clean input $\mathbf{x} := \mathbf{x}_0$ from the real data distribution with density $q(\mathbf{x}_0)$, the forward diffusion process corrupts the data into a sequence of noisy samples $\{\mathbf{x}_t\}_{t=1}^{T}$[1] by gradually adding Gaussian noise with a fixed scaling schedule $\{\alpha_t\}_{t=1}^{T}$ and a fixed noise schedule $\{\sigma_t\}_{t=1}^{T}$, as defined by

$$q(\mathbf{x}_t \,|\, \mathbf{x}_0) = \mathcal{N}(\mathbf{x}_t; \alpha_t \mathbf{x}_0, \sigma_t^2 \mathbf{I}), \tag{1}$$

where $\mathbf{I}$ denotes the identity matrix and $\mathcal{N}(\mathbf{x}; \boldsymbol{\mu}, \boldsymbol{\Sigma})$ denotes the Gaussian density with mean $\boldsymbol{\mu}$ and covariance matrix $\boldsymbol{\Sigma}$. Assuming that the signal-to-noise ratio $\text{SNR}(t) = \alpha_t^2/\sigma_t^2$ decreases monotonically over time, the sample $\mathbf{x}_t$ becomes increasingly noisier during the forward process. The scaling and noise schedules are prescribed such that $\mathbf{x}_T$ nearly follows an isotropic Gaussian distribution. The reverse process for Eq. (1) is defined as a Markov chain, which aims to approximate $q(\mathbf{x}_0)$ by gradually denoising from the standard Gaussian distribution $p(\mathbf{x}_T) = \mathcal{N}(\mathbf{x}_T; \mathbf{0}, \mathbf{I})$:

$$p_\theta(\mathbf{x}_{0:T}) = p(\mathbf{x}_T) \prod_{t=1}^{T} p_\theta(\mathbf{x}_{t-1} \,|\, \mathbf{x}_t), \tag{2}$$

$$p_\theta(\mathbf{x}_{t-1} \,|\, \mathbf{x}_t) = \mathcal{N}\big(\mathbf{x}_{t-1}; \, \boldsymbol{\mu}_\theta(\mathbf{x}_t, t), \, \tilde{\sigma}_t^2 \mathbf{I}\big), \tag{3}$$

where $\boldsymbol{\mu}_\theta$ is generally parameterized by a time-conditioned score prediction network $\mathbf{s}_\theta(\mathbf{x}_t, t)$ (Song et al., 2020; 2021; Song & Ermon, 2019; 2020):

$$\boldsymbol{\mu}_\theta(\mathbf{x}_t, t) = \frac{\alpha_{t-1}}{\alpha_t}\Big[\mathbf{x}_t + \Big(\sigma_t^2 - \frac{\alpha_t^2}{\alpha_{t-1}^2}\sigma_{t-1}^2\Big)\mathbf{s}_\theta(\mathbf{x}_t, t)\Big]. \tag{4}$$

The reverse process can be learned by optimizing the variational lower bound on log-likelihood as

$$\log p_\theta(\mathbf{x}) \geq -\mathbb{E}_t\Big[w_t\big\|\mathbf{s}_\theta(\mathbf{x}_t, t) - \nabla_{\mathbf{x}_t} \log q_t(\mathbf{x}_t)\big\|_2^2\Big] + C_1, \tag{5}$$

where $\mathbb{E}$ denotes the expectation, $w_t = \frac{\sigma_t^2}{2}\big(\frac{\sigma_t^2 \alpha_{t-1}^2}{\sigma_{t-1}^2 \alpha_t^2} - 1\big)$, and $C_1$ is a constant that is typically small and can be dropped (Song et al., 2020). The expectation term is called the *score matching loss* (Kingma et al., 2021), where $\nabla \log q_t(\mathbf{x}_t)$ is the gradient of data density at $\mathbf{x}_t$ in data space.

The above definition can be reformulated to match other commonly used diffusion models, such as those in Ho et al. (2020), Karras et al. (2022) and Song et al. (2020). The corresponding conversions are detailed in Appendix B.1. For clarity, we adopt the the elucidated diffusion model (EDM) (Karras et al., 2022) as the default diffusion model throughout this paper, as it offers a unified structure and well-optimized parameterization.

**Imprecise Supervision.** Imprecise-label data typically refers to settings where the true label is not directly available, and instead only imprecise label information is provided. Let $\mathcal{Y} = [c] := \{1, \ldots, c\}$ represent the label space with $c$ distinct classes. In this work, we primarily focus on three representative forms of imprecise supervision that have been widely studied in the literature:

- *Partial-label data*, where each instance $X$ is associated with a candidate label set $S \subset [c]$ that is guaranteed to contain the true label $Y$, i.e., $p(Y \in S \,|\, X, S) = 1$. This setting is widely studied in partial-label learning (Tian et al., 2023).

- *Supplementary-unlabeled data*, consisting of a small labeled subset $(X^1, Y^1)$ together with a large number of unlabeled samples $(X^u, \emptyset)$. This scenario is the focus of semi-supervised learning (Yang et al., 2022), which aims to exploit unlabeled data to improve generalization.

---

[1]We use the subscript $t$ of the sample $\mathbf{x}$ to denote the noisy version of the sample at timestep $t$.

- *Noisy-label data*, where the observed label $\hat{Y}$ is a corrupted version of the underlying true label $Y$, modeled by a conditional distribution $p(\hat{Y} \mid X, Y)$. This gives rise to noisy-label learning (Han et al., 2020), which seeks to build models robust to label corruption.

# 4 METHODOLOGY

In this section, we first introduce the unified learning objective that integrates generative and classification components. Then we elaborate on the formulation and optimization of these components.

## 4.1 UNIFIED LEARNING OBJECTIVE

To robustly learn a diffusion model with learnable parameters $\theta$ under imprecise supervision (denoted as $Z \subseteq \mathcal{Y}$), we treat the true label $Y$ as a latent variable and maximize the likelihood of the joint distribution of the input $X$ and $Z$. By the maximum likelihood principle, our objective is to find

$$\theta^* = \arg\max_\theta \log p_\theta(X, Z) = \arg\max_\theta \log \sum_Y p_\theta(X, Y, Z), \qquad (6)$$

where $\theta^*$ denotes the optimal parameter. Eq. (6) involves the log of the marginalization over latent variables and cannot generally be solved in closed form. To circumvent this intractability, we instead maximize a variational lower bound on the marginal log-likelihood:

$$\begin{aligned}
\theta^n &= \arg\max_\theta \; \mathbb{E}_{p_\phi(Y|X,Z)}\big[\log p_\theta(X, Y, Z)\big] \\
&= \arg\max_\theta \; \Big\{ \log p_\theta(X \mid Z) + \mathbb{E}_{p_\phi(Y|X,Z)}\big[\log p_\theta(Y \mid X, Z)\big] \Big\},
\end{aligned} \qquad (7)$$

where $\theta^n$ denotes the $n$-th estimate of $\theta$, and $\phi$ is instantiated as the exponential moving average (EMA) of $\theta$ over its 1st through $(n-1)$ iterates. A complete derivation of this variational lower bound is provided in Appendix B.3. From Eq. (7), we can observe that maximizing the marginal likelihood can be performed from generative and classification perspectives. The former focuses on modeling the data distribution conditioned on the imprecise supervision, while the latter aims to infer the posterior distribution based on the feature and the imprecise label. In this paper, we adopt the commonly used class-conditional setting, where the generation of the imprecise label $Z$ is assumed to be independent of the input $X$ given the true label $Y$ (Yao et al., 2020; Wen et al., 2021).

## 4.2 GENERATIVE OBJECTIVE: MODELING THE IMPRECISE DATA DISTRIBUTION

Since samples are assumed to be independent of each other, we present the analysis in this and the following subsections using a single sample $(\mathbf{x}, z)$ for notational clarity, with the final objective computed over the entire dataset. Following the standard formulation of diffusion models in Eq. (5), we parameterize the conditional generative process $p_\theta(\mathbf{x} \mid z)$ using a score network $\mathbf{s}_\theta(\mathbf{x}_t, z, t)$. The corresponding variational lower bound on the conditional log-likelihood is given by

$$\log p_\theta(\mathbf{x}_0 \mid z) \geq -\mathbb{E}_t\Big[w_t\big\|\mathbf{s}_\theta(\mathbf{x}_t, z, t) - \nabla_{\mathbf{x}_t} \log q_{t|0}(\mathbf{x}_t \mid \mathbf{x}_0, z)\big\|_2^2\Big] + C_2, \qquad (8)$$

where $C_2$ is another constant. Directly optimizing the score network with this objective on imprecise-label data would lead it to converge to the score of the imprecise conditional distribution.

**Remark 1.** Let $\hat{\theta}$ denote the parameters obtained by maximizing the lower bound in Eq. (8) using denoising score matching. In this case, the learned score function satisfies $\mathbf{s}_{\hat{\theta}}(\mathbf{x}_t, z, t) = \nabla_{\mathbf{x}_t} \log q_t(\mathbf{x}_t \mid z)$ for all $\mathbf{x}_t \in \mathcal{X}$, $z \subseteq \mathcal{Y}$, and $t \in [T]$. However, since $q_t(\mathbf{x}_t \mid z)$ corresponds to the imprecise-label density, the resulting generation is biased and thus fails to fully recover the true data distribution. The derivation and visualization of this bias is deferred to Appendix B.5.

Therefore, to align the learned score with the clean-label conditional score, we propose modifying the objective to correct the gradient signal from score matching (Kingma et al., 2021). Building on the linear relationship between clean- and noisy-label conditional scores modeled by Na et al. (2024), we further derive an explicit relationship that connects imprecise-label conditional scores to their clean-label counterparts.

**Theorem 1.** *Under the class-conditional setting, for all $\mathbf{x}_t \in \mathcal{X}$, $z \subseteq \mathcal{Y}$, and $t \in [T]$,*

$$\nabla_{\mathbf{x}_t} \log q_t(\mathbf{x}_t \mid z) = \sum\nolimits_{y=1}^c p(y \mid \mathbf{x}_t, z) \, \nabla_{\mathbf{x}_t} \log q_t(\mathbf{x}_t \mid y). \tag{9}$$

The formal proof is in Appendix B.6. Since $p(y \mid \mathbf{x}_t, z) \geq 0$ and $\sum_{y=1}^c p(y \mid \mathbf{x}_t, z) = 1$, Theorem 1 implies that the imprecise-label conditional score can be expressed as a convex combination of the clean-label conditional scores, weighted by $p(y \mid \mathbf{x}_t, z)$. These weights represent the model's posterior probability over labels given $\mathbf{x}_t$ and $z$, implicitly requiring the model to perform classification during training. To our knowledge, this is the first work to explicitly reveal and exploit the classification capability of diffusion models within the training process under imprecise supervision.

According to Remark 1, directly optimizing the denoising score matching objective in Eq. (8) drives the score network to approximate the imprecise-label conditional score. However, Theorem 1 shows that this score can be decomposed as a convex combination of clean-label conditional scores, weighted by the posterior probability $p(y \mid \mathbf{x}_t, z)$. Motivated by this insight, we propose a new training objective that supervises the clean-label score network $\mathbf{s}_\theta(\mathbf{x}_t, y, t)$ through a reweighted aggregation of its posterior outputs. The resulting weighted denoising score matching loss is

$$\mathcal{L}_{\text{Gen}}(\theta) = \mathbb{E}_t \left[ w_t \left\| \sum\nolimits_{y=1}^c p(y \mid \mathbf{x}_t, z) \, \mathbf{s}_\theta(\mathbf{x}_t, y, t) - \nabla_{\mathbf{x}_t} \log q_{t|0}(\mathbf{x}_t \mid \mathbf{x}_0, z) \right\|_2^2 \right]. \tag{10}$$

This loss encourages the weighted aggregation of clean-label scores to approximate the imprecise score derived from data, thereby enabling label-conditioned learning without the need for explicit clean annotations. The following Proposition 1, with proof provided in Appendix B.7, guarantees that the optimal solution recovers the clean-label conditional scores:

**Proposition 1.** *Let $\theta_{\text{Gen}}^* = \arg\min_\theta \mathcal{L}_{\text{Gen}}(\theta)$ be the minimizer of Eq. (10). Then, for all $\mathbf{x}_t \in \mathcal{X}$, $z \subseteq \mathcal{Y}$, and $t \in [T]$, the learned score function satisfies $\mathbf{s}_{\theta_{\text{Gen}}^*}(\mathbf{x}_t, y, t) = \nabla_{\mathbf{x}_t} \log q_t(\mathbf{x}_t \mid y)$.*

### 4.3 CLASSIFICATION OBJECTIVE: INFERRING LABELS FROM IMPRECISE SIGNALS

We assume the class prior to be uniform, i.e., $p(y) = 1/c$. To infer the class-posterior probability $p_\theta(y \mid \mathbf{x}_t)$, we adopt a diffusion-based approximation as defined below:

**Definition 1** (Approximated Posterior Noised Diffusion Classifier (Chen et al., 2024b))**.** Assuming the uniform prior $p(y)$, the class-posterior probability for a noisy input $\mathbf{x}_t$ under a conditional diffusion model can be derived using Bayes' rule, as follows:

$$p_\theta(y \mid \mathbf{x}_t) = \frac{p_\theta(\mathbf{x}_t \mid y)}{\sum_{y'} p_\theta(\mathbf{x}_t \mid y')} = \frac{\exp\{\log p_\theta(\mathbf{x}_t \mid y)\}}{\sum_{y'} \exp\{\log p_\theta(\mathbf{x}_t \mid y')\}}. \tag{11}$$

Here, following Chen et al. (2024b), the conditional likelihood $\log p_\theta(\mathbf{x}_t \mid y)$ is approximated by the conditional evidence lower bound (ELBO), given by

$$\log p_\theta(\mathbf{x}_t \mid y) \approx - \sum\nolimits_{\tau=t+1}^{T-1} w_\tau \mathbb{E}_{q(\mathbf{x}_\tau \mid \mathbf{h}_\theta(\mathbf{x}_t, y, t))} \left[ \left\| \mathbf{h}_\theta(\mathbf{x}_\tau, y, \tau) - \mathbf{x}_0 \right\|_2^2 \right], \tag{12}$$

where $\mathbf{h}_\theta(\mathbf{x}_\tau, y, \tau) = \frac{\mathbf{x}_\tau}{\alpha_\tau} + \frac{\sigma_\tau^2}{\alpha_\tau} \mathbf{s}_\theta(\mathbf{x}_\tau, y, \tau)$ and $w_\tau = \frac{\sigma_\tau^2 + \sigma_{\text{data}}^2}{\sigma_\tau^2 \sigma_{\text{data}}^2} \cdot \frac{P_{\text{std}}^{-1}}{\sigma_\tau \sqrt{2\pi}} \exp\left\{ -\frac{(\log \sigma_\tau - P_{\text{mean}})^2}{2P_{\text{std}}^2} \right\}$.[2]

This diffusion classifier can be extended to non-uniform priors by incorporating $p(y)$ into the logits of class $y$, where $p(y)$ is estimated from the training set (Luo et al., 2024; Wang et al., 2022a), as detailed in Appendix C.2. As training proceeds, the conditional ELBO converges towards the true distribution $q_t(\mathbf{x}_t \mid y)$, thereby yielding increasingly accurate posterior estimates. For convenience, we denote the class probability of a noisy input $\mathbf{x}_t$ with the diffusion classifier as $f(\mathbf{x}_t)$.

To derive the classification loss, we transform the maximization problem of the classification term in Eq. (7) into the minimization of the negative log-likelihood. We show that the resulting objective, i.e., $- \sum_Y p_\phi(Y \mid X, Z) \log p_\theta(Y \mid X, Z)$, naturally aligns closely with prior work (Lv et al., 2020; Tarvainen & Valpola, 2017; Liu et al., 2020) and has been shown to be effective in practice.

---

[2]As specified in the EDM (Karras et al., 2022), we use $\sigma_{\text{data}} = 0.5$, $P_{\text{mean}} = -1.2$ and $P_{\text{std}} = 1.2$.

**Partial-label data.** For partial-label data, the imprecise label $Z$ is given as a candidate set $S$ that is guaranteed to include the true label. In this case, the posterior distribution $p_\theta(Y|X, S)$ is restricted to assign non-zero probability only to labels within the candidate set. Accordingly, for each sample $(\mathbf{x}, s)$, we compute the classification loss from Eq. (7) as

$$\mathcal{L}_{\text{Cls}}^{\text{PL}}(\mathbf{x}) = -\sum_{y \in \mathcal{Y}} p_\phi(y|\mathbf{x}, s) \log p_\theta(y|\mathbf{x}, s) = -\sum_{y \in s} \tilde{f}_\phi^{\text{PL}}(\mathbf{x})_y \log f_\theta(\mathbf{x})_y, \tag{13}$$

where $\tilde{f}_\phi^{\text{PL}}(\mathbf{x})$ denotes the normalized probability over $s$ such that $\sum_{y \in s} \tilde{f}_\phi^{\text{PL}}(\mathbf{x})_y = 1$ and $\tilde{f}_\phi^{\text{PL}}(\mathbf{x})_y = 0$ for all $y \notin s$. Eq. (13) can be interpreted as an EMA-stabilized variant of the method called progressive identification (*PRODEN*) (Lv et al., 2020), where EMA predictions serve as soft pseudo-targets.

**Supplementary-unlabeled data.** In this scenario, the training set consists of a small portion of labeled data and a larger number of unlabeled data. This setting can be regarded as a special case of the partial-label formulation: labeled instances are assigned singleton candidate sets containing the ground-truth label, while unlabeled instances are associated with the full label space. Accordingly, the classification loss for each instance is defined as

$$\mathcal{L}_{\text{Cls}}^{\text{SU}}(\mathbf{x}) = -\sum_{y \in \mathcal{Y}} p_\phi(y|\mathbf{x}, z) \log p_\theta(y|\mathbf{x}, z) = -\sum_{y \in \mathcal{Y}} \tilde{f}_\phi^{\text{SU}}(\mathbf{x})_y \log f_\theta(\mathbf{x})_y, \tag{14}$$

where $\tilde{f}_\phi^{\text{SU}}(\mathbf{x})$ denotes the pseudo-target distribution: for labeled samples, it reduces to a one-hot vector of the ground-truth label, while for unlabeled samples, it corresponds to the EMA model's prediction over the entire label set. This loss can thus be viewed as an EMA-stabilized self-training objective (Tarvainen & Valpola, 2017), a widely used strategy in semi-supervised learning that leverages unlabeled data through soft pseudo-labels.

**Noisy-label data.** In practice, accurately distinguishing clean labels from noisy ones is often difficult, making it challenging to retain reliable supervision while applying self-training for label refinement. To mitigate this, we leverage the memorization effect in noisy-label learning, where neural networks typically fit clean labels before overfitting to noise (Han et al., 2020). Drawing inspiration from the noisy-label learning method called early learning regularization (*ELR*) (Liu et al., 2020), we propose a simpler yet effective loss function that retains its core idea, defined as

$$\mathcal{L}_{\text{Cls}}^{\text{NL}}(\mathbf{x}) = -\sum_{y \in \mathcal{Y}} \text{sg}(\mathbf{r}(\mathbf{x}))_y \log f_\theta(\mathbf{x})_y, \quad \mathbf{r}(\mathbf{x}) = \hat{\mathbf{y}} - \frac{f_\theta(\mathbf{x}) \odot \left(\langle f_\theta(\mathbf{x}), f_\phi(\mathbf{x})\rangle \mathbf{1} - f_\phi(\mathbf{x})\right)}{1 - \langle f_\theta(\mathbf{x}), f_\phi(\mathbf{x})\rangle}, \tag{15}$$

where $\hat{\mathbf{y}}$ denotes the one-hot vector of the noisy label $\hat{y}$, $\text{sg}(\cdot)$ is the stop-gradient operator [3], $\odot$ is the Hadamard product, and $\langle \cdot, \cdot \rangle$ denotes the inner product. This formulation inherits the core principle of *ELR*, stabilizing training through soft pseudo-targets derived from the EMA model. It effectively amplifies the gradient contribution of cleanly labeled samples while suppressing the influence of mislabeled ones, which we further analyze in detail in Appendix C.1.

## 5 TIME COMPLEXITY REDUCTION

The oracle diffusion classifier requires repeated calculations of the conditional ELBO across all classes to make a prediction, resulting in a substantial computation cost. To address this issue, Chen et al. (2024b) showed that when estimating ELBO with Monte Carlo sampling, reusing the same $\mathbf{x}_\tau$ across classes and selecting timesteps at uniform intervals is sufficient for effective classification. However, our experiments reveal that this strategy is empirically suboptimal as illustrated in Figure 1(a). We identify the core reason to be the model's varying discriminative ability across different timesteps, with notable disparities in performance, as shown in Figure 1(b) where the accuracy is evaluated using only a single timestep. Specifically, when the timestep $\tau$ is small, the added noise is negligible, leading to reconstructions with low label sensitivity. Conversely, when the timestep $\tau$ is large, the input becomes overwhelmed by noise, rendering the predictions highly unreliable.

To this end, we aim to identify a compact subset of timesteps that enables efficient ELBO estimation while maintaining sufficient classification performance. Let $p(\tau)$ be a probability density function

---

[3]The stop-gradient operator $\text{sg}(\cdot)$ returns its input but blocks gradient flow, i.e., $\nabla_\mathbf{x} \text{sg}(\mathbf{r}(\mathbf{x})) = 0$.

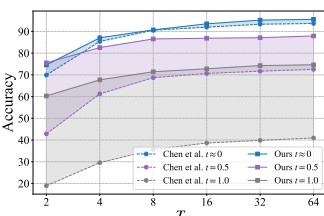 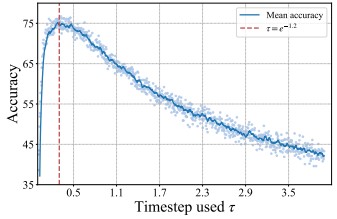 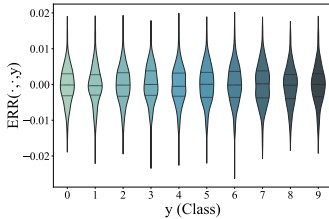

(a) Results across different $T$ and $t$. (b) Results with a single timestep. (c) Class-wise ERR distribution.

Figure 1: (a): Test accuracy (%) comparison on CIFAR-10 dataset under time complexity reduction technique from Chen et al. (2024c) and ours. (b): Test accuracy (%) on CIFAR-10 dataset evaluated with only a single timestep per class. (c): Violin plot of class-wise $\text{ERR}(\cdot, \cdot, y)$ computed across samples using a fixed subinterval length $\Delta$. Wider regions of the violin indicate higher density.

over the interval $\tau \in (0, +\infty)$, satisfying $\int_0^{+\infty} p(\tau)\,\mathrm{d}\tau = 1$. Our objective is to select a subinterval $\tau \in [l, r]$ such that

$$\underset{0 \leq l \leq r}{\text{minimize}} \ \left\| \mathbb{E}_{\tau \sim p(\tau | \tau \in [l, r])}[\hbar(\tau, y)] - \mathbb{E}_{\tau \sim p(\tau)}[\hbar(\tau, y)] \right\|_2^2, \tag{16}$$

where $\hbar(\tau, y) = w_\tau \mathbb{E}_{\mathbf{x}_\tau}[\|\mathbf{h}_\theta(\mathbf{x}_\tau, y, \tau) - \mathbf{x}_0\|_2^2]$. Eq. (16) formalizes the goal of finding a representative range where the expected reconstruction error closely matches that of the full distribution. To strike a compromise between signal and noise within the selected subinterval, we propose choosing it around the median of $p(\tau)$, so that signal-dominant early timesteps and noise-dominant later timesteps complement each other. This strategy yields a more stable and representative approximation, especially when $p(\tau)$ is skewed. Therefore, we provide a formal characterization of the subinterval construction for the EDM by the following theorem.

**Theorem 2.** *Consider an EDM where $\tau$ is sampled from a log-normal distribution, i.e., $\ln(\tau) \sim \mathcal{N}(\tau; P_{\text{mean}}, P_{\text{std}}^2)$, where $P_{\text{mean}} \in \mathbb{R}$ and $P_{\text{std}} > 0$. Given a fixed subinterval length $\Delta$, a sampling range centered around the median of $p(\tau)$ can be constructed by solving the following equation for the left boundary $l$:*

$$l = \text{Solve}_\tau \left( F(\tau) + F(\tau + \Delta) - 1 = 0 \right), \qquad r = l + \Delta,$$

*where $\text{Solve}_\tau(\cdot)$ denotes a numerical root-finding algorithm over $\tau$, such as the Brent method (Brent, 2013), and $F(\cdot)$ is the cumulative distribution function of $p(\tau)$.*

The proof of Theorem 2 as well as a similar conclusion for denoising diffusion probabilistic model (DDPM) (Ho et al., 2020) are provided in Appendix B.8. Notably, our finding aligns with the effective timestep hypothesis proposed in Li et al. (2023) for the DDPM setting. Furthermore, based on Eq. (16), we can derive a necessary condition that any theoretically optimal subinterval must satisfy, as formalized in the following theorem:

**Theorem 3** (Necessary Condition for Optimal Subinterval). *Given $(l^*, r^*)$ be an optimal subinterval of the support of $p(\tau)$, a necessary condition for attaining the theoretical minimum of the squared error objective in Eq. (16) is*

$$\text{ERR}(l^*, r^*, y) = \mathbb{E}_{\tau \sim p(\tau) | \tau \in [l^*, r^*]}[\hbar(\tau, y)] - \frac{\hbar(l^*, y) + \hbar(r^*, y)}{2} = 0. \tag{17}$$

The proof of Theorem 3 can be found in Appendix B.9. Based on Theorem 3, we empirically present the class-wise distribution of $\text{ERR}(\cdot, \cdot, y)$ across samples in Figure 1(c), where the errors are generally concentrated around zero, supporting the effectiveness of our proposed time complexity reduction strategy. Notably, when the subinterval is reduced to a single sampling point, choosing the median of $p(\tau)$ (i.e., $e^{P_{\text{mean}}}$) yields the best classification performance as shown in Figure 1(b). This observation is consistent with our earlier hypothesis regarding the informativeness of the median timestep. In practical posterior inference, we combine timestep subinterval reduction strategy with $\mathbf{x}_\tau$ reuse technique (Chen et al., 2024c) to further improve inference efficiency.

Table 1: Generative results on CIFAR-10 and ImageNette under various settings. 'uncond' and 'cond' indicate unconditional and conditional metrics. **Bold** numbers indicate better performance.

| | Metric | | Noisy-label supervision | | | | Partial-label supervision | | | | Suppl-unlabeled supervision | | | | Clean |
| | | | Sym-40% | | Asym-40% | | Random | | Class-50% | | Random-1% | | Random-10% | | |
| | | | Vanilla | DMIS | Vanilla | DMIS | Vanilla | DMIS | Vanilla | DMIS | Vanilla | DMIS | Vanilla | DMIS | Vanilla |
|---|---|---|---|---|---|---|---|---|---|---|---|---|---|---|---|
| CIFAR-10 uncond | FID | (↓) | **3.33** | 3.47 | 3.23 | **3.10** | 7.76 | **2.26** | 11.75 | **2.77** | 3.16 | **3.12** | 2.93 | **2.89** | 2.05 |
| | IS | (↑) | 9.56 | **9.68** | 9.02 | **9.73** | 9.09 | **9.80** | 9.62 | **9.68** | 10.03 | **10.57** | 9.80 | **9.83** | 10.61 |
| | Density | (↑) | 101.39 | **109.75** | 100.06 | **109.69** | 103.21 | **106.49** | 108.76 | **109.06** | 97.19 | **108.18** | 99.96 | **108.87** | 112.59 |
| | Coverage | (↑) | 81.12 | **81.21** | 80.71 | **81.30** | 68.45 | **82.69** | 64.90 | **81.52** | 78.44 | **81.00** | 81.85 | **82.00** | 83.27 |
| CIFAR-10 cond | CW-FID | (↓) | 29.84 | **13.85** | 14.70 | **13.24** | 27.18 | **10.65** | 32.44 | **11.56** | 16.25 | **16.12** | 11.84 | **11.77** | 9.83 |
| | CW-Density | (↑) | 72.98 | **107.23** | 90.85 | **107.07** | 102.04 | **105.75** | 102.43 | **108.66** | 89.99 | **100.73** | 96.29 | **107.94** | 111.70 |
| | CW-Coverage | (↑) | 73.39 | **80.11** | 79.63 | **79.65** | 65.45 | **82.09** | 61.45 | **81.24** | 75.03 | **76.84** | 80.80 | **81.12** | 83.91 |
| ImageNette uncond | FID | (↓) | 14.11 | **13.44** | 13.93 | **13.91** | 79.13 | **72.62** | 91.28 | **79.12** | 23.88 | **19.26** | 14.32 | **12.84** | 11.52 |
| | IS | (↑) | 12.69 | **13.21** | 12.51 | **13.73** | 9.19 | **9.40** | 9.27 | 9.11 | 12.23 | **13.72** | 12.80 | **13.16** | 13.81 |
| | Density | (↑) | 109.31 | **112.52** | 111.66 | 106.78 | 95.33 | **99.83** | 94.29 | **102.58** | 115.94 | **125.68** | 105.27 | **109.23** | 117.23 |
| | Coverage | (↑) | 76.62 | **76.81** | 78.32 | **79.81** | 21.44 | **32.48** | 16.69 | **22.30** | 53.53 | **55.39** | 73.79 | **75.55** | 80.12 |
| ImageNette cond | CW-FID | (↓) | 80.12 | **60.12** | 62.26 | **58.20** | 157.76 | **63.58** | 163.45 | **67.92** | 71.66 | **70.27** | 49.22 | **44.31** | 40.20 |
| | CW-Density | (↑) | 73.99 | **81.12** | 93.53 | **94.58** | 93.38 | **95.83** | 91.50 | **95.21** | 115.90 | **118.69** | 103.41 | **115.67** | 120.35 |
| | CW-Coverage | (↑) | 67.89 | **71.94** | 74.18 | **75.82** | 19.76 | **24.35** | 15.88 | **18.93** | 51.73 | **52.15** | 72.61 | **74.85** | 78.48 |

## 6 EXPERIMENTS

We present experiments on three tasks including image generation, weakly supervised learning, and dataset condensation to demonstrate the utility and versatility of our method. Evaluations are performed on three benchmark datasets widely used for both generation and classification, covering image resolutions from 28×28 (Fashion-MNIST (Xiao et al., 2017)) and 32×32 (CIFAR-10 (Krizhevsky et al., 2009)) to 64×64 (ImageNette (Deng et al., 2009)). As a baseline, we refer to the model trained with the generative objective in Eq. (8) as the *Vanilla* method. The training hyperparameters are kept consistent with those used in the EDM model (Karras et al., 2022).

**Dataset construction**. For partial-label data, we generate synthetic candidate label sets using both class-dependent (Wen et al., 2021) and random generation models (Feng et al., 2020). In the class-dependent setting, we construct a transition matrix that maps each true label to a set of semantically similar labels, where each similar label is included in the candidate set with probability 50%. In contrast, the random setting assign each incorrect label an equal probability 50% of being included in the candidate set. For supplementary-unlabeled data, we follow a standard semi-supervised setup by randomly selecting 10% and 1% of the training data classwise as labeled samples, and treating the remaining data as unlabeled. For noisy-label data, we consider both symmetric and asymmetric noise. In the symmetric case, labels are uniformly flipped to any incorrect class, whereas in the asymmetric case, they are flipped to semantically similar classes according to a predefined mapping. In both cases, the corruption probability is referred to as the noise rate, which is set to 40%.

### 6.1 TASK1: IMAGE GENERATION

**Setup**. We evaluate the trained CDMs using four unconditional metrics, including Fréchet Inception Distance (FID) (Heusel et al., 2017), Inception Score (IS) (Salimans et al., 2016), Density, and Coverage (Naeem et al., 2020), as well as three conditional metrics, namely CW-FID, CW-Density, and CW-Coverage (Chao et al., 2022). The Class-Wise (CW) metrics are computed per class and then averaged. Detailed descriptions of these metrics are provided in the Appendix E.1.

**Results**. Table 10 reports the generative performance of the *Vanilla* model and our proposed *DMIS* model on various settings. It can be seen that our model outperforms the baseline across almost all cases with respect to both unconditional and conditional metrics. The performance gap is especially

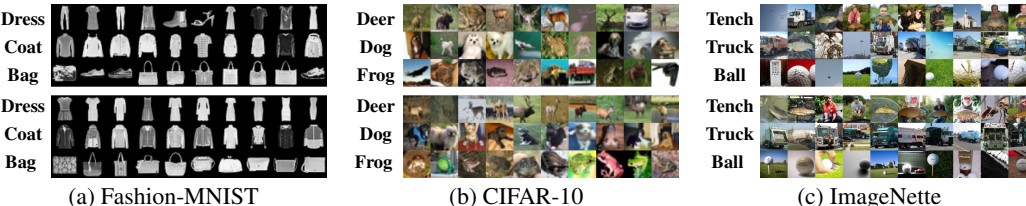

|  (a) Fashion-MNIST | (b) CIFAR-10 | (c) ImageNette |

Figure 2: Comparison of conditionally generated images from *Vanilla* (top) and our *DMIS* model (bottom), each trained with 40% symmetric noise on Fashion-MNIST, CIFAR-10, and ImageNette.

Table 2: Classification results (test accuracy, %) on Fashion-MNIST, CIFAR-10, and ImageNette datasets under various types of imprecise supervision (♠: partial-label, ♡: supplementary-unlabeled, ♣: noisy-label). **Bold** numbers indicate the best performance.[5]

| Dataset♠ | Type | PRODEN | IDGP | PiCO | CRDPLL | DIRK | Vanilla | DMIS^CE | DMIS |
|---|---|---|---|---|---|---|---|---|---|
| F-MNIST | Random | 93.31±0.07 | 92.26±0.25 | 93.32±0.12 | 94.03±0.14 | 94.11±0.22 | 80.20±1.29 | 84.24±0.37 | **94.27±0.55** |
| | Class-50% | 93.44±0.21 | 93.07±0.16 | 93.32±0.33 | 93.80±0.23 | 93.99±0.24 | 66.03±1.43 | 78.45±0.46 | **94.20±0.15** |
| CIFAR-10 | Random | 90.02±0.22 | 89.65±0.53 | 86.40±0.89 | 92.74±0.26 | 93.48±0.14 | 60.25±0.17 | 91.47±0.15 | **94.70±0.49** |
| | Class-50% | 90.44±0.44 | 90.83±0.34 | 87.51±0.66 | 92.89±0.27 | 93.22±0.37 | 56.34±0.50 | 90.52±0.35 | **93.53±0.12** |
| ImageNette | Random | 84.75±0.13 | 84.07±0.26 | 82.15±0.23 | 84.31±0.25 | 87.90±0.11 | 56.04±0.61 | 84.49±0.05 | **89.31±0.21** |
| | Class-50% | 83.50±0.60 | 82.18±0.13 | 84.41±0.93 | 88.08±0.34 | 87.47±0.17 | 59.47±0.51 | 82.34±0.27 | **88.42±0.43** |

| Dataset♡ | Type | Dash | CoMatch | FlexMatch | SimMatch | SoftMatch | Vanilla | DMIS^CE | DMIS |
|---|---|---|---|---|---|---|---|---|---|
| F-MNIST | Random-1% | 84.73±0.09 | 85.31±0.29 | 84.43±0.30 | 84.69±0.17 | 84.72±0.23 | 78.37±0.72 | 82.92±0.17 | **85.92±0.13** |
| | Random-10% | 91.16±0.20 | 90.52±0.12 | 90.69±0.03 | 91.18±0.13 | 91.22±0.11 | 90.50±1.00 | 91.07±0.18 | **92.97±0.21** |
| CIFAR-10 | Random-1% | 70.14±0.69 | 61.45±1.46 | 70.72±0.93 | 73.33±1.02 | 73.74±0.82 | 53.49±0.15 | 75.30±0.17 | **76.40±0.54** |
| | Random-10% | 81.50±0.68 | 77.79±0.53 | 81.35±0.48 | 82.90±0.43 | 88.66±0.60 | 85.13±0.12 | 89.85±0.08 | **92.47±0.39** |
| ImageNette | Random-1% | 57.68±2.19 | 63.88±0.78 | 61.39±0.70 | 58.12±2.66 | 58.50±2.31 | 49.55±0.99 | 62.64±0.24 | **68.23±0.19** |
| | Random-10% | 74.66±0.81 | 73.20±0.46 | 73.08±0.13 | 76.12±0.45 | 75.75±0.25 | 74.70±0.53 | 71.39±0.45 | **77.30±0.15** |

| Dataset♣ | Type | CE | Mixup | Coteaching | ELR | PENCIL | Vanilla | DMIS^CE | DMIS |
|---|---|---|---|---|---|---|---|---|---|
| F-MNIST | Sym-40% | 76.18±0.26 | 92.21±0.03 | 92.17±0.34 | 93.13±0.13 | 90.85±0.58 | 90.11±1.24 | 87.76±0.57 | **93.40±0.40** |
| | Asym-40% | 82.01±0.06 | 92.01±1.02 | 92.78±0.25 | 92.82±0.09 | 91.77±0.69 | 85.41±0.96 | 83.39±0.24 | **93.20±0.30** |
| CIFAR-10 | Sym-40% | 67.22±0.26 | 84.26±0.64 | 86.54±0.57 | 85.68±0.13 | 85.91±0.26 | 80.22±0.10 | 84.75±0.36 | **88.63±0.12** |
| | Asym-40% | 76.98±0.42 | 83.21±0.85 | 79.38±0.39 | 81.32±0.31 | 84.89±0.49 | 86.31±0.10 | 84.21±0.18 | **88.83±0.33** |
| ImageNette | Sym-40% | 58.43±0.77 | 76.65±1.62 | 66.55±1.00 | 84.33±2.86 | 81.94±1.26 | 55.86±1.95 | 80.47±0.56 | **84.12±0.18** |
| | Asym-40% | 71.81±0.38 | 77.16±0.71 | 75.12±0.50 | 73.51±0.31 | 77.20±1.15 | 53.91±1.07 | 77.21±0.19 | **79.30±0.27** |

pronounced under partial-label supervision. These results indicate that *DMIS* not only enhances the quality of samples but also produces generative distributions that more closely align with the true data distribution. Furthermore, Figure 2 compares conditionally generated samples from the *Vanilla* and *DMIS* models across different datasets. Compared to the *Vanilla* model which often produces samples misaligned with the class, our model produces images of higher visual fidelity and class-conditional generations that more accurately reflect the intended semantic categories.

## 6.2 TASK2: WEAKLY SUPERVISED LEARNING

**Setup**. We evaluate our method under three weakly supervised scenarios. In partial-label learning, we compare against approaches including *PRODEN* (Lv et al., 2020), *IDGP* (Qiao et al., 2023), *PiCO* (Wang et al., 2023), *CRDPLL* (Wu et al., 2022) and *DIRK* (Wu et al., 2024). For semi-supervised learning, we adopt *Dash* (Xu et al., 2021), *CoMatch* (Li et al., 2021a), *FlexMatch* (Zhang et al., 2021a), *SimMatch* (Zheng et al., 2022) and *SoftMatch* (Chen et al., 2023) as comparison methods. For noisy-label learning, we compare with *Coteaching* (Han et al., 2018), *ELR* (Liu et al., 2020), *PENCIL* (Yi & Wu, 2019), as well as standard normal cross-entropy *(CE)* training and *Mixup* (Zhang et al., 2018). To ensure a fair comparison, the discriminative classifier is implemented as Wide-ResNet-40-10 with 55.84M parameters, while our generative model contains 55.73M parameters, and all models are trained from scratch without pre-training.

**Results**. The classification results for weakly supervised learning are reported in Table 2. Overall, our method *DMIS*, evaluated via a diffusion classifier, achieves the best performance, demonstrating the stronger generalization capability of diffusion models over prior discriminative approaches. Interestingly, the *Vanilla* method still outperforms several baselines, particularly in the noisy-label setting, suggesting that the vanilla denoising score matching objective still acts as an implicit regularizer against label noise. Moreover, compared to standard *CE* training, the regenerate-classification variant *DMIS^CE* improves accuracy by up to 11.58%, 17.53%, and 22.13% on Fashion-MNIST, CIFAR-10, and ImageNette dataset, respectively, showing that the regenerated dataset effectively mitigates label imprecision and yields cleaner supervision for downstream discriminative training.

## 6.3 TASK3: NOISY DATASET CONDENSATION

While the task of dataset condensation has achieved remarkable progress recently, existing methods are typically developed under the assumption of clean labels. However, label noise is inevitable and cannot be fully eliminated in practice. Therefore, exploring how to condense a clean dataset from

---

[5]*DMIS^CE* denotes regenerate-classification results, i.e., we regenerate datasets of the same size under conditional sampling and then train a discriminative model on them using standard *CE* loss.

Table 3: Classification results (test accuracy, %) on noisy-label Fashion-MNIST, CIFAR-10, and ImageNette datasets. 'IPC' indicates the number of images per class in the condensed dataset. **Bold** numbers indicate the best performance.

| Dataset | Type | IPC | Random | DC | DSA | DM | MTT | RDED | SRE2L | DMIS |
|---|---|---|---|---|---|---|---|---|---|---|
| F-MNIST | Sym-40% | 10 | 34.42±0.69 | 22.85±1.69 | 42.07±2.49 | 57.06±1.52 | 9.03±3.81 | 18.57±1.06 | 15.80±0.38 | **70.18±0.37** |
| | | 50 | 52.36±0.60 | 35.64±2.26 | 55.22±1.51 | 68.23±0.47 | 10.91±0.82 | 23.19±0.74 | 19.51±0.96 | **80.73±0.07** |
| | | 100 | 55.14±0.06 | 30.46±1.74 | 41.30±0.85 | 73.21±0.69 | 13.73±3.96 | 25.43±0.21 | 19.66±1.91 | **84.26±0.02** |
| | Asym-40% | 10 | 48.28±0.34 | 53.17±1.59 | 57.15±2.37 | 63.27±1.60 | 8.75±0.82 | 18.42±1.62 | 16.45±1.96 | **65.02±1.85** |
| | | 50 | 69.44±0.17 | 49.21±0.69 | 77.20±0.34 | 76.39±0.57 | 8.76±2.11 | 22.31±0.67 | 27.07±0.35 | **79.65±0.63** |
| | | 100 | 70.80±0.91 | 36.95±0.57 | 80.24±0.54 | 78.43±0.63 | 12.59±1.22 | 24.03±0.97 | 26.52±1.46 | **83.22±0.33** |
| CIFAR-10 | Sym-40% | 10 | 16.30±0.96 | 18.11±1.02 | 18.06±1.72 | 23.71±0.40 | 12.06±0.46 | 19.85±0.88 | 13.12±1.04 | **27.83±0.98** |
| | | 50 | 26.59±0.70 | 20.63±0.22 | 28.76±0.57 | 29.50±0.56 | 17.96±2.10 | 34.64±0.58 | 14.23±1.67 | **46.47±0.41** |
| | | 100 | 31.19±0.74 | 19.91±0.54 | 29.45±0.34 | 32.26±0.75 | 18.04±3.55 | 44.03±0.21 | 14.21±0.93 | **56.53±0.03** |
| | Asym-40% | 10 | 24.89±1.65 | 18.51±1.35 | 22.23±1.80 | 26.53±0.07 | 9.62±1.45 | 23.48±0.65 | 14.64±1.03 | **24.94±0.49** |
| | | 50 | 40.95±0.59 | 25.97±0.97 | 40.81±0.29 | 43.09±0.76 | 16.54±1.88 | 39.12±0.13 | 16.03±0.21 | **47.77±0.78** |
| | | 100 | 47.49±0.64 | 27.76±0.72 | 42.96±0.84 | 51.61±0.60 | 17.67±2.53 | 44.45±0.19 | 17.55±0.91 | **55.89±0.39** |
| ImageNette | Sym-40% | 10 | 23.09±0.19 | 15.89±0.73 | 27.70±1.25 | 28.83±0.73 | 21.15±1.05 | 25.03±1.17 | | **34.36±1.05** |
| | | 50 | 33.83±0.28 | 24.62±0.73 | 32.07±1.01 | 42.66±1.27 | 38.39±1.67 | 35.87±0.39 | 35.37±0.82 | **44.93±0.28** |
| | | 100 | 40.04±0.71 | 22.81±1.22 | 36.05±1.76 | 43.25±2.13 | 39.61±1.52 | 35.87±0.39 | 41.74±1.37 | **56.23±0.84** |
| | Asym-40% | 10 | 26.54±0.88 | 19.26±0.98 | 30.62±2.09 | 33.40±0.48 | 33.65±1.29 | 26.23±0.06 | 25.74±2.21 | **37.09±0.29** |
| | | 50 | 47.91±0.61 | 31.68±2.15 | 43.41±1.24 | 50.97±1.61 | 38.71±1.24 | 32.75±0.43 | 35.29±0.14 | **55.20±0.46** |
| | | 100 | 59.10±1.41 | 29.19±0.21 | 53.79±0.84 | 60.70±1.88 | 37.69±1.29 | 35.48±0.22 | 42.37±0.34 | **68.97±0.12** |

noisy-label data is natural and meaningful. To the best of our knowledge, this is the first work to investigate dataset condensation under noisy supervision, which we term *noisy dataset condensation*.

**Setup**. During condensation, we employ our trained CDMs to synthesize images according to the specified IPC. For evaluation, we compare against both hard-label-based methods, including *DC* (Zhao et al., 2021), *DSA* (Zhao & Bilen, 2021), *DM* (Zhao & Bilen, 2023), and *MTT* (Cazenavette et al., 2022b), as well as soft-label-based methods, namely *RDED* (Sun et al., 2024) and *SRE2L* (Yin et al., 2024). Following common protocols (Sun et al., 2024; Yin et al., 2024), we adopt ResNet-18 as the backbone during condensation and evaluate the condensed datasets on a test set using ResNet-34.

**Results**. Table 3 presents the results of noisy dataset condensation, with qualitative visualizations provided in Appendix E.5. Our method consistently surpasses prior approaches across datasets and noise types. These results highlight the advantage of generative condensation: rather than memorizing noisy labels, *DMIS* implicitly denoises them during generation, leading to cleaner condensed datasets. Notably, unlike the trends observed in clean dataset condensation, distribution-matching methods (e.g., *DM*) achieve the second-best results in this noisy setting, suggesting that distribution alignment helps regularize the effect of label noise. Moreover, instance-selection methods generally outperform synthetic-generation methods (e.g., *Random* vs. *DC/DSA/MTT* and *RDED* vs. *SRE2L*), indicating that discarding noisy samples during condensation is also an effective strategy to mitigate label noise. Collectively, these findings not only demonstrate the superiority of our approach but also provide useful insights for future work on noisy dataset condensation.

## 7 CONCLUSION

In this paper, we addressed the challenge of training CDMs under imprecise supervision, a setting that frequently arises in real-world applications. We introduced a unified framework that formulates the learning problem as likelihood maximization and decomposes it into generative and classification components. Based on this formulation, we proposed a weighted denoising score matching objective that enables label-conditioned learning without clean annotations, and developed an efficient timestep sampling strategy to reduce the computational cost of posterior inference. Extensive experiments across image generation, weakly supervised learning, and noisy dataset condensation verified the effectiveness and versatility of our approach. Beyond establishing strong baselines, our work also pioneers the study of noisy dataset condensation, opening new opportunities for future exploration in robust and scalable diffusion modeling under weak supervision.

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

CONTENTS

**The Use of Large Language Models (LLMs).** LLMs were only used for language polishing and proofreading. No part of the technical content, experiments, or analysis was generated by LLMs.

# A   NOTATION AND DEFINITIONS

We present the notation table for each symbol used in this paper in Table 4.

Table 4: List of common mathematical symbols used in this paper.

| Symbol | Definition |
| --- | --- |
| $\mathbf{x}$ | A sample of training data |
| $z$ | Imprecise label associated with a sample |
| $s$ | Candidate label set for a sample |
| $y$ | Class index label |
| $c$ | Total number of classes |
| $\mathcal{X}$ | Input space from which $\mathbf{x}$ is drawn |
| $\mathcal{Y}$ | Label space from which $y$ is drawn |
| $X$ | Random variable for training instances |
| $Y$ | Random variable for true labels |
| $Z$ | Random variable for imprecise labels |
| $S$ | Random variable for partial labels |
| $\hat{Y}$ | Random variable for noisy labels |
| $X^{\mathrm{l}}$ | Set of labeled data instances |
| $X^{\mathrm{u}}$ | Set of unlabeled data instances |
| $Y^{\mathrm{l}}$ | Set of labels corresponding to $X^{\mathrm{l}}$ |
| $\emptyset$ | Empty label set |
| $\theta$ | Parameters of the diffusion model to be optimized |
| $\phi$ | Exponential moving average of $\theta$ over training iteration |
| $\mathbf{0}$ | Zero vector |
| $\mathbf{I}$ | Identity matrix |
| $\mathbf{x}_t$ | Noisy version of the sample at timestep $t$ |
| $\tau$ | Continuous timestep variable |
| $\alpha_t$ | Scaling factor at timestep $t$ |
| $\sigma_t$ | Noise scale at timestep $t$ |
| $l$ | Left boundary of a subsampled timestep interval |
| $r$ | Right boundary of a subsampled timestep interval |
| $\Delta$ | Length of a subsampled timestep interval |
| $q(\cdot)$ | Real Data distribution |
| $q(\cdot \mid \cdot)$ | Real conditional data distribution |
| $p(\cdot)$ | Marginal probability distribution |
| $p(\cdot \mid \cdot)$ | Model-infered conditional distribution |
| $F(\cdot)$ | Cumulative distribution function of $p(\cdot)$ |
| $f(\cdot)$ | Diffusion classifier |
| $\mathbf{s}(\cdot, \cdot)$ | Time-conditioned score prediction network |
| $\mathcal{N}(\cdot, \cdot)$ | Gaussian distribution |

# B   PROOF

## B.1   CONNECTIONS AMONG DIFFERENT DIFFUSION MODELS.

The diffusion model we define in this paper can be reformulated to align with other common diffusion frameworks, such as DDPM (Ho et al., 2020), SMLD (Song & Ermon, 2019), VE-SDE (Song et al., 2020) and VP-SDE (Song et al., 2020), as well as with approaches like x-prediction (Ho et al., 2020), v-prediction (Salimans & Ho, 2022), and $\epsilon$-prediction (Ho et al., 2020). This demonstrates that our formulation is compatible with diverse diffusion paradigms while facilitating unified theoretical analysis. To better demonstrate this transformation, we present the following pseudocodes.

---

**Algorithm 1** Our models to EDM

---

**Require:** A score network $\mathbf{s}_\theta$, a noisy input $\mathbf{x}_t$, noise level $t$, linear schedule $\{\alpha_i\}_{i=1}^{T}$ and $\{\sigma_i\}_{i=1}^{T}$.

1: Calculate the denoised image $\mathbf{x}_0$ using $\mathbf{s}_\theta$: $\mathbf{x}_0 = (\mathbf{x}_t + \sigma_t^2 \mathbf{s}_\theta(\mathbf{x}_t/\alpha_t, \sigma_t/\alpha_t))/\alpha_t$

2: **if** performing $\mathbf{x}_0$-prediction **then**

3:      **return** $\mathbf{x}_0$.

4: **end if**

5: Calculate the noise component $\epsilon$: $\epsilon = \frac{\mathbf{x}_t - \alpha_t \mathbf{x}_0}{\sigma_t}$

6: **if** performing $\epsilon$-prediction **then**

7:      **return** $\epsilon$.

8: **end if**

9: Calculate the noise component $v$: $v = \alpha_t \epsilon - \sigma_t \mathbf{x}_0$

10: **if** performing $v$-prediction **then**

11:      **return** $v$.

12: **end if**

---

**DDPM.** DDPM define a sequence $\{\beta_t\}_{t=0}^{T}$ and $\mathbf{x}_t = \sqrt{\prod_{i=0}^{t}(1-\beta_i)}\mathbf{x}_0 + \sqrt{1 - \prod_{i=0}^{t}(1-\beta_i)}\epsilon$, which can be seen as a special case of Eq. (1) where we can set $\alpha_t = \sqrt{\prod_{i=0}^{t}(1-\beta_i)}$ and $\sigma_t = \sqrt{1 - \prod_{i=0}^{t}(1-\beta_i)}$.

**SMLD.** SMLD defines a noise schedule $\sigma(t)_{t=0}^{T}$ and $\mathbf{x}_t = \mathbf{x}_0 + \sigma(t)\epsilon$, with $\sigma(1) < \sigma(2) < \cdots < \sigma(T)$. In this setup, Eq. (1) reduces to $\alpha_t = 1$, $\sigma_t = \sigma(t)$.

**VP-SDE.** VP-SDE is the continuous case of DDPM, which define a stochastic differential equation (SDE) as

$$dX_t = -\frac{1}{2}\beta(t)X_t dt + \sqrt{\beta(t)}dW_t, \quad t \in [0,1],$$

where $\beta(t) = \beta_{t \cdot T} \cdot T$. In this setup, $\alpha_t = \sqrt{\exp\left(-\int_0^t \beta(s)ds\right)}$, $\sigma_t = 1 - \exp\left(-\int_0^t \beta(s)ds\right)$.

**VE-SDE.** VE-SDE is the continuous case of SMLD, whose forward process of VE-SDE is defined as

$$dX_t = \sqrt{\frac{d\sigma(t)^2}{dt}}\, dW_t.$$

In this setup, $\alpha_t = 1$ and $\sigma_t = \sqrt{\sigma^2(t) - \sigma^2(0)}$.

While the models above each define their own specific frameworks for the diffusion process, EDM (Karras et al., 2022) proposes a unified structure and optimizes the parameters choice within the diffusion process, making it both robust and adaptable. Therefore, for our implementation, we adopt EDM as the foundational diffusion model. In EDM, the scaling and noise schedules are a special case of VE-SDE, where the variance of the noise is given by $\sigma(t) = t$. Accordingly, we use $\mathbf{s}_\theta(\mathbf{x}/\alpha_t, \sigma_t/\alpha_t)$ to obtain the predicted score, as shown in Algorithm 1.

## B.2 DERIVATION OF EQ. (5)

Maximizing the variational lower bound, or equivalently evidence lower bound (ELBO), to optimize the diffusion model is a common approach. To avoid redundant proofs, we directly use the conclusion from Eq. (58) in Luo (2022) as below:

$$\log p_\theta(\mathbf{x}) \geq \mathbb{E}_q[-D_{\mathrm{KL}}(q(\mathbf{x}_T|\mathbf{x}_0)\|p(\mathbf{x}_T)) + \log p_\theta(\mathbf{x}_0|\mathbf{x}_1) - \sum_{t>1} D_{\mathrm{KL}}(q(\mathbf{x}_{t-1}|\mathbf{x}_t,\mathbf{x}_0)\|p_\theta(\mathbf{x}_{t-1}|\mathbf{x}_t))]$$

Although each KL divergence term $D_{\mathrm{KL}}(q(\mathbf{x}_{t-1}|\mathbf{x}_t,\mathbf{x}_0)\|p_\theta(\mathbf{x}_{t-1}|\mathbf{x}_t))$ is difficult to minimize for arbitrary posteriors, we can leverage the Gaussian transition assumption to make optimization tractable. By Bayes rule, we have:

$$q(\mathbf{x}_{t-1}|\mathbf{x}_t,\mathbf{x}_0) = \frac{q(\mathbf{x}_t|\mathbf{x}_{t-1},\mathbf{x}_0)q(\mathbf{x}_{t-1}|\mathbf{x}_0)}{q(\mathbf{x}_t|\mathbf{x}_0)}$$

As we already know that $q(\mathbf{x}_t|\mathbf{x}_0)$ and $q(\mathbf{x}_{t-1}|\mathbf{x}_0)$ from Eq. (1), $q(\mathbf{x}_t|\mathbf{x}_{t-1}, \mathbf{x}_0)$ can be derived from its equivalent form $q(\mathbf{x}_t|\mathbf{x}_{t-1})$ as follows:

$$\mathbf{x}_t = \alpha_t \mathbf{x}_0 + \sigma_t \boldsymbol{\epsilon}_0$$

$$= \alpha_t \left( \frac{\mathbf{x}_{t-1} - \sigma_{t-1} \boldsymbol{\epsilon}_0^*}{\alpha_{t-1}} \right) + \sigma_t \boldsymbol{\epsilon}_0$$

$$= \frac{\alpha_t}{\alpha_{t-1}} \mathbf{x}_{t-1} + \sigma_t \boldsymbol{\epsilon}_0 - \frac{\alpha_t}{\alpha_{t-1}} \sigma_{t-1} \boldsymbol{\epsilon}_0^*$$

$$= \frac{\alpha_t}{\alpha_{t-1}} \mathbf{x}_{t-1} + \sqrt{\sigma_t^2 - \frac{\alpha_t^2}{\alpha_{t-1}^2} \sigma_{t-1}^2} \boldsymbol{\epsilon}_{t-1}$$

$$= \mathcal{N}\left( \mathbf{x}_t; \frac{\alpha_t}{\alpha_{t-1}} \mathbf{x}_{t-1}, \sigma_t^2 - \frac{\alpha_t^2}{\alpha_{t-1}^2} \sigma_{t-1}^2 \mathbf{I} \right).$$

Now, knowing the forms of $q(\mathbf{x}_t|\mathbf{x}_{t-1}, \mathbf{x}_0)$, we can proceed to calculate the form of $q(\mathbf{x}_{t-1}|\mathbf{x}_t, \mathbf{x}_0)$ by substituting into the Bayes rule expansion:

$$q(\mathbf{x}_{t-1}|\mathbf{x}_t, \mathbf{x}_0) = \frac{q(\mathbf{x}_t|\mathbf{x}_{t-1}, \mathbf{x}_0) q(\mathbf{x}_{t-1}|\mathbf{x}_0)}{q(\mathbf{x}_t|\mathbf{x}_0)}$$

$$= \frac{\mathcal{N}\left( \mathbf{x}_t; \frac{\alpha_t}{\alpha_{t-1}} \mathbf{x}_{t-1}, \sqrt{\sigma_t^2 - \frac{\alpha_t^2}{\alpha_{t-1}^2} \sigma_{t-1}^2} \mathbf{I} \right) \mathcal{N}(\mathbf{x}_{t-1}; \alpha_{t-1} \mathbf{x}_0, \sigma_{t-1} \mathbf{I})}{\mathcal{N}(\mathbf{x}_t; \alpha_t \mathbf{x}_0, \sigma_t \mathbf{I})}$$

$$\propto \exp \left\{ - \left[ \frac{(\mathbf{x}_t - \frac{\alpha_t}{\alpha_{t-1}} \mathbf{x}_{t-1})^2}{2(\sigma_t^2 - \frac{\alpha_t^2}{\alpha_{t-1}^2} \sigma_{t-1}^2)} + \frac{(\mathbf{x}_{t-1} - \alpha_{t-1} \mathbf{x}_0)^2}{2\sigma_{t-1}^2} - \frac{(\mathbf{x}_t - \alpha_t \mathbf{x}_0)^2}{2\sigma_t^2} \right] \right\}$$

$$= \exp \left\{ - \frac{1}{2} \left[ \frac{-2 \frac{\alpha_t}{\alpha_{t-1}} \mathbf{x}_t \mathbf{x}_{t-1} + (\frac{\alpha_t}{\alpha_{t-1}})^2 \mathbf{x}_{t-1}^2}{\sigma_t^2 - \frac{\alpha_t^2}{\alpha_{t-1}^2} \sigma_{t-1}^2} + \frac{\mathbf{x}_{t-1}^2 - 2\alpha_{t-1} \mathbf{x}_{t-1} \mathbf{x}_0}{\sigma_{t-1}^2} + C(\mathbf{x}_t, \mathbf{x}_0) \right] \right\}$$

$$\propto \exp \left\{ - \frac{1}{2} \left[ \left( \frac{(\frac{\alpha_t}{\alpha_{t-1}})^2}{\sigma_t^2 - \frac{\alpha_t^2}{\alpha_{t-1}^2} \sigma_{t-1}^2} + \frac{1}{\sigma_{t-1}^2} \right) \mathbf{x}_{t-1}^2 - 2 \left( \frac{\frac{\alpha_t}{\alpha_{t-1}} \mathbf{x}_t}{\sigma_t^2 - \frac{\alpha_t^2}{\alpha_{t-1}^2} \sigma_{t-1}^2} + \frac{\alpha_{t-1} \mathbf{x}_0}{\sigma_{t-1}^2} \right) \mathbf{x}_{t-1} \right] \right\}$$

$$= \exp \left\{ - \frac{1}{2} \left[ \frac{\sigma_{t-1}^2 (\frac{\alpha_t}{\alpha_{t-1}})^2 + (\sigma_t^2 - \frac{\alpha_t^2}{\alpha_{t-1}^2} \sigma_{t-1}^2)}{(\sigma_t^2 - \frac{\alpha_t^2}{\alpha_{t-1}^2} \sigma_{t-1}^2) \sigma_{t-1}^2} \mathbf{x}_{t-1}^2 - 2 \left( \frac{\frac{\alpha_t}{\alpha_{t-1}} \mathbf{x}_t}{\sigma_t^2 - \frac{\alpha_t^2}{\alpha_{t-1}^2} \sigma_{t-1}^2} + \frac{\alpha_{t-1} \mathbf{x}_0}{\sigma_{t-1}^2} \right) \mathbf{x}_{t-1} \right] \right\}$$

$$= \exp \left\{ - \frac{1}{2} \left[ \frac{\sigma_t^2}{(\sigma_t^2 - \frac{\alpha_t^2}{\alpha_{t-1}^2} \sigma_{t-1}^2) \sigma_{t-1}^2} \mathbf{x}_{t-1}^2 - 2 \left( \frac{\frac{\alpha_t}{\alpha_{t-1}} \mathbf{x}_t}{\sigma_t^2 - \frac{\alpha_t^2}{\alpha_{t-1}^2} \sigma_{t-1}^2} + \frac{\alpha_{t-1} \mathbf{x}_0}{\sigma_{t-1}^2} \right) \mathbf{x}_{t-1} \right] \right\}$$

$$= \exp \left\{ - \frac{1}{2} \left( \frac{\sigma_t^2}{(\sigma_t^2 - \frac{\alpha_t^2}{\alpha_{t-1}^2} \sigma_{t-1}^2) \sigma_{t-1}^2} \right) \left[ \mathbf{x}_{t-1}^2 - 2 \left( \frac{\frac{\frac{\alpha_t}{\alpha_{t-1}} \mathbf{x}_t}{\sigma_t^2 - \frac{\alpha_t^2}{\alpha_{t-1}^2} \sigma_{t-1}^2} + \frac{\alpha_{t-1} \mathbf{x}_0}{\sigma_{t-1}^2}}{\frac{\sigma_t^2}{(\sigma_t^2 - \frac{\alpha_t^2}{\alpha_{t-1}^2} \sigma_{t-1}^2) \sigma_{t-1}^2}} \right) \mathbf{x}_{t-1} \right] \right\}$$

$$= \exp \left\{ - \frac{1}{2} \left( \frac{1}{\frac{(\sigma_t^2 - \frac{\alpha_t^2}{\alpha_{t-1}^2} \sigma_{t-1}^2) \sigma_{t-1}^2}{\sigma_t^2}} \right) \left[ \mathbf{x}_{t-1}^2 - 2 \left( \frac{\frac{\alpha_t}{\alpha_{t-1}} \mathbf{x}_t \sigma_{t-1}^2 + (\sigma_t^2 - \frac{\alpha_t^2}{\alpha_{t-1}^2} \sigma_{t-1}^2) \alpha_{t-1} \mathbf{x}_0}{\sigma_t^2} \right) \mathbf{x}_{t-1} \right] \right\}$$

$$\propto \mathcal{N}\left( \mathbf{x}_{t-1}; \frac{\frac{\alpha_t}{\alpha_{t-1}} \mathbf{x}_t \sigma_{t-1}^2 + (\sigma_t^2 - \frac{\alpha_t^2}{\alpha_{t-1}^2} \sigma_{t-1}^2) \alpha_{t-1} \mathbf{x}_0}{\sigma_t^2}, \frac{(\sigma_t^2 - \frac{\alpha_t^2}{\alpha_{t-1}^2} \sigma_{t-1}^2) \sigma_{t-1}^2}{\sigma_t^2} \mathbf{I} \right)$$

where in the fourth Equation, $C(\mathbf{x}_t, \mathbf{x}_0)$ is a constant term with respect to $\mathbf{x}_{t-1}$ computed as a combination of only $\mathbf{x}_t$, $\mathbf{x}_0$, and $\alpha$ values. We have therefore shown that at each step, $\mathbf{x}_{t-1} \sim$

$q(\mathbf{x}_{t-1}|\mathbf{x}_t, \mathbf{x}_0)$ is normally distributed, with mean $\boldsymbol{\mu}_q(\mathbf{x}_t, \mathbf{x}_0)$ that is a function of $\mathbf{x}_t$ and $\mathbf{x}_0$, and variance $\boldsymbol{\Sigma}_q(t)$ as a function of $\alpha$ and $\sigma$ coefficients. These coefficients are known and fixed at each timestep; they are either set permanently when modeled as hyperparameters, or treated as the current inference output of a network that seeks to model them.

We can then set the variances of the two Gaussians to match exactly, optimizing the KL Divergence term reduces to minimizing the difference between the means of the two distributions:

$$\arg\min_{\theta} D_{\mathrm{KL}}(q(\mathbf{x}_{t-1}|\mathbf{x}_t, \mathbf{x}_0)\|p_{\theta}(\mathbf{x}_{t-1}|\mathbf{x}_t))$$

$$= \arg\min_{\theta} \frac{1}{2\sigma_q^2(t)} \left[\|\boldsymbol{\mu}_{\theta}(\mathbf{x}_t, t) - \boldsymbol{\mu}_q(\mathbf{x}_t, \mathbf{x}_0)\|_2^2\right], \tag{18}$$

where $\sigma_q^2(t) = \frac{(\sigma_t^2 - \frac{\alpha_t^2}{\alpha_{t-1}^2}\sigma_{t-1}^2)\sigma_{t-1}^2}{\sigma_t^2}$, the derivation is the same as in Eq. (92) in Luo (2022), so we skip the derivation here. To derive the score matching funciton, we appeal to Tweedie's Formula Efron (2011), which states $\mathbb{E}[\boldsymbol{\mu}_z|z] = z + \boldsymbol{\Sigma}_z \nabla_z \log q(z)$ for a given Gausssion variable $z \sim \mathcal{N}(z; \boldsymbol{\mu}_z, \boldsymbol{\Sigma}_z)$. In this case, we apply it to predict the true posterior mean of $\mathbf{x}_t$ given its samples. We can obtain:

$$\mathbb{E}[\boldsymbol{\mu}_{\mathbf{x}_t}|\mathbf{x}_t] = \mathbf{x}_t + \sigma_t^2 \nabla_{\mathbf{x}_t} \log q(\mathbf{x}_t) = \alpha_t \mathbf{x}_0$$

$$\therefore \mathbf{x}_0 = \frac{\mathbf{x}_t + \sigma_t^2 \nabla_{\mathbf{x}_t} \log q(\mathbf{x}_t)}{\alpha_t} \tag{19}$$

Then, we can plug Eq. (19) into our ground-truth denoising transition mean $\boldsymbol{\mu}_q(\mathbf{x}_t, \mathbf{x}_0)$ once again and derive a new form:

$$\boldsymbol{\mu}_q(\mathbf{x}_t, \mathbf{x}_0) = \frac{\frac{\alpha_t}{\alpha_{t-1}}\sigma_{t-1}^2 \mathbf{x}_t + (\sigma_t^2 - \frac{\alpha_t^2}{\alpha_{t-1}^2}\sigma_{t-1}^2)\alpha_{t-1} \cdot \frac{\mathbf{x}_t + \sigma_t^2 \nabla_{\mathbf{x}_t} \log q(\mathbf{x}_t)}{\alpha_t}}{\sigma_t^2}$$

$$= \frac{\frac{\alpha_t}{\alpha_{t-1}}\sigma_{t-1}^2}{\sigma_t^2}\mathbf{x}_t + \frac{\sigma_t^2 - \frac{\alpha_t^2}{\alpha_{t-1}^2}\sigma_{t-1}^2}{\sigma_t^2 \frac{\alpha_t}{\alpha_{t-1}}}\mathbf{x}_t + \frac{(\sigma_t^2 - \frac{\alpha_t^2}{\alpha_{t-1}^2}\sigma_{t-1}^2)\sigma_t^2 \nabla_{\mathbf{x}_t} \log q(\mathbf{x}_t)}{\sigma_t^2 \frac{\alpha_t}{\alpha_{t-1}}}$$

$$= \frac{\alpha_{t-1}}{\alpha_t}\mathbf{x}_t + \left(\frac{\alpha_{t-1}}{\alpha_t}\sigma_t^2 - \frac{\alpha_t}{\alpha_{t-1}}\sigma_{t-1}^2\right) \nabla_{\mathbf{x}_t} \log q(\mathbf{x}_t)$$

$$= \frac{\alpha_{t-1}}{\alpha_t}\left[\mathbf{x}_t + \left(\sigma_t^2 - \frac{\alpha_t^2}{\alpha_{t-1}^2}\sigma_{t-1}^2\right) \mathbf{s}_{\theta}(\mathbf{x}_t, t)\right] \tag{20}$$

Finally, we plug Eq. (20) into our optimization function Eq. (18), and we can get:

$$\arg\min_{\theta} D_{\mathrm{KL}}(q(\mathbf{x}_{t-1}|\mathbf{x}_t, \mathbf{x}_0)\|p_{\theta}(\mathbf{x}_{t-1}|\mathbf{x}_t))$$

$$= \arg\min_{\theta} \frac{1}{2\sigma_q^2(t)} \left[\|\boldsymbol{\mu}_{\theta}(\mathbf{x}_t, t) - \boldsymbol{\mu}_q(\mathbf{x}_t, \mathbf{x}_0)\|_2^2\right]$$

$$= \arg\min_{\theta} \frac{1}{2\frac{(\sigma_t^2 - \frac{\alpha_t^2}{\alpha_{t-1}^2}\sigma_{t-1}^2)\sigma_{t-1}^2}{\sigma_t^2}} \cdot (\frac{\alpha_{t-1}}{\alpha_t})^2 \cdot (\sigma_t^2 - \frac{\alpha_t^2}{\alpha_{t-1}^2}\sigma_{t-1}^2)^2 \|\mathbf{s}_{\theta}(\mathbf{x}_t, t) - \nabla_{\mathbf{x}_t} \log q(\mathbf{x}_t)\|_2^2$$

$$= \frac{\sigma_t^2}{2}(\frac{\sigma_t^2 \alpha_{t-1}^2}{\sigma_{t-1}^2 \alpha_t^2} - 1)\|\mathbf{s}_{\theta}(\mathbf{x}_t, t) - \nabla_{\mathbf{x}_t} \log q(\mathbf{x}_t)\|_2^2$$

### B.3 DERIVATION OF VARITIONAL LOWER BOUND EQ. (7)

To model $\log p_{\theta}(X, Z)$, we introduce an auxiliary distribution $Q(Y)$ over the latent variable $Y$:

$$\log p_{\theta}(X, Z) = \int Q(Y) \log p_{\theta}(X, Z) dY$$

$$= \int Q(Y) \log p_{\theta}(X, Z) \frac{p_{\theta}(Y|X, Z)}{p_{\theta}(Y|X, Z)} dY$$

$$= \int Q(Y) \log \frac{p_{\theta}(X, Y, Z)}{Q(Y)} dY - \int Q(Y) \log \frac{p_{\theta}(Y|X, Z)}{Q(Y)} dY,$$

where the first term is the ELBO and the second term is the KL divergence $\mathcal{D}_{\mathrm{KL}}(Q(Y)\|p_\theta(Y|X,Z))$. Since the KL divergence is non-negative, maximizing the ELBO provides a valid surrogate for maximizing $\log p_\theta(X,Z)$. Replacing $Q(Y)$ with $p_\phi(Y|X,Z)$ at each iteration will obtain as follows:

$$
\begin{aligned}
\theta^* =& \arg\max_\theta \ \log p_\theta(X,Z) \\
=& \arg\max_\theta \ \mathbb{E}_{p_\phi(Y|X,Z)}[\log p_\theta(X,Y,Z)] \\
=& \arg\max_\theta \ \mathbb{E}_{p_\phi(Y|X,Z)}[\log p_\theta(X|Z) + \log p_\theta(Y|X,Z) + \log p_\theta(Z)] \\
=& \arg\max_\theta \ \mathbb{E}_{p_\phi(Y|X,Z)}[\log p_\theta(X|Z)] + \mathbb{E}_{p_\phi(Y|X,Z)}[\log p_\theta(Y|X,Z)] \\
=& \arg\max_\theta \ \log p_\theta(X|Z) + \mathbb{E}_{p_\phi(Y|X,Z)}[\log p_\theta(Y|X,Z)].
\end{aligned}
$$

which is exactly the variational lower bound presented in Eq. (7).

## B.4    DERIVATION OF CONDITIONAL ELBO IN EQ. (8)

We provide a derivation of conditional ELBO in the following, which is similar to the unconditional ELBO in Ho et al. (2020).

$$
\begin{aligned}
& \log p_\theta\left(\mathbf{x}_0 \,|\, z\right) \\
=& \log \int \frac{p_\theta\left(\mathbf{x}_{0:T} \,|\, z\right) q\left(\mathbf{x}_{1:T} \,|\, \mathbf{x}_0, z\right)}{q\left(\mathbf{x}_{1:T} \,|\, \mathbf{x}_0, z\right)} d\mathbf{x}_{1:T} \\
=& \log \mathbb{E}_{q(\mathbf{x}_{1:T}|\mathbf{x}_0,z)}\left[\frac{p_\theta\left(\mathbf{x}_T \,|\, z\right) p_\theta\left(\mathbf{x}_{0:T-1} \,|\, \mathbf{x}_T, z\right)}{q\left(\mathbf{x}_{1:T} \,|\, \mathbf{x}_0, z\right)}\right] \\
\geq& \mathbb{E}_{q(\mathbf{x}_{1:T}|\mathbf{x}_0,z)}\left[\log \frac{p_\theta\left(\mathbf{x}_T \,|\, z\right) p_\theta\left(\mathbf{x}_{0:T-1} \,|\, \mathbf{x}_T, z\right)}{q\left(\mathbf{x}_{1:T} \,|\, \mathbf{x}_0, z\right)}\right] \\
=& \mathbb{E}_{q(\mathbf{x}_{1:T}|\mathbf{x}_0,z)}\left[\log \frac{p_\theta\left(\mathbf{x}_T \,|\, z\right) \prod_{i=0}^{T-1} p_\theta\left(\mathbf{x}_i \,|\, \mathbf{x}_{i+1}, z\right)}{\prod_{i=0}^{T-1} q\left(\mathbf{x}_{i+1} \,|\, \mathbf{x}_i, \mathbf{x}_0, z\right)}\right] \\
=& \mathbb{E}_{q(\mathbf{x}_{1:T}|\mathbf{x}_0,z)}\left[\log \frac{p_\theta\left(\mathbf{x}_T \,|\, z\right) \prod_{i=0}^{T-1} p_\theta\left(\mathbf{x}_i \,|\, \mathbf{x}_{i+1}, z\right)}{\prod_{i=0}^{T-1} \frac{q(\mathbf{x}_{i+1}|\mathbf{x}_0,z)q(\mathbf{x}_i|\mathbf{x}_{i+1},\mathbf{x}_0,z)}{q(\mathbf{x}_i|\mathbf{x}_0,z)}}\right] \\
=& \mathbb{E}_{q(\mathbf{x}_{1:T}|\mathbf{x}_0,z)}\left[\log \frac{p_\theta\left(\mathbf{x}_T \,|\, z\right) \prod_{i=0}^{T-1} p_\theta\left(\mathbf{x}_i \,|\, \mathbf{x}_{i+1}, z\right)}{\prod_{i=0}^{T-1} q\left(\mathbf{x}_i \,|\, \mathbf{x}_{i+1}, \mathbf{x}_0, z\right)} - \log q\left(\mathbf{x}_T \,|\, \mathbf{x}_0, z\right)\right] \\
=& \mathbb{E}_{q(\mathbf{x}_{1:T}|\mathbf{x}_0,z)}\left[\log \frac{\prod_{i=0}^{T-1} p_\theta\left(\mathbf{x}_i \,|\, \mathbf{x}_{i+1}, z\right)}{\prod_{i=0}^{T-1} q\left(\mathbf{x}_i \,|\, \mathbf{x}_{i+1}, \mathbf{x}_0, z\right)} - \log \frac{q\left(\mathbf{x}_T \,|\, \mathbf{x}_0, z\right)}{p_\theta\left(\mathbf{x}_T \,|\, z\right)}\right] \\
=& \sum_{i=0}^{T-1} \mathbb{E}_{q(\mathbf{x}_i,\mathbf{x}_{i+1}|\mathbf{x}_0,z)}\left[\log \frac{p_\theta\left(\mathbf{x}_i \,|\, \mathbf{x}_{i+1}, z\right)}{q\left(\mathbf{x}_i \,|\, \mathbf{x}_{i+1}, \mathbf{x}_0, z\right)}\right] - D_{\mathrm{KL}}\left(q\left(\mathbf{x}_T \,|\, \mathbf{x}_0, z\right) \| p_\theta\left(\mathbf{x}_T \,|\, z\right)\right) \\
=& \sum_{i=0}^{T-1} \mathbb{E}_{q(\mathbf{x}_{i+1}|\mathbf{x}_0,z)} \mathbb{E}_{q(\mathbf{x}_i|\mathbf{x}_{i+1},\mathbf{x}_0,z)}\left[\log \frac{p_\theta\left(\mathbf{x}_i \,|\, \mathbf{x}_{i+1}, z\right)}{q\left(\mathbf{x}_i \,|\, \mathbf{x}_{i+1}, \mathbf{x}_0, z\right)}\right] - D_{\mathrm{KL}}\left(q\left(\mathbf{x}_T \,|\, \mathbf{x}_0, z\right) \| p_\theta\left(\mathbf{x}_T \,|\, z\right)\right) \\
=& C_3 - \sum_{i=1}^{T-1} \mathbb{E}_{q(\mathbf{x}_{i+1}|\mathbf{x}_0,z)}\left[D_{\mathrm{KL}}\left(q\left(\mathbf{x}_i \,|\, \mathbf{x}_{i+1}, \mathbf{x}_0, z\right) \| p_\theta\left(\mathbf{x}_i \,|\, \mathbf{x}_{i+1}, z\right)\right)\right] \\
=& -\mathbb{E}_t\left[w_t \left\|\mathbf{s}_\theta\left(\mathbf{x}_t, z, t\right) - \nabla \log q(\mathbf{x}_t \,|\, \mathbf{x}_0, z)\right\|_2^2\right] + C_2.
\end{aligned}
$$

We get the result of Eq. (8).

## B.5    DERIVATION OF REMARK 1

Although this result follows directly from prior studies (Vincent, 2011; Song & Ermon, 2019), we provide a brief derivation here for completeness. Let $\mathcal{L}_{\mathrm{DSM}}(\theta; q(X,Y))$ and $\mathcal{L}_{\mathrm{ESM}}(\theta; q(X,Y))$ denote

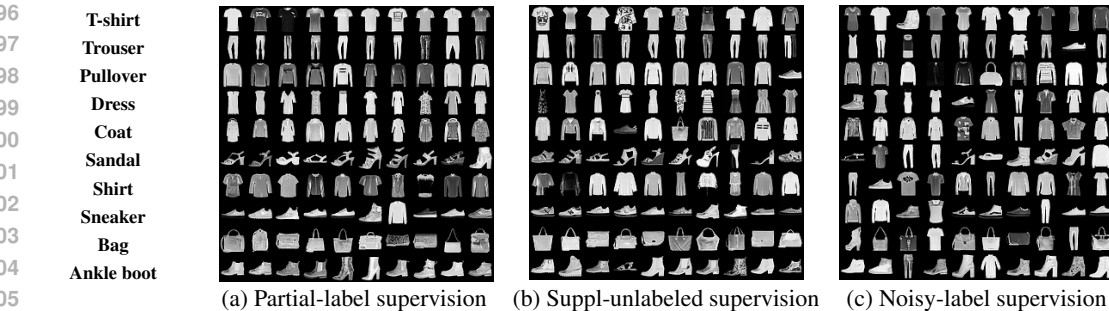

|   |   |   |
|---|---|---|
| (a) Partial-label supervision | (b) Suppl-unlabeled supervision | (c) Noisy-label supervision |

Figure 3: Examples of randomly generated Fashion-MNIST images from *Vanilla* models trained under different types of imprecise supervision.

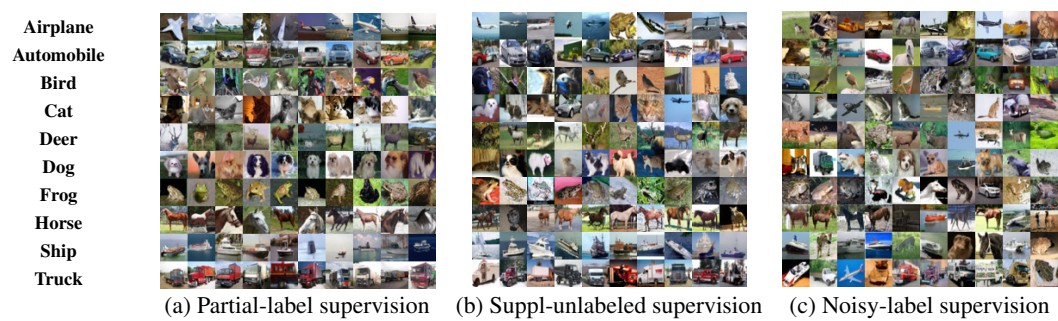

|   |   |   |
|---|---|---|
| (a) Partial-label supervision | (b) Suppl-unlabeled supervision | (c) Noisy-label supervision |

Figure 4: Examples of randomly generated CIFAR-10 images from *Vanilla* models trained under different types of imprecise supervision.

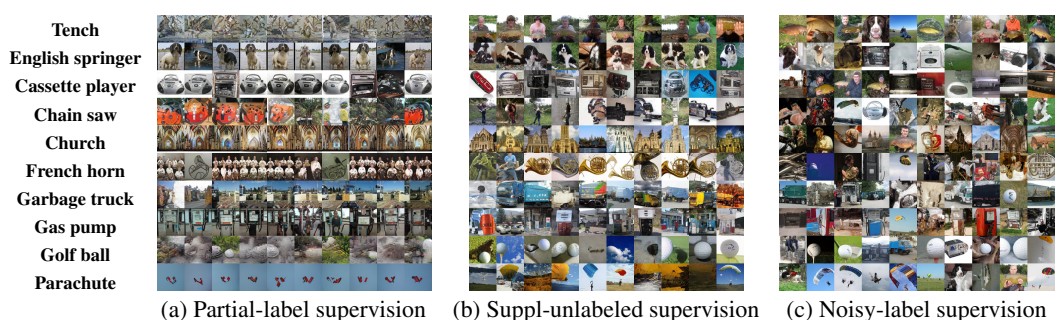

|   |   |   |
|---|---|---|
| (a) Partial-label supervision | (b) Suppl-unlabeled supervision | (c) Noisy-label supervision |

Figure 5: Examples of randomly generated ImageNette images from *Vanilla* models trained under different types of imprecise supervision.

the denoising score matching (DSM) and explicit score matching (ESM) objectives, respectively:

$$\mathcal{L}_{\text{DSM}}(\theta; q(X,Y)) := \mathbb{E}_t\left[\lambda(t)\mathbb{E}_{y \sim q(Y)}\mathbb{E}_{\mathbf{x}_t \sim q_{t|0}(\mathbf{x}_t|\mathbf{x},y)}\big\|\mathbf{s}_\theta(\mathbf{x}_t, y, t) - \nabla_{\mathbf{x}_t}\log q_{t|0}(\mathbf{x}_t \,|\, \mathbf{x}, Y = y)\big\|_2^2\right],$$

$$\mathcal{L}_{\text{ESM}}(\theta; q(X,Y)) := \mathbb{E}_t\left[\lambda(t)\mathbb{E}_{y \sim q(Y)}\mathbb{E}_{\mathbf{x}_t \sim q_t(\mathbf{x}_t|y)}\big\|\mathbf{s}_\theta(\mathbf{x}_t, y, t) - \nabla_{\mathbf{x}_t}\log q_t(\mathbf{x}_t \,|\, Y = y)\big\|_2^2\right].$$

It has been established (Vincent, 2011; Song & Ermon, 2019) that these two formulations differ only by an additive constant independent of $\theta$:

$$\mathcal{L}_{\text{ESM}}(\theta; q(X,Y)) = \mathcal{L}_{\text{DSM}}(\theta; q(X,Y)) + C_3,$$

where $C_3$ does not depend on $\theta$. Hence, both objectives admit the same minimizer.

Applying this result to an imprecise-label dataset by identifying $Y = Z$, let $\theta^*_{\text{ESM}} := \arg\min_\theta \mathcal{L}_{\text{ESM}}(\theta; q(X,Z))$. Then the optimal score function satisfies $\mathbf{s}_{\theta^*_{\text{ESM}}}(\mathbf{x}_t, z, t) = \nabla_{\mathbf{x}_t}\log q_t(\mathbf{x}_t \,|\, Z = z)$. Since the same conclusion holds for $\mathcal{L}_{\text{DSM}}$, we obtain $\mathbf{s}_{\theta^*_{\text{ESM}}} = \mathbf{s}_{\theta^*_{\text{DSM}}} = \nabla_{\mathbf{x}_t}\log q_t(\mathbf{x}_t \,|\, z)$, which is precisely the statement of Remark 1.

To directly illustrate this bias, we train CDMs under different forms of imprecise supervision by applying Eq. (8) directly, a baseline we refer to as *Vanilla*. We then visualize the images generated by these biased models, as shown in the figures below. The results reveal the following patterns:

- **Partial-label supervision:** The generated images often lack diversity and typically capture only the dominant object. This effect is particularly pronounced on the ImageNette dataset, where samples within the same class appear highly similar. Interestingly, the generated categories generally align with the ground-truth labels, suggesting that diffusion models can still extract correct class information under partial-label supervision. However, the inherent label ambiguity prevents the model from capturing intra-class variation.

- **Noisy-label supervision:** The generated samples tend to contain visual noise. Although the model is able to capture class diversity, corrupted labels cause mismatches between generated samples and their true categories.

- **Supplementary-unlabeled supervision:** The generated images are often both less diverse and noisier. This phenomenon combines the limitations of partial-label supervision with the challenge of abundant unlabeled samples. Because the model has limited access to labeled examples, it relies on averaging confidence across all classes, which reduces its discriminative boundaries and introduces noise.

### B.6 PROOF OF THEOREM 1

The derivation here is analogous to that of Theorem 1 in Na et al. (2024), and we provide the full proof below for completeness. First, for all $t$, the perturbed distribution $q_t(\mathbf{x}_t|z)$ satisified:

$$q_t(\mathbf{x}_t|z) = \sum_{y=1}^{c} p(y|z) q_t(\mathbf{x}_t|y) \quad \forall \mathbf{x}_t \in \mathcal{X}, z \subset \mathcal{Y}.$$

This implies that the transition from imprecise labels to clean labels is independent of the timesteps. Consequently, Eq. (9) can be derived as follows,

$$\nabla_{\mathbf{x}_t} \log q_t(\mathbf{x}_t|z)$$
$$= \frac{\nabla_{\mathbf{x}_t} q_t(\mathbf{x}_t|z)}{q_t(\mathbf{x}_t|z)}$$
$$= \frac{\sum_{y=1}^{c} p(y|z) \nabla_{\mathbf{x}_t} q_t(\mathbf{x}_t|y)}{q_t(\mathbf{x}_t|z)}$$
$$= \sum_{y=1}^{c} \frac{p(y|z) q_t(\mathbf{x}_t|y)}{q_t(\mathbf{x}_t|z)} \cdot \frac{\nabla_{\mathbf{x}_t} q_t(\mathbf{x}_t|y)}{q_t(\mathbf{x}_t|y)}$$
$$= \sum_{y=1}^{c} \frac{p(y|z) q_t(\mathbf{x}_t|y)}{q_t(\mathbf{x}_t|z)} \cdot \nabla_{\mathbf{x}_t} \log q_t(\mathbf{x}_t|y)$$
$$= \sum_{y=1}^{c} p(y|z) \cdot \frac{p(z)}{p(y)} \cdot \frac{p(y|\mathbf{x}_t)}{p(z|\mathbf{x}_t)} \cdot \frac{q_t(\mathbf{x}_t)}{q_t(\mathbf{x}_t)} \cdot \nabla_{\mathbf{x}_t} \log q_t(\mathbf{x}_t|y)$$
$$= \sum_{y=1}^{c} p(z|y) \cdot \frac{p(y|\mathbf{x}_t)}{p(z|\mathbf{x}_t)} \cdot \nabla_{\mathbf{x}_t} \log q_t(\mathbf{x}_t|y)$$
$$= \sum_{y=1}^{c} p(z|y, \mathbf{x}_t) \cdot \frac{p(y|\mathbf{x}_t)}{p(z|\mathbf{x}_t)} \cdot \nabla_{\mathbf{x}_t} \log q_t(\mathbf{x}_t|y) \quad \text{(Conditional indep. of } z \text{ and } \mathbf{x}_t \text{ given } y.)$$
$$= \sum_{y=1}^{c} \frac{p(z|y, \mathbf{x}_t) p(y|\mathbf{x}_t)}{p(z|\mathbf{x}_t)} \cdot \nabla_{\mathbf{x}_t} \log q_t(\mathbf{x}_t|y)$$
$$= \sum_{y=1}^{c} p(y|\mathbf{x}_t, z) \nabla_{\mathbf{x}_t} \log q_t(\mathbf{x}_t|y)$$

### B.7 PROOF OF PROPOSITION 1

By Remark 1 and Theorem 1, the optimal solution $\theta_{\text{Gen}}^*$ to Eq. (10) satisfies

$$\sum_{y=1}^{c} p(y \,|\, \mathbf{x}_t, z) \, \mathbf{s}_{\theta_{\text{Gen}}^*}(\mathbf{x}_t, y, t) = \nabla_{\mathbf{x}_t} \log q_t(\mathbf{x}_t \,|\, z) = \sum_{y=1}^{c} p(y \,|\, \mathbf{x}_t, z) \, \nabla_{\mathbf{x}_t} \log q_t(\mathbf{x}_t \,|\, y),$$

for all $\mathbf{x}_t \in \mathcal{X}$, $z \subseteq \mathcal{Y}$, and $t \in [T]$.

Next, recall the weighted denoising score matching loss:

$$\mathcal{L}_{\text{Gen}}(\theta) = \mathbb{E}_t \left[ w_t \left\| \sum_{y=1}^{c} p(y \,|\, \mathbf{x}_t, z) \, \mathbf{s}_{\theta}(\mathbf{x}_t, y, t) - \sum_{y=1}^{c} p(y \,|\, \mathbf{x}_t, z) \, \nabla_{\mathbf{x}_t} \log q_t(\mathbf{x}_t \,|\, y) \right\|_2^2 \right].$$

Differentiating with respect to $\mathbf{s}_{\theta}(\mathbf{x}_t, y, t)$ and setting the derivative to zero yields

$$\frac{\partial}{\partial \mathbf{s}_{\theta}(\mathbf{x}_t, y, t)} \mathcal{L}_{\text{Gen}}(\theta) = 2 w_t \, p(y \,|\, \mathbf{x}_t, z) \left( \mathbf{s}_{\theta_{\text{Gen}}^*}(\mathbf{x}_t, y, t) - \nabla_{\mathbf{x}_t} \log q_t(\mathbf{x}_t \,|\, y) \right) = 0.$$

Since $w_t > 0$, for any $y$ such that $p(y \,|\, \mathbf{x}_t, z) > 0$, the optimality condition implies

$$\mathbf{s}_{\theta_{\text{Gen}}^*}(\mathbf{x}_t, y, t) = \nabla_{\mathbf{x}_t} \log q_t(\mathbf{x}_t \,|\, y).$$

In particular, under the partial-label learning setting, if $p(y \,|\, \mathbf{x}_t, z) = 0$, the loss does not depend on $\mathbf{s}_{\theta}(\mathbf{x}_t, y, t)$, and the equality can be established without loss of generality. This completes the proof.

### B.8 PROOF OF THEOREM 2

We first consider the case where the timestep $\tau$ is sampled from a log-normal distribution, as defined in the EDM framework. Specifically,

$$\ln(\tau) \sim \mathcal{N}(P_{\text{mean}}, P_{\text{std}}^2),$$

where the parameters are set to $P_{\text{mean}} = 1.2$ and $P_{\text{std}} = -1.2$. Accordingly, the probability density function of $\tau$ is given by

$$p(\tau) = \frac{1}{\tau \, P_{\text{std}} \sqrt{2\pi}} \exp\left( -\frac{(\ln \tau - P_{\text{mean}})^2}{2 P_{\text{std}}^2} \right), \quad \tau > 0.$$

The corresponding cumulative distribution function (CDF) is denoted as:

$$F(\tau) = \frac{1}{2} \left[ 1 + \text{erf}\left( \frac{\ln \tau - P_{\text{mean}}}{P_{\text{std}} \sqrt{2}} \right) \right],$$

where $\text{erf}(x)$ denotes the error function.

The median of this distribution $\tau_{\text{mid}}$ is the value at which the CDF equals 0.5, i.e., $F(\tau_{\text{mid}}) = 0.5$.

To ensure that the selected subinterval allows signal-dominant early timesteps and noise-dominant later timesteps to complement each other, we require the cumulative probability mass on either side of the median to be equal. Formally, for subinterval boundaries $(l, r)$ with $r = l + \Delta$, we enforce the following symmetry condition:

$$F(r) - F(\tau_{\text{mid}}) = F(\tau_{\text{mid}}) - F(l).$$

Rewriting this with $r = l + \Delta$ gives:

$$F(l + \Delta) + F(l) = 2 F(\tau_{\text{mid}}) = 1.$$

This implicit equation defines the subinterval $(l, l + \Delta)$ such that the cumulative probability mass is centered around the median of $p(\tau)$. To compute the left boundary $l$, we solve:

$$l = \text{Solve}_{\tau} \left( F(\tau) + F(\tau + \Delta) - 1 = 0 \right), \tag{21}$$

and set $r = l + \Delta$. The solution can be obtained using any standard root-finding algorithm, such as the Brent method (Brent, 2013).

We then consider the DDPM setting, where the timestep $\tau$ is uniformly sampled from a fixed interval. Specifically, we assume $\tau \sim \mathcal{U}(0,1)$, whose CDF is given by

$$F(\tau) = \tau$$

Under this distribution, the symmetry condition in Eq. (21) simplifies to

$$
\begin{aligned}
l &= \text{Solve}_\tau \left( F(\tau) + F(\tau + \Delta) - 1 = 0 \right) \\
&= \text{Solve}_\tau \left( \tau + \tau + \Delta - 1 = 0 \right) \\
&= \frac{1 - \Delta}{2},
\end{aligned}
$$

and thus $r = l + \Delta = \frac{1-\Delta}{2}$. This result implies that the optimal subinterval is symmetric around the midpoint of the distribution. In the special case where only a single timestep is used (i.e., $\Delta \to 0$), the best estimate of the conditional ELBO occurs exactly at the median. As the sampled timestep deviates further from the midpoint, classification accuracy tends to degrade. This observation aligns with the empirical findings of Li et al. (2023), who reported that classification accuracy is maximized near the median and declines towards the edges. Their use of evenly spaced timesteps centered around the median further supports our strategy.

### B.9 PROOF OF THEOREM 3

For clarity, we abbreviate $\hbar(\tau, y)$ as $\hbar(\tau)$, since the proof does not depend explicitly on $y$. Define the weighted integral of $\hbar$ and the normalization factor over an interval $[l, r]$ as

$$A(l, r) := \int_l^r \hbar(\tau)\, p(\tau)\, \mathrm{d}\tau, \qquad Z(l, r) := \int_l^r p(\tau)\, \mathrm{d}\tau,$$

so that the local expectation can be written as $\mu' = A(l,r)/Z(l,r)$. Let $\mu'' = \mathbb{E}_{\tau \sim p(\tau)}[\hbar(\tau)]$ denote the global expectation. The squared error objective in Eq. (16) then becomes

$$g(l, r) := \left( \mu' - \mu'' \right)^2,$$

subject to the probability-mass constraint $Z(l, r) = \alpha$.

We apply the method of Lagrange multipliers with

$$L(l, r, \lambda) := \left( \mu' - \mu'' \right)^2 + \lambda \left( \int_l^r p(\tau)\, \mathrm{d}\tau - \alpha \right).$$

By the Leibniz rule, the derivatives of $A(l, r)$ and $Z(l, r)$ with respect to the interval boundaries are

$$\frac{\partial A(l, r)}{\partial l} = -\hbar(l)\, p(l), \quad \frac{\partial A(l, r)}{\partial r} = \hbar(r)\, p(r), \quad \frac{\partial Z(l, r)}{\partial l} = -p(l), \quad \frac{\partial Z(l, r)}{\partial r} = p(r).$$

Hence, the derivatives of $\mu' = A/Z$ are

$$\frac{\partial \mu'}{\partial l} = \frac{p(l)}{Z(l, r)}\left( \mu' - \hbar(l) \right), \qquad \frac{\partial \mu'}{\partial r} = \frac{p(r)}{Z(l, r)}\left( \hbar(r) - \mu' \right).$$

Differentiating $L$ w.r.t. $l$ and $r$ gives

$$\frac{\partial L}{\partial l} = 2(\mu' - \mu'') \cdot \frac{p(l)}{Z(l, r)}\left( \mu' - \hbar(l) \right) - \lambda p(l),$$

$$\frac{\partial L}{\partial r} = 2(\mu' - \mu'') \cdot \frac{p(r)}{Z(l, r)}\left( \hbar(r) - \mu' \right) + \lambda p(r).$$

Setting both derivatives to zero yields the necessary conditions

$$2(\mu' - \mu'')(\mu' - \hbar(l)) = \lambda Z(l, r), \qquad 2(\mu' - \mu'')(\hbar(r) - \mu') = \lambda Z(l, r).$$

Equating the two expressions gives

$$\mu' - \hbar(l) = \hbar(r) - \mu' \quad \implies \quad \mu' = \tfrac{1}{2}\left( \hbar(l) + \hbar(r) \right).$$

Substituting back, we obtain

$$\int_l^r p(\tau)\,\hbar(\tau,y)\,\mathrm{d}\tau = \frac{Z(l,r)}{2}\big(\hbar(l,y)+\hbar(r,y)\big), \qquad Z(l,r):=\int_l^r p(\tau)\,\mathrm{d}\tau.$$

Equivalently, the necessary optimality condition is $\text{ERR}(l^*,r^*,y)=0$. Since $Z(l^*,r^*)>0$, this is also equivalent to

$$\text{ERR}(l^*,r^*,y) := \mathbb{E}_{\tau\sim p(\tau\mid\tau\in[l^*,r^*])}[\hbar(\tau,y)] \; - \; \tfrac{1}{2}\big(\hbar(l^*,y)+\hbar(r^*,y)\big) = 0.$$

This establishes the necessary condition for an optimal subinterval.

## C  DISCCUSION

### C.1  ANALYSIS OF EARLY-LEARNING REGULARIZATION IN EQ. (15)

The effectiveness of Eq. (15) can be better understood by examining the form of its gradient. For clarity, we restate the loss with the following notation: given a noisy-labeled input $(\mathbf{x},\hat{y})$, we denote the model's output probabilities as $f_\theta(\mathbf{x})$ and the corresponding EMA target as $f_\phi(\mathbf{x})$.

Let $\hat{\mathbf{y}}\in\mathbb{R}^c$ be the one-hot vector corresponding to the noisy label $\hat{y}$. Then the loss over the whole dataset $\mathcal{D}=\{(\mathbf{x}^{[i]},\hat{\mathbf{y}}^{[i]})\}_{i=1}^n$ can be computed according to Eq. (15) as

$$\mathcal{L}_{\text{Cls}}^{\text{NL}}(\mathcal{D}) = -\frac{1}{n}\sum_{i=1}^n \big\langle \text{sg}(\mathbf{r}^{[i]}),\,\log f_\theta(\mathbf{x}^{[i]})\big\rangle, \quad \mathbf{r}^{[i]}=\hat{\mathbf{y}}^{[i]}-\lambda\frac{f_\theta(\mathbf{x}^{[i]})\odot\big(\delta^{[i]}\mathbf{1}-f_\phi(\mathbf{x}^{[i]})\big)}{1-\delta^{[i]}}, \quad (22)$$

where $\delta^{[i]}=\langle f_\theta(\mathbf{x}^{[i]}),f_\phi(\mathbf{x}^{[i]})\rangle$, $\text{sg}(\cdot)$ denotes the stop-gradient operator, and $\odot$ is the Hadamard product. By construction $\mathbf{r}^{[i]}$ is treated as a *constant* w.r.t. $\theta$ due to the stop-gradient.

**Lemma 1.** *Let $\psi_\theta(\mathbf{x})$ denote the pre-softmax logits such that $f_\theta(\mathbf{x})=\text{softmax}(\psi_\theta(\mathbf{x}))$. For the loss in Eq. (15), the gradients are*

$$\frac{\partial\mathcal{L}_{\text{Cls}}^{\text{NL}}(\mathbf{x}^{[i]})}{\partial\psi_\theta(\mathbf{x}^{[i]})} = f_\theta(\mathbf{x}^{[i]})-\text{sg}\big(\mathbf{r}^{[i]}\big), \quad \text{for each } i=1,\ldots,n, \quad (23)$$

*and, by the chain rule,*

$$\nabla_\theta\mathcal{L}_{\text{Cls}}^{\text{NL}}(\mathcal{D}) = \frac{1}{n}\sum_{i=1}^n J_{\mathbf{z}_\theta}(\mathbf{x}^{[i]})^\top\Big[f_\theta(\mathbf{x}^{[i]})-\text{sg}(\mathbf{r}^{[i]})\Big], \quad (24)$$

*where $J_{\mathbf{z}_\theta}(\mathbf{x})=\partial\mathbf{z}_\theta(\mathbf{x})/\partial\theta$ is the Jacobian of the logits w.r.t. the parameters. Equivalently, expanding $\mathbf{r}^{[i]}$ gives*

$$\nabla_\theta\mathcal{L}_{\text{Cls}}^{\text{NL}}(\mathcal{D}) = \frac{1}{n}\sum_{i=1}^n J_{\mathbf{z}_\theta}(\mathbf{x}^{[i]})^\top\left[f_\theta(\mathbf{x}^{[i]})-\hat{\mathbf{y}}^{[i]}+\lambda\,\text{sg}\Big(\frac{f_\theta(\mathbf{x}^{[i]})\odot\big(\delta^{[i]}\mathbf{1}-f_\phi(\mathbf{x}^{[i]})\big)}{1-\delta^{[i]}}\Big)\right]. \quad (25)$$

*Proof.* For any $i\in\{1,\ldots,n\}$, let us first verify that $\mathbf{r}^{[i]}$ sums to 1. With

$$\mathbf{r}^{[i]}=\hat{\mathbf{y}}^{[i]}-\lambda\frac{f_\theta(\mathbf{x}^{[i]})\odot\big(\delta^{[i]}\mathbf{1}-f_\phi(\mathbf{x}^{[i]})\big)}{1-\delta^{[i]}},$$

we sum over classes and using $\langle f_\theta(\mathbf{x}^{[i]}),\mathbf{1}\rangle=1$ yields

$$\mathbf{1}^\top\mathbf{r}^{[i]}=1-\lambda\frac{\delta^{[i]}-\langle f_\theta(\mathbf{x}^{[i]}),f_\phi(\mathbf{x}^{[i]})\rangle}{1-\delta^{[i]}}=1,$$

so $\mathbf{r}^{[i]}$ lies on the simplex (hence Eq. (22) is an ordinary cross-entropy with a fixed target). Let $\psi^{[i]}=\psi_\theta(\mathbf{x}^{[i]})$ be the logits and recall $\frac{\partial\log\text{softmax}(\psi)}{\partial\psi}=I-\text{softmax}(\psi)\mathbf{1}^\top$. For the per-sample loss $\ell^{[i]}=-\langle\text{sg}(\mathbf{r}^{[i]}),\log\text{softmax}(\psi^{[i]})\rangle$, the derivative w.r.t. logits is

$$\frac{\partial\ell^{[i]}}{\partial\psi^{[i]}}=\text{softmax}(\psi^{[i]})-\text{sg}(\mathbf{r}^{[i]})=f_\theta(\mathbf{x}^{[i]})-\text{sg}(\mathbf{r}^{[i]}),$$

which is Eq. (23). Applying the chain rule and averaging over $i$ gives Eq. (24). Replacing $\text{sg}(\mathbf{r}^{[i]})$ by its explicit form produces Eq. (25). $\qquad\qquad\square$

**Remark.** Eq. (25) shows that $\mathcal{L}_{\mathrm{Cls}}^{\mathrm{NL}}$ behaves like the standard cross-entropy gradient plus an ELR-like corrective term. This term amplifies gradients on clean samples and counteracts gradients on noisy samples. Specifically, we expand this ELR-like corrective term into:

$$\mathbf{g}_y^{[i]} := \frac{f_\theta(\mathbf{x}^{[i]})}{1 - \langle f_\theta(\mathbf{x}^{[i]}), f_\phi(\mathbf{x}^{[i]}) \rangle} \sum_{k=1}^{c} (f_\phi(\mathbf{x}^{[i]})_k - f_\phi(\mathbf{x}^{[i]})_y) f_\theta(\mathbf{x}^{[i]})_k. \tag{26}$$

If $y^*$ is the true class, then the $y^*$th entry of $f_\phi(\mathbf{x}^{[i]})$ tends to be dominant during early-learning. In that case, the $y^*$th entry of $\mathbf{g}^{[i]}$ is negative. This is useful both for examples with clean labels and for examples with noisy labels. For examples with clean labels, the cross-entropy term $f_\theta(\mathbf{x}^{[i]}) - \hat{\mathbf{y}}^{[i]}$ tends to vanish after the early-learning stage because $f_\theta(\mathbf{x}^{[i]})$ is very close to $\hat{\mathbf{y}}^{[i]}$, allowing examples with wrong labels to dominant the gradient. Adding $\mathbf{g}^{[i]}$ counteracts this effect by ensuring that the magnitudes of the coefficients on examples with clean labels remain large. Thus, $\mathbf{g}^{[i]}$ fulfils the two desired properties that boosting the gradient of examples with clean labels, and neutralizing the gradient of the examples with false labels.

## C.2 Class-Prior Estimation in Imprecise-Label Datasets

When the class priors $p(y)$ (here we slightly abuse notation and denote them as $\pi_y$) are not directly accessible to the learning algorithm, they can be estimated using off-the-shelf estimation methods (Luo et al., 2024; Wang et al., 2022a). In this section, we present the problem formulation and outline how class priors can be estimated in practice.

### C.2.1 Class-prior estimation in Partial-label datasets

In partial-label learning, each instance is associated with a candidate label set rather than a single ground-truth label. This label ambiguity makes it difficult to estimate the class prior distribution, since simply counting training samples per class is no longer feasible. To address this issue, we adopt an iterative estimation strategy that updates the class prior in a moving-average manner.

We use the model's predicted labels as a proxy for class prior estimation. Since predictions in the early stage of training are often unreliable, we design a moving-average update rule that gradually stabilizes the estimated distribution. The update starts from a uniform prior $\mathbf{r} = [1/c, \ldots, 1/c]$, and is refined at each training epoch as

$$\mathbf{r} \leftarrow \mu \mathbf{r} + (1 - \mu)\mathbf{z}, \qquad \mathbf{z}_j = \frac{1}{n} \sum_{i=1}^{n} \mathbb{I}\left( j = \arg \max_{y \in S_i} f_j(x_i) \right), \tag{27}$$

where $\mu \in [0, 1]$ is a momentum parameter, $S_i$ is the candidate label set for sample $x_i$, and $f_j(x_i)$ denotes the model prediction for class $j$. This rule progressively refines $\mathbf{r}$ as the model improves, leading to more accurate and stable class-prior estimates.

### C.2.2 Class-prior estimation in Supplementary-unlabeled datasets

In the case of supplementary-unlabeled datasets, which also is called semi-supervised datasets, the estimation of class-prior is relatively straightforward. We assume that the distribution of the labeled dataset is consistent with that of the unlabeled dataset. Therefore, the class-prior can be directly obtained by counting the class distribution over the labeled dataset, which serves as a reliable approximation of the overall data distribution.

### C.2.3 Class-Prior Estimation in Noisy-Label Datasets

We consider the widely adopted class-dependent label noise setting (Yao et al., 2020), where the observed noisy label of each $\mathbf{x} \in \mathcal{X}$ depends only on its underlying clean label. Formally, the transition probability from class $i$ to class $j$ is defined as

$$P(\widetilde{Y} = e_j \,|\, Y = e_i, X = \mathbf{x}) = P(\widetilde{Y} = e_j \,|\, Y = e_i) = T_{ij}, \quad \forall i, j \in [[c]],$$

where $\mathbf{T} = [T_{ij}] \in [0, 1]^{c \times c}$ is the noise transition matrix. To make the estimation of $\mathbf{T}$ feasible, we follow prior work and impose the following assumptions.

**Assumption 1** (Sufficiently Scattered Assumption (Li et al., 2021b)). *The clean class posterior $P(Y \mid X) = [P(Y = e_1 \mid X), \ldots, P(Y = e_c \mid X)]^\top \in [0,1]^c$ is said to be sufficiently scattered if there exists a set $\mathcal{H} = \{\mathbf{x}_1, \ldots, \mathbf{x}_m\}$ such that the matrix $\mathbf{H} = [P(Y \mid X = \mathbf{x}_1), \ldots, P(Y \mid X = \mathbf{x}_m)]$ satisfies: (i) $\mathcal{Q} \subseteq \text{cone}\{\mathbf{H}\}$, where $\mathcal{Q} = \{\mathbf{v} \in \mathbb{R}^c \mid \mathbf{v}^\top \mathbf{1} \geq \sqrt{c-1}\|\mathbf{v}\|_2\}$, and $\text{cone}\{\mathbf{H}\}$ denotes the convex cone generated by the columns of $\mathbf{H}$; (ii) $\text{cone}\{\mathbf{H}\} \not\subseteq \text{cone}\{\mathbf{U}\}$ for any unitary matrix $\mathbf{U} \in \mathbb{R}^{c \times c}$ that is not a permutation matrix.*

**Assumption 2** (Nonsingular $\mathbf{T}$). *The transition matrix $\mathbf{T}$ is nonsingular, i.e., $\text{Rank}(\mathbf{T}) = c$.*

Assumption 1 ensures that the clean posteriors are sufficiently scattered so that the ground-truth $\mathbf{T}$ can be identified, while Assumption 2 guarantees the invertibility of $\mathbf{T}$.

Let $\epsilon$ denote the noise rate. For symmetric label noise, we have $T_{ii} = 1 - \epsilon$ and $T_{ij} = \frac{\epsilon}{c-1}$ with $j \neq i$. In practice, the transition matrix can be estimated by solving the following optimization problem (Li et al., 2021b):

$$\min_{\theta, \widehat{\mathbf{T}}} L(\theta, \widehat{\mathbf{T}}) = \frac{1}{n} \sum_{i=1}^{n} \ell\big(\widehat{\mathbf{T}}^\top h_\theta(\mathbf{x}_i), \widetilde{y}_i\big) + \lambda \cdot \log \det(\widehat{\mathbf{T}}), \tag{28}$$

where $\ell$ is a loss function (typically cross-entropy), $h_\theta(\cdot)$ is the output of a neural network parameterized by $\theta$, and the regularization term $\log \det(\widehat{\mathbf{T}})$ encourages the estimated transition matrix to have minimal simplex volume. Here $\lambda > 0$ is a trade-off hyperparameter. By Assumption 1, the solution $\widehat{\mathbf{T}}$ converges to the true $\mathbf{T}$ given sufficient noisy data (Theorem 1 in (Li et al., 2021b)).

Once the transition matrix $\mathbf{T}$ is estimated, the clean class prior $\pi = [\pi_1, \ldots, \pi_c]^\top$ can be obtained by solving the following system of linear equations:

$$\begin{cases} \widetilde{\pi}_1 = T_{11}\pi_1 + T_{21}\pi_2 + \cdots + T_{c1}\pi_c \\ \widetilde{\pi}_2 = T_{12}\pi_1 + T_{22}\pi_2 + \cdots + T_{c2}\pi_c \\ \quad \vdots \\ \widetilde{\pi}_c = T_{1c}\pi_1 + T_{2c}\pi_2 + \cdots + T_{cc}\pi_c \end{cases}, \tag{29}$$

where $\widetilde{\pi}_i = P(\widetilde{Y} = e_i)$ is the noisy class prior of the $i$-th class. The empirical estimate of $\widetilde{\pi}_i$ can be computed as

$$\widehat{\widetilde{\pi}}_i = \frac{1}{n} \sum_{j=1}^{n} \mathbf{1}\{\widetilde{y}_j = e_i\}, \quad \forall i \in [[c]]. \tag{30}$$

Solving this system yields the clean class prior $\pi$, which is then used in subsequent modeling.

# D  IMPLEMENTATION DETAILS

Our implementation is based on PyTorch 1.12 (Paszke et al., 2019), and all experiments were conducted on NVIDIA Tesla A100 GPUs with CUDA 12.4.

**Imprecise-label construction.** For all class-dependent partial-label datasets, we construct a $10 \times 10$ circulant transition matrix $\begin{bmatrix} 1 & q+0.2 & q & q-0.2 & \cdots & q+0.2 & q & q-0.2 \\ q-0.2 & 1 & q+0.2 & q & \cdots & q & q-0.2 & q+0.2 \\ q & q-0.2 & 1 & q+0.2 & \cdots & q-0.2 & q+0.2 & q \\ \vdots & \vdots & \vdots & \vdots & \ddots & \vdots & \vdots & \vdots \\ q+0.2 & q & q-0.2 & 1 & \cdots & q & q-0.2 & 1 \end{bmatrix}$, where each row maps a true label to a candidate set of labels with varying probabilities, and $q$ is set to 0.5. For noisy-label datasets with asymmetric noise (40% flip probability), we adopt the following mappings: *Fashion-MNIST:* 'Pullover'→'Sneaker', 'Dress'→'Bag', 'Sandal'→'Shirt', 'Shirt'→'Sandal'. *CIFAR-10:* 'Truck'→'Automobile', 'Bird'→'Airplane', 'Deer'→'Horse', 'Cat'→'Dog', 'Dog'→'Cat'. *ImageNette:* 'Tench'→'English springer', 'Cassette player'→'Garbage truck', 'Chain saw'→'Church', 'Golf ball'→'Parachute', 'Parachute'→'Golf ball'.

**Model setup.** The overall diffusion framework follows EDM (Karras et al., 2022), and the training hyperparameters are kept consistent with those reported therein. For all experiments, we adopt the DDPM++ network architecture with a U-Net backbone. Specifically, we employ the Adam optimizer

with a learning rate of $1\mathrm{e}-3$, parameters $(\beta_1, \beta_2) = (0.9, 0.999)$, and $\epsilon = 1\mathrm{e}-8$. The EMA decay is set to 0.5. We use a batch size of 128 for Fashion-MNIST, 64 for CIFAR-10, and 16 for ImageNette. For the diffusion classifier, we set the timestep interval length $\Delta$ to 6.4. All models are trained from scratch for 200k iterations.

# E  EXPERIMENTS

## E.1  EVALUATION METRICS

We evaluate the trained CDMs using four unconditional metrics, including Fréchet Inception Distance (FID) (Heusel et al., 2017), Inception Score (IS) (Salimans et al., 2016), Density, and Coverage (Naeem et al., 2020), and three conditional metrics, namely CW-FID, CW-Density, CW-Coverage (Chao et al., 2022). All metrics are computed using the official implementation of DLSM (Chao et al., 2022). Although these metrics have been introduced in related work (Na et al., 2024), we briefly recap them here for completeness and clarity.

**Unconditional metrics.** Unconditional metrics evaluate generated samples without reference to class labels. In our experiments, images are first generated conditionally per class but then pooled without labels when computing the metrics. This evaluation protocol is consistent with prior studies (Kaneko et al., 2019; Chao et al., 2022).

- FID measures the distance between real and generated image distributions in the pre-trained feature space (Szegedy et al., 2016), indicating the fidelity and diversity of generated images.
- IS evaluates whether generated images belong to distinct classes and whether each image is class-consistent, reflecting the realism and class separability of generated images.
- Density and Coverage are reliable versions of Precision and Recall (Naeem et al., 2020), respectively. Density measures how well generated samples cover real data distribution, while Coverage assesses how well real samples are represented by generated ones.

**Conditional metrics.** To measure conditional consistency, we adopt class-wise (CW) variants of the above metrics, which compute each metric separately within each class and then average across classes. Notably, CW-FID (also called intra-FID) is widely used in conditional generative modeling (Miyato & Koyama, 2018; Kaneko et al., 2019), and has been highlighted as a key measure of conditional distribution quality.

**Remark**: It should be noted that the Fashion-MNIST dataset is not suitable for evaluation using these metrics, so we do not perform evaluation on the Fahsion-MNIST dataset.

## E.2  FULL RESULTS IN WEAKLY SUPERVISED LEARNING

Building on the experiments presented in the main text, we further provide an extended comparison with a broader set of methods to ensure a comprehensive evaluation. The details are summarized as

**Partial-label learning.** We compare against ten representative baselines: *PRODEN* (Lv et al., 2020), *CAVL* (Zhang et al., 2021b), *POP* (Xu et al., 2023), *CC* (Feng et al., 2020), *LWS* (Wen et al., 2021), *IDGP* (Qiao et al., 2023), *PiCO* (Wang et al., 2023), *ABLE* (Xia et al., 2022), *CRDPLL* (Wu et al., 2022), and *DIRK* (Wu et al., 2024). For a fair comparison, we follow the hyperparameter settings used in *PLENCH* (Wang et al., 2025b). The complete results are reported in Table 5.

**Semi-supervised learning.** We follow the training and evaluation protocols of *USB* (Wang et al., 2022c), a widely adopted benchmark for fair and unified SSL comparisons. Our baselines cover a broad spectrum of recent approaches. First, we include confidence-thresholding methods such as *FixMatch* (Sohn et al., 2020), *FlexMatch* (Zhang et al., 2021a), *FreeMatch* (Wang et al., 2022d), *ReMixMatch* (Berthelot et al., 2019a), *Dash* (Xu et al., 2021) and *UDA* (Xie et al., 2020). Second, we consider contrastive-learning based and pseudo-label based methods, including *CoMatch* (Li et al., 2021a), *SoftMatch* (Chen et al., 2023) and *SimMatch* (Zheng et al., 2022). Finally, we add several classical and widely studied SSL approaches, including *Pseudo-Labeling* (Lee et al., 2013), *VAT* (Miyato et al., 2018) and *Mean Teacher* (Tarvainen & Valpola, 2017). This diverse collection of baselines allows us to rigorously examine whether our framework remains competitive against both state-of-the-art and classical SSL methods under consistent experimental setups.

Table 5: Classification results on Fashion-MNIST, CIFAR-10, and ImageNette datasets under various types of partial-label supervision. **Bold** numbers indicate the best performance.

| Method | Fashion-MNIST | | CIFAR-10 | | ImageNette | |
|---|---|---|---|---|---|---|
| | Random | Class-50% | Random | Class-50% | Random | Class-50% |
| PRODEN | 93.31±0.07 | 93.44±0.21 | 90.02±0.22 | 90.44±0.44 | 84.75±0.13 | 83.50±0.60 |
| CAVL | 93.09±0.17 | 92.67±0.25 | 87.28±0.64 | 87.16±0.58 | 41.69±4.12 | 46.46±7.15 |
| POP | 93.59±0.17 | 93.57±0.19 | 89.13±0.22 | 90.19±0.10 | 84.65±0.55 | 84.29±0.17 |
| CC | 93.17±0.32 | 92.65±0.29 | 88.40±0.24 | 89.12±0.23 | 81.11±0.50 | 80.74±0.68 |
| IDGP | 92.26±1.25 | 93.07±0.16 | 89.65±0.53 | 90.83±0.34 | 84.07±0.26 | 82.18±0.13 |
| PiCO | 93.32±0.12 | 93.32±0.33 | 86.40±0.89 | 87.51±0.66 | 82.15±0.23 | 84.41±0.93 |
| ABLE | 93.02±0.26 | 93.20±0.16 | 90.77±0.33 | 90.74±0.48 | 71.81±2.46 | 76.53±1.28 |
| CRDPLL | 94.03±0.14 | 93.80±0.23 | 92.74±0.26 | 92.89±0.27 | 84.31±0.25 | 88.08±0.34 |
| DIRK | 94.11±0.22 | 93.99±0.24 | 93.48±0.14 | 93.22±0.37 | 87.90±0.11 | 87.47±0.17 |
| Vanilla | 80.20±1.29 | 66.03±1.43 | 60.25±0.17 | 56.34±0.50 | 56.04±0.61 | 59.47±0.51 |
| DMIS$^{CE}$ | 84.24±0.37 | 78.45±0.46 | 91.47±0.15 | 90.52±0.35 | 84.49±0.05 | 82.34±0.27 |
| DMIS | **94.27±0.55** | **94.20±0.15** | **94.70±0.49** | **93.53±0.12** | **89.31±0.21** | **88.42±0.43** |

Table 6: Classification results on Fashion-MNIST, CIFAR-10, and ImageNette datasets under various types of supplementary-unlabeled supervision. **Bold** numbers indicate the best performance.

| Method | Fashion-MNIST | | CIFAR-10 | | ImageNette | |
|---|---|---|---|---|---|---|
| | Random-1% | Random-10% | Random-1% | Random-10% | Random-1% | Random-10% |
| Pseudo-Labeling | 83.53±0.46 | 89.59±0.23 | 50.10±0.95 | 72.92±0.17 | 43.00±0.82 | 68.03±0.32 |
| Mean Teacher | 82.34±0.09 | 89.91±0.15 | 47.69±0.27 | 73.01±0.78 | 40.53±1.56 | 65.72±0.55 |
| VAT | 83.31±0.61 | 89.35±0.12 | 49.64±0.90 | 71.07±1.27 | 38.63±8.39 | 63.93±5.18 |
| UDA | 84.28±0.41 | 90.83±0.34 | 69.20±1.41 | 80.50±0.55 | 50.52±3.79 | 72.53±1.17 |
| FixMatch | 84.32±0.33 | 90.76±0.38 | 67.48±1.42 | 80.00±0.63 | 50.41±4.43 | 71.32±1.93 |
| Dash | 84.73±0.09 | 91.16±0.20 | 70.14±0.69 | 81.50±0.68 | 57.68±2.19 | 74.66±0.81 |
| CoMatch | 85.31±0.29 | 90.52±0.12 | 61.45±1.46 | 77.79±0.53 | 63.88±0.78 | 73.20±0.46 |
| FlexMatch | 84.43±0.30 | 90.69±0.03 | 70.72±0.93 | 81.35±0.48 | 61.39±0.70 | 73.08±0.13 |
| FreeMatch | 84.30±0.37 | 90.92±0.24 | 70.15±0.44 | 80.99±0.56 | 60.37±1.11 | 73.14±1.03 |
| SimMatch | 84.69±0.17 | 91.18±0.13 | 73.33±1.02 | 82.90±0.43 | 58.12±2.66 | 76.12±0.45 |
| SoftMatch | 84.72±0.23 | 91.22±0.11 | 73.24±0.82 | 88.66±0.60 | 58.50±2.31 | 75.75±0.25 |
| Vanilla | 78.37±3.72 | 90.50±1.00 | 53.49±0.15 | 85.13±0.12 | 49.55±0.99 | 74.70±0.53 |
| DMIS$^{CE}$ | 82.92±0.17 | 91.07±0.18 | 75.40±0.54 | 89.85±0.08 | 62.64±0.24 | 71.39±0.45 |
| DMIS | **85.92±0.13** | **92.97±0.21** | **76.30±0.17** | **92.47±0.39** | **68.23±0.19** | **77.30±0.15** |

Table 7: Classification results on Fashion-MNIST, CIFAR-10, and ImageNette datasets under various types of noisy-label supervision. **Bold** numbers indicate the best performance.

| Method | Fashion-MNIST | | CIFAR-10 | | ImageNette | |
|---|---|---|---|---|---|---|
| | Sym-40% | Asym-40% | Sym-40% | Asym-40% | Sym-40% | Asym-40% |
| CE | 76.18±0.26 | 82.01±0.06 | 67.22±0.26 | 76.98±0.42 | 58.43±0.77 | 71.81±0.38 |
| Co-learning | 90.85±0.63 | 84.10±2.01 | 84.97±0.53 | 80.36±1.09 | 76.16±0.96 | 75.37±0.49 |
| Co-teaching | 92.17±0.34 | 92.78±0.25 | 86.54±0.57 | 79.38±0.39 | 66.55±1.00 | 75.12±0.50 |
| Co-teaching+ | 91.05±0.06 | 91.62±0.20 | 67.28±1.85 | 79.43±0.47 | 75.79±0.79 | 75.17±0.40 |
| SCE | 93.62±0.22 | 88.60±0.20 | 82.82±0.40 | 81.54±0.64 | 77.99±0.39 | 74.81±1.04 |
| GCE | 93.64±0.03 | 87.48±0.09 | 85.00±0.27 | 77.97±3.69 | 81.18±0.35 | 72.61±1.14 |
| Decoupling | 92.24±0.23 | 92.10±0.44 | 82.24±0.28 | 79.89±0.58 | 75.53±0.69 | 78.24±0.21 |
| ELR | 93.13±0.13 | 92.82±0.09 | 85.68±0.13 | 81.32±0.31 | 84.03±2.86 | 73.51±0.31 |
| JoCoR | 84.05±1.11 | 89.45±4.43 | 77.92±3.92 | 78.68±0.07 | 67.82±1.97 | 74.67±0.43 |
| Mixup | 92.21±0.03 | 92.01±1.02 | 84.26±0.64 | 83.21±0.85 | 76.65±1.62 | 77.16±0.71 |
| PENCIL | 90.85±0.58 | 91.77±0.69 | 85.91±0.26 | 84.89±1.49 | 81.94±1.26 | 77.20±1.15 |
| Vanilla | 90.11±1.24 | 85.41±0.96 | 80.22±0.10 | 86.31±0.10 | 55.86±1.95 | 53.91±1.07 |
| DMIS$^{CE}$ | 82.76±0.57 | 83.39±0.24 | 84.75±0.36 | 84.21±0.18 | 80.47±0.56 | 77.21±0.19 |
| DMIS | **93.40±0.40** | **93.20±0.30** | **88.63±0.12** | **88.83±0.33** | **84.12±0.18** | **79.30±0.27** |

**Noisy-label learning.** We further benchmark our method against nine widely used approaches: *Coteaching* (Han et al., 2018), *Coteaching+* (Yu et al., 2019), *SCE* (Wang et al., 2019), *GCE* (Zhang & Sabuncu, 2018), *Decoupling* (Malach & Shalev-Shwartz, 2017), *ELR* (Liu et al., 2020), and *JoCoR* (Wei et al., 2020). These methods cover a range of strategies, from sample selection and reweighting to robust loss design, thus providing a diverse and rigorous benchmark.

Across all three weakly supervised scenarios, our method consistently achieves the best performance compared to existing baselines, reinforcing both its robustness and versatility under different forms of imprecise supervision.

### E.3 INTEGRATION WITH EXISTING IMPRECISE-LABEL CORRECTORS

Existing weakly supervised learning methods often rely on pseudo-labeling strategies that aim to correct imprecise labels by assigning refined labels to training samples. From this perspective, our approach is orthogonal to such methods: while pseudo-labeling seeks to approximate the true labels as closely as possible, our framework focuses on robustly learning from the remaining label noise. In practice, pseudo-labeling methods inevitably produce imperfect corrections. While most samples may be relabeled correctly, a non-negligible portion of instances still receive erroneous pseudo-labels because no classifier is perfect. As a result, imprecise supervision is effectively transformed into a noisy-label supervision.

This naturally complements our framework: by combining a pseudo-label corrector with *DMIS*, one can first reduce label uncertainty through correction and then leverage the robustness of diffusion models to learn from the residual noise. To validate this premise, we conduct a case study where a noisy-label learning method trained on CIFAR-10 with 40% symmetric noise achieves a pseudo-label accuracy of 80% on the training set. Using this pseudo-labeled dataset as input, our *DMIS* framework further improves the classification performance. As illustrated in Figure 6, integrating pseudo-label correction with *DMIS* consistently improves the performance across all datasets. Thus, we believe that our framework addresses the challenge of imprecise labels through the lens of diffusion model, offering a complementary perspective to conventional noisy-label approaches.

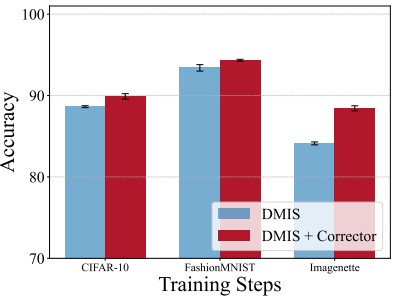
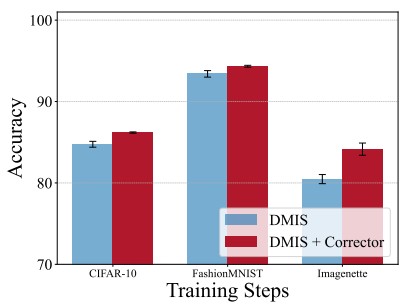

(a) Diffusion classifier results.     (b) Regenerate-classification results.

Figure 6: Test accuracy before and after applying pseudo-label correction with *DMIS*.

### E.4 COMPARISON OF ACCURACY CURVES BETWEEN *DMIS* AND *Vanilla*

To better illustrate the difference between the *Vanilla* method and our proposed *DMIS*, we plot the test accuracy curves during training, as shown in Figure 7. Across all settings, the *Vanilla* model exhibits an initial rise in accuracy followed by a gradual decline as training progresses, suggesting that it struggles to maintain stable performance under prolonged training. In contrast, *DMIS* consistently sustains high accuracy throughout training, showing its robustness across diverse supervision types.

Specifically, in the noisy-label setting, the decline of *Vanilla* is especially pronounced, reflecting its sensitivity to label corruption. In partial-label learning, *Vanilla* also exhibits instability, whereas *DMIS* maintains reliable performance. Even in semi-supervised learning, where labels are clean but scarce, *DMIS* achieves higher and more stable accuracy compared to *Vanilla*, demonstrating that our framework is not only noise-robust but also effective in leveraging limited supervision.

### E.5 VISUALIZATION OF NOISY CONDENSED DATASETS

We visualize the condensed images on CIFAR-10 and Fashion-MNIST in Figure 8 and Figure 9, respectively. It is evident that datasets generated by our method exhibit both higher diversity and stronger realism compared to other approaches. In particular, for the condensed Fashion-MNIST images, methods such as *DC* and *DM* often produce samples that do not faithfully correspond to their assigned class, resulting in condensed datasets that still contain noisy labels and thus degrade performance. By contrast, our proposed *DMIS* generates class-consistent and visually recognizable samples across categories, yielding condensed datasets that better preserve label fidelity and semantic

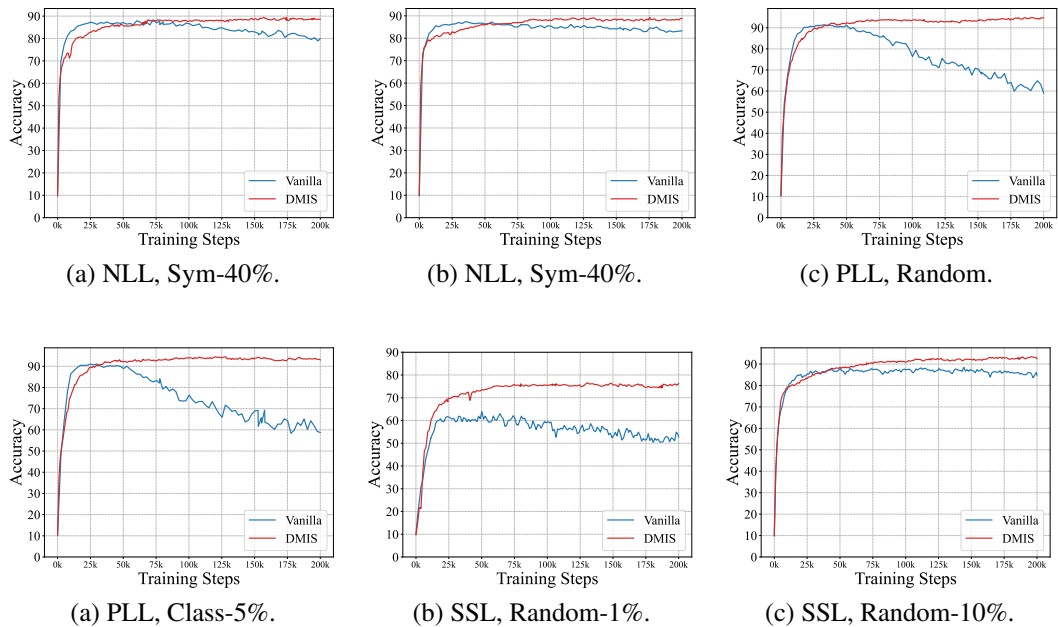

(a) NLL, Sym-40%.     (b) NLL, Sym-40%.     (c) PLL, Random.

(a) PLL, Class-5%.     (b) SSL, Random-1%.     (c) SSL, Random-10%.

Figure 7: Test accuracy curves of the *Vanilla* and *DMIS* models on CIFAR-10 under different forms of imprecise supervision, including noisy-label learning (NLL), partial-label learning (PLL), and semi-supervised learning (SSL).

alignment. These visualizations further support the quantitative results, highlighting the advantage of generative condensation under noisy supervision.

### E.6 ADDITIONAL RESULTS ON DATASET CONDENSATION UNDER DIFFERENT FORMS OF IMPRECISE SUPERVISION

To illustrate the extreme case of noisy dataset condensation, we report the results when the IPC is set to 1. As shown in Table 8, *DMIS* consistently achieves the best performance across all datasets and noise types, even under the extreme case of IPC = 1. Notably, while most existing condensation methods collapse under severe supervision noise, our method maintains a clear advantage, outperforming the strongest baselines by a large margin. These results further demonstrate the robustness of *DMIS* in distilling informative representations despite highly limited and imprecisely labeled data.

Table 8: Classification results (test accuracy, %) on noisy-label Fashion-MNIST, CIFAR-10, and ImageNette datasets. 'IPC' indicates the number of images per class in the condensed dataset. **Bold** numbers indicate the best performance.

| Dataset | Type | IPC | DC | DSA | DM | MTT | RDED | SRE2L | DMIS |
|---------|------|-----|------|------|------|------|------|-------|------|
| F-MNIST | Sym-40% | 1 | 15.21±0.75 | 19.55±0.58 | 15.56±0.20 | 10.86±1.90 | 18.07±3.33 | 14.33±1.20 | **33.18±2.15** |
| | Asym-40% | 1 | 20.17±0.29 | 17.61±0.89 | 23.91±0.36 | 7.39±0.84 | 13.20±0.83 | 13.13±0.21 | **25.78±0.70** |
| CIFAR-10 | Sym-40% | 1 | 8.99±1.59 | 10.00±0.00 | **14.41±1.03** | 9.99±0.00 | 11.20±0.41 | 11.06±0.83 | 11.81±1.04 |
| | Asym-40% | 1 | 11.88±1.55 | 10.00±0.00 | 10.00±0.00 | 9.96±0.05 | 13.96±1.38 | 15.49±0.34 | **15.88±0.57** |
| ImageNette | Sym-40% | 1 | 9.87±0.00 | 9.87±0.00 | 9.87±0.00 | 19.17±2.35 | 12.98±1.16 | 11.90±0.78 | **19.32±0.84** |
| | Asym-40% | 1 | 9.87±0.00 | 9.87±0.00 | 9.87±0.00 | 17.36±0.10 | 12.98±0.42 | 18.55±2.19 | **21.13±0.95** |

Furthermore, even when samples are imprecisely annotated with candidate labels, our method is still able to perform effective condensation on partial-label datasets. In contrast, most existing dataset condensation methods rely on the assumption of having single labels for each instance and therefore fail under this type of supervision. The only exception lies in decoupled condensation approaches such as *RDED* and *SRE2L*, where a teacher model can still be trained on partial-label data. We present the corresponding results under partial-label supervision below in Table 9.

As shown in the table, our method consistently achieves substantial improvements across both Random and Class-50% candidate set generation strategies, and under all IPC settings. In particular, on Fashion-MNIST, our approach yields dramatic performance gains, reaching above **87%** accuracy even with partial label supervision, whereas both *RDED* and *SRE2L* fail to exceed 16% under the same setting. On the more challenging CIFAR-10 benchmark, our method also demonstrates strong robustness, especially under larger IPCs where the gap over baseline methods becomes increasingly pronounced (e.g., over **20%** absolute improvement at IPC = 100). These results highlight that our condensation strategy can effectively leverage weak supervision and generate compact yet highly informative synthetic datasets, even when label noise is introduced by the partial-label scenario.

Table 9: Classification results (test accuracy, %) on partial-label Fashion-MNIST and CIFAR-10 datasets under different IPCs. 'Random' and 'Class-50%' denote two candidate set generation strategies. **Bold** numbers indicate the best performance.

| Dataset | Random | | | | Class-50% | | | |
|---|---|---|---|---|---|---|---|---|
| | IPC | RDED | SRe2L | Ours | IPC | RDED | SRe2L | Ours |
| F-MNIST | 1 | 10.48±0.82 | 9.72±1.02 | **44.06±1.64** | 1 | 10.73±0.78 | 10.93±0.76 | **33.99±2.48** |
| | 10 | 13.17±4.66 | 8.80±0.70 | **72.02±0.77** | 10 | 14.58±1.19 | 10.83±0.35 | **70.46±0.26** |
| | 50 | 13.06±2.36 | 10.30±0.40 | **83.98±0.12** | 50 | 15.55±0.55 | 11.33±0.76 | **79.67±0.13** |
| | 100 | 13.39±2.40 | 9.34±1.77 | **87.30±0.31** | 100 | 11.90±2.67 | 11.05±0.77 | **81.42±0.25** |
| CIFAR-10 | 1 | 15.32±1.72 | **20.69±0.88** | 16.31±1.54 | 1 | 14.52±1.06 | 15.55±0.91 | **16.61±1.14** |
| | 10 | 26.28±0.29 | 19.45±1.12 | **30.50±0.27** | 10 | 10.00±0.00 | 18.56±0.05 | **25.00±0.70** |
| | 50 | 34.96±0.92 | 20.56±0.71 | **44.39±0.65** | 50 | 25.59±1.32 | 19.39±1.09 | **45.94±1.27** |
| | 100 | 28.88±2.56 | 19.65±1.09 | **58.46±0.64** | 100 | 25.81±1.64 | 18.65±1.51 | **58.06±0.32** |

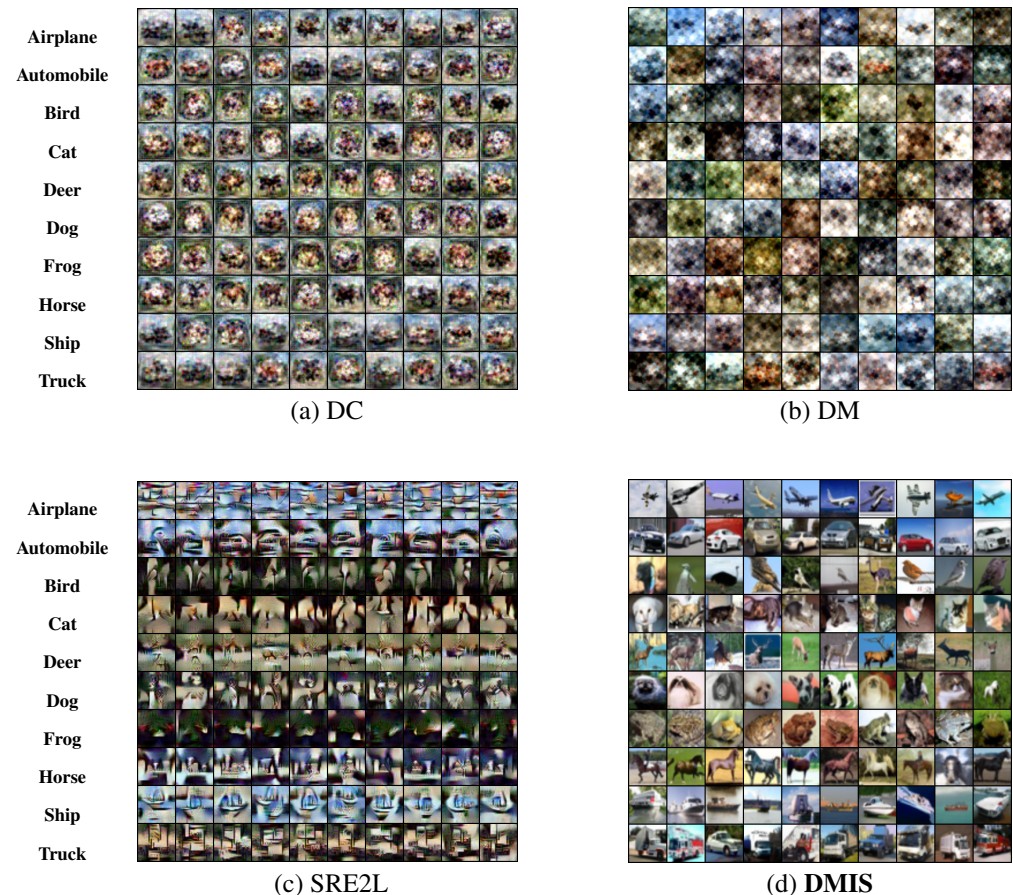

(a) DC          (b) DM

(c) SRE2L          (d) **DMIS**

Figure 8: Visualization of condensed CIFAR-10 images generated by *DC*, *DM*, *SRE2L*, and our method **DMIS**.

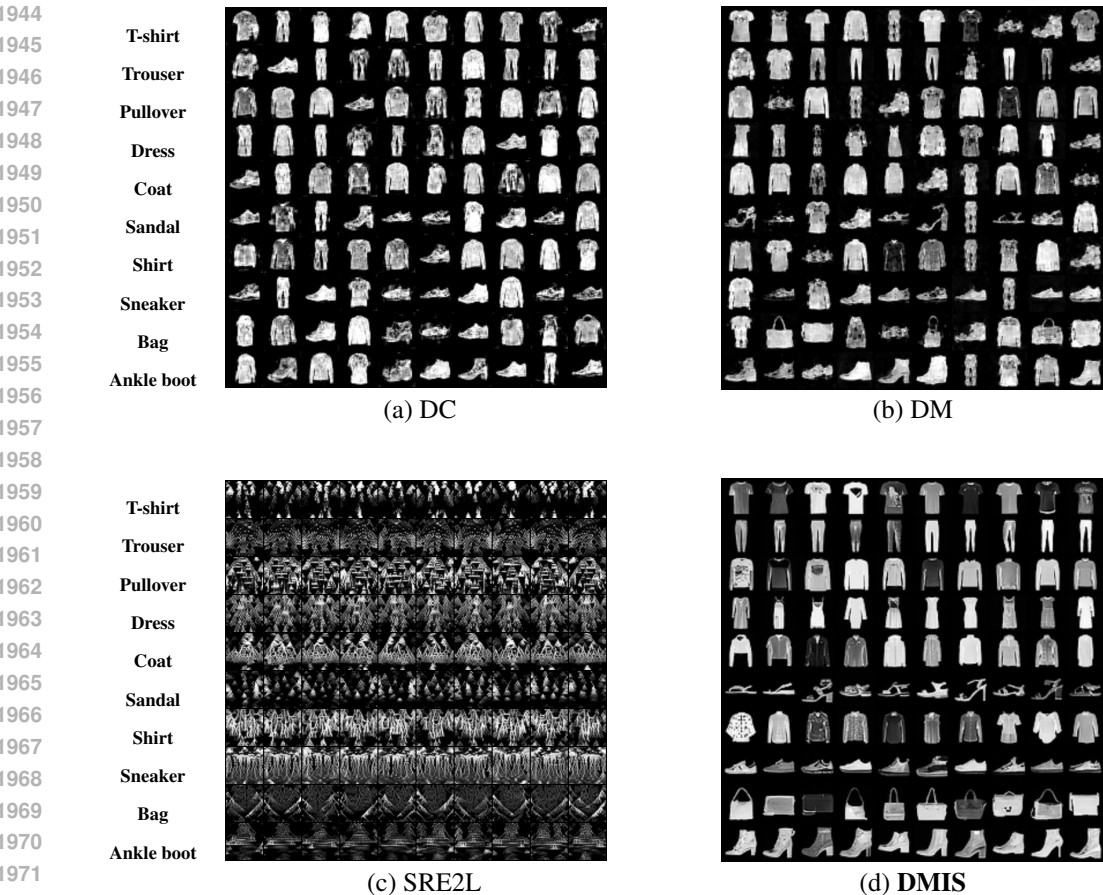

Figure 9: Visualization of condensed Fashion-MNIST images generated by *DC*, *DM*, *SRE2L*, and our method ***DMIS***.

## F ADDITIONAL EXPERIMENTS RESULTS

### F.1 THE FULL RSULTS OF TABLE 1.

Table 10: Generative results on CIFAR-10 and ImageNette under various settings. 'uncond' and 'cond' indicate unconditional and conditional metrics. **Bold** numbers indicate better performance.

| | Metric | | Noisy-label supervision | | | | Partial-label supervision | | | | Suppl-unlabeled supervision | | | | Clean |
|---|---|---|---|---|---|---|---|---|---|---|---|---|---|---|---|
| | | | Sym-40% | | Asym-40% | | Random | | Class-50% | | Random-1% | | Random-10% | | |
| | | | Vanilla | DMIS | Vanilla | DMIS | Vanilla | DMIS | Vanilla | DMIS | Vanilla | DMIS | Vanilla | DMIS | Vanilla |
| CIFAR-10 uncond | FID | (↓) | **3.33**±0.06 | 3.47±0.11 | 3.23±0.07 | **3.10**±0.11 | 7.76±0.25 | **2.26**±0.08 | 11.75±0.42 | **2.77**±0.09 | 3.16±0.07 | **3.12**±0.05 | 2.93±0.11 | **2.89**±0.08 | 2.05±0.05 |
| | IS | (↑) | 9.56±0.08 | **9.68**±0.05 | 9.02±0.12 | **9.73**±0.06 | 9.09±0.15 | **9.80**±0.04 | 9.62±0.11 | **9.68**±0.07 | 10.03±0.09 | **10.57**±0.06 | 9.80±0.08 | **9.83**±0.05 | 10.61±0.04 |
| | Density | (↑) | 101.39±0.85 | **109.75**±0.62 | 100.06±0.91 | **109.69**±0.55 | 103.21±1.12 | **106.49**±0.48 | 108.76±0.75 | **109.06**±0.52 | 97.19±1.05 | **108.18**±0.66 | 99.96±0.88 | **108.87**±0.59 | 112.59±0.45 |
| | Coverage | (↑) | 81.12±0.35 | **81.21**±0.28 | 80.71±0.41 | **81.30**±0.32 | 68.45±0.65 | **82.69**±0.25 | 64.90±0.72 | **81.52**±0.30 | 78.44±0.45 | **81.00**±0.29 | 81.85±0.38 | **82.00**±0.26 | 83.27±0.22 |
| CIFAR-10 cond | CW-FID | (↓) | 29.84±1.15 | **13.85**±0.45 | 14.70±0.52 | **13.24**±0.38 | 27.18±0.95 | **10.65**±0.35 | 32.44±1.22 | **11.56**±0.41 | 16.25±0.65 | **16.12**±0.55 | 11.84±0.48 | **11.77**±0.42 | 9.83±0.35 |
| | CW-Density | (↑) | 72.98±0.82 | **107.23**±0.65 | 90.85±0.75 | **107.07**±0.58 | 102.04±0.92 | **105.75**±0.61 | 102.43±0.88 | **108.66**±0.55 | 89.99±0.78 | **100.73**±0.68 | 96.29±0.72 | **107.94**±0.62 | 111.70±0.52 |
| | CW-Coverage | (↑) | 73.39±0.45 | **80.11**±0.35 | 79.63±0.42 | **79.65**±0.38 | 65.45±0.68 | **82.09**±0.32 | 61.45±0.75 | **81.24**±0.36 | 75.03±0.52 | **76.84**±0.41 | 80.80±0.45 | **81.12**±0.39 | 83.91±0.30 |
| ImageNette uncond | FID | (↓) | 14.11±0.55 | **13.44**±0.48 | 13.93±0.52 | **13.91**±0.45 | 79.13±2.15 | **72.62**±1.85 | 91.28±2.45 | **79.12**±1.95 | 23.88±0.85 | **19.26**±0.72 | 14.32±0.58 | **12.84**±0.46 | 11.52±0.42 |
| | IS | (↑) | 12.69±0.15 | **13.21**±0.12 | 12.51±0.14 | **13.73**±0.11 | 9.19±0.25 | **9.40**±0.18 | 9.27±0.22 | **9.11**±0.24 | 12.23±0.16 | **13.72**±0.13 | 12.80±0.15 | **13.16**±0.12 | 13.81±0.10 |
| | Density | (↑) | 76.62±0.42 | **76.81**±0.38 | 78.32±0.45 | **79.81**±0.35 | 95.33±1.25 | **99.83**±0.75 | 94.29±1.32 | **102.58**±0.85 | 115.94±1.15 | **125.68**±0.95 | 105.27±0.85 | **109.23**±0.78 | 117.23±0.65 |
| ImageNette cond | CW-FID | (↓) | 80.31±2.55 | **60.12**±1.85 | 62.26±1.95 | **58.20**±1.65 | 157.76±3.55 | **63.58**±2.15 | 163.45±3.85 | **67.92**±2.25 | 71.66±2.35 | **70.27**±2.10 | 49.22±1.55 | **44.31**±1.45 | 40.20±1.25 |
| | CW-Density | (↑) | 73.99±0.85 | **81.12**±0.65 | 93.53±0.95 | **94.58**±0.72 | 93.38±0.98 | **95.83**±0.68 | 91.50±1.05 | **95.21**±0.75 | 115.90±1.15 | **118.69**±0.85 | 103.41±0.92 | **115.67**±0.78 | 120.35±0.65 |
| | CW-Coverage | (↑) | 67.89±0.55 | **71.94**±0.45 | 74.18±0.52 | **75.82**±0.48 | 19.76±0.85 | **24.35**±0.55 | 15.88±0.92 | **18.93**±0.62 | 51.73±0.75 | **52.15**±0.65 | 72.61±0.58 | **74.85**±0.52 | 78.48±0.45 |

### F.2 COMPARISON AGAINST OTHER NOISE-ROBUST DIFFUSION METHODS.

We have also shown comparisons against noise-robust diffusion methods (Na et al., 2024; Dufour et al., 2024) that are designed to handle noisy-label data. For image generation task, we compare them and our DMIS model on CIFAR-10 with 40% symmetric and asymmetric label noise, reporting FID, IS under the same architecture and training budget. For noisy-label learning task, we compare them and our DMIS as data generators for downstream classification. We use each model to synthesize the same number of labeled samples and then train a Wide-ResNet-40-10 classifier on top of these

| Metric | Sym-40% | | | Asym-40% | | |
|---|---|---|---|---|---|---|
| | FID | IS | Accuracy | FID | IS | Accuracy |
| DMIS (Ours) | **3.47** | **9.80** | **88.63** | **3.10** | 9.73 | **88.83** |
| CAD (Dufour et al., 2024) | 4.10 | 9.68 | 81.75 | 3.87 | 9.16 | 82.33 |
| TDSM (Na et al., 2024) | 3.85 | 9.40 | 66.40 | 3.96 | **10.12** | 72.32 |

Table 11: Results under 40% symmetric and asymmetric noise.

synthetic datasets. Importantly, both two compared methods assume access to additional prior information, which can give them an advantage in this setting. Despite this, our method still achieves the best overall performance under the same backbone and training budget. This suggests that our approach is competitive while relying on strictly weaker assumptions about the available supervision.

### F.3    TOP-$k$ TRUNCATION FOR LARGE LABEL SPACES

When extending to datasets with many classes, a straightforward implementation becomes expensive because it requires estimating and backpropagating a per-class objective at every step. To reduce this cost in practice, we restrict gradients to classes that carry non-negligible probability mass.

Concretely, we apply a top-$k$ strategy to both the diffusion posterior and the pseudo-label distribution: only the $k$ largest entries are retained, while all remaining entries are zero-masked and do not contribute to the gradient. In this way, the effective complexity scales with the number of active classes $k$ per example, rather than with the total number of classes.

To assess the impact of this approximation, we conduct an experiment on the Caltech-15 dataset (Pan et al., 2023) with $40\%$ symmetric label noise and set $k = 10$. As shown below, the top-$k$ variant achieves performance comparable to the full model while reducing computational cost, indicating that this strategy is a practical mechanism for scaling our method to larger label spaces.

| | Generation Metric | | | | Classification Metric |
|---|---|---|---|---|---|
| | FID | IS | Density | Coverage | Test accuracy (%) |
| DMIS (Ours) | 4.25 | 12.39 | 103.83 | 96.20 | 78.92 |

Table 12: Performance of DMIS under Caltech-15 dataset with 40% symmetric noise.

### F.4    EXPERIMENTS BEYOND SYNTHETIC CLASS-CONDITIONAL NOISE

We primarily use synthetic noisy labels to obtain a controlled setting that supports our theory, where both the noise rate and the noise type (e.g., symmetric, class-dependent) can be precisely specified.

Starting from such controlled synthetic-noise regimes is a necessary first step to validate both the theoretical predictions and the basic empirical behavior of our method. To further demonstrate its practicality under more realistic supervision, we also evaluate DMIS on real noisy-label and partial-label benchmarks. Specifically, we report results on the real noisy-label dataset CIFAR-10N (Wei et al., 2022) and the real partial-label dataset PLCIFAR-10 (Wang et al., 2025b), whose labels are provided by human annotators.

In addition, we consider instance-dependent label noise on CIFAR-10, following standard instance-dependent noise protocols in the noisy-label literature. The results below show that DMIS maintains strong generative quality and competitive classification accuracy under these more complex and realistic noise conditions.

| Dataset | FID | IS | Test accuracy (%) |
|---|---|---|---|
| CIFAR10-N | 3.22 | 9.66 | 93.21 |
| Instance-dependent CIFAR10 | 4.85 | 9.21 | 81.32 |
| PLCIFAR10 | 2.95 | 9.82 | 93.65 |

Table 13: Performance of DMIS under more complex imprecise supervision datasets.

