# OpenReview forum: "Learning Robust Diffusion Models from Imprecise Supervision"
_ICLR.cc/2026/Conference — Submitted to ICLR 2026_

### Official Review · Reviewer_9nKU · 2025-10-30

**Soundness:** 2
**Presentation:** 2
**Contribution:** 2
**Rating:** 4
**Confidence:** 4

**Summary:**

The paper introduces DMIS (Diffusion Models from Imprecise Supervision) — a unified framework designed to train conditional diffusion models (CDMs) robustly when the supervision data is noisy, ambiguous, or incomplete.

In real-world datasets, conditional inputs such as labels or text prompts often contain imprecise information (e.g., mislabeled, missing, or uncertain data). This leads to condition mismatch and reduces generation quality. To address this, DMIS formulates the training process as a likelihood maximization problem, treating the true label as a latent variable and decomposing the objective into:

- a generative component, which models the distribution of imprecise labels, and

- a classification component, which uses a diffusion-based classifier to estimate posterior class probabilities efficiently.

To further improve computational efficiency, the method introduces an optimized timestep sampling strategy, allowing accurate ELBO estimation without redundant computation.

**Strengths:**

- The paper effectively uses a diffusion model to handle generative, discriminative, and condensation tasks under various weak-supervision settings, and presents the framework with clear and consistent notation.

**Weaknesses:**

- **Originality**: The paper covers many tasks, but most of them have already been addressed using diffusion models before. Is there real value in simply combining them into one unified paper? Moreover, Theorem 1 appears to be identical to Theorem 1 in [1], and Proposition 1 seems equivalent to Theorem 3 in [1].

- **No baseline**: There is no baseline shown in Table 1. Why didn’t you compare your method with [1]? Is your method identical to [1]?

- **Limited experimental scope**: Recently, zero-shot tasks using text-conditional diffusion models have shown greater potential and received more attention than those based on class-conditional models. It would be more important to address such tasks using text-to-image (T2I) diffusion models.

[1] (ICLR 24) Label-Noise Robust Diffusion Model.

**Questions:**

What is the distinction between your Theorem 1 and Proposition 1 compared to [1]?

---

> ### Author Response · Authors · 2025-11-27
> **Response to Reviewer 9nKU (W1)**
>
> First of all, we greatly appreciate your time and efforts in reviewing our paper and thank the reviewer for the constructive comments. Below are our answers to each weakness (W) and question (Q) point by point.
>
> > **W1 & Q1: Originality: The paper covers many tasks, but most of them have already been addressed using diffusion models before. Is there real value in simply combining them into one unified paper? Moreover, Theorem 1 appears to be identical to Theorem 1 in [1], and Proposition 1 seems equivalent to Theorem 3 in [1].**
>
> 1. **[Why a unified framework is non-trivial and useful]**
>
> We respectfully clarify that our goal is not to simply “bundle” several known tasks, but to show that a *single probabilistic formulation* can explain and handle a broad range of imprecise supervision regimes in a coherent way. Concretely, we model the joint $p_\theta(X,Z)$, treat the clean label $Y$ as latent, and derive **one likelihood-based objective** that simultaneously yields a generative term for training the CDM, and a classification term for estimating $p(Y\mid X,Z)$. This is different from prior work, where each supervision type typically requires a separate objective and algorithm.
>
> Importantly, we **do not assume access to a noise transition matrix or a time-dependent noisy-label classifier**. In settings where such priors are unavailable or unreliable (which is common in practice), existing methods such as [1] cannot be applied directly, whereas our framework still provides a well-defined objective and training procedure. We view this unified, prior-free treatment as the main conceptual contribution, beyond any individual task.
>
>
>
> 2. **[Differences between our Theorem 1 / Proposition 1 and [1] ]**
>
> We acknowledge that our work is inspired by [1], but our work addresses a more general problem and differs in both assumptions and usage of the results.
>
> In [1], Na et al. rewrite the **noisy-label conditional score** $\nabla_{x_t}\log p_t(x_t\mid \tilde{Y}=\tilde{y})$ as a convex combination of clean-label scores with coefficient
> $$
> w(x_t,\tilde{y},y,t)
> := p(Y=y\mid \tilde{Y}=\tilde{y})\,
>    \frac{p_t(x_t\mid Y=y)}{p_t(x_t\mid \tilde{Y}=\tilde{y})}.
> $$
> These weights are explicitly expressed in terms of a **known noise transition matrix** $p(Y\mid \tilde{Y})$ and a **time-dependent noisy-label classifier** $p_t(x_t\mid Y)$. This decomposition is tailored to their setting, where both the label-noise model and the auxiliary classifier are assumed as priors.
>
> In contrast, our Theorem 1 shows that the **imprecise-label conditional score** $\nabla_{x_t}\log q_t(x_t\mid Z=z)$ can *always* be written as a convex combination of clean-label scores with weights given by the **posterior probability**
> $$
> p(Y=y\mid x_t,z),
> $$
> which is estimated *by the diffusion classifier itself*. We do not require any noise transition matrix or external noise classifier. This posterior-weighted mixture is then used in our training objective (Proposition 1) specifically to *avoid* relying on such priors and to cover partial labels and supplementary-unlabeled data, where a transition matrix is not even defined.
>
> [1] "Label-Noise Robust Diffusion Models." *The Twelfth International Conference on Learning Representations*. 2024.

---

> ### Author Response · Authors · 2025-11-27
> **Response to Reviewer 9nKU (W2)**
>
> > **W2: No baseline: There is no baseline shown in Table 1. Why didn’t you compare your method with [1]? Is your method identical to [1]?**
>
> We appreciate the reviewer for drawing our attention to [1]. Our initial version did not include these methods in Table 1 because there was **no existing diffusion-based method that operates under the same assumptions as ours** across all three imprecise-supervision regimes. For example,
>
> ​	(i) The method [1] assumes access to a **known noise transition matrix** and a **time-dependent noisy-label classifier**, and
>
> ​	(ii) The method [2] requires **additional coherence information** beyond the noisy labels.
>
> By contrast, DMIS purposely **does not rely on any such priors** and is designed to work when only imprecise supervision $(X,Z)$ is available.
>
>
>
> 1. **[New comparison with noise-robust diffusion baselines]**
>
> To strengthen the empirical evidence, we have now added comparisons against **noise-robust diffusion methods** [1,2] in the noisy-label setting, where their assumptions can be instantiated using the synthetic noise model. Below we report CIFAR-10 results under 40% symmetric and 40% asymmetric label noise (same backbone and training budget for all methods).
>
>
>
> Despite [1,2] having access to *stronger prior information* (known noise matrix or extra coherence signals), our method achieves the best overall performance, especially in terms of classification accuracy and FID. This indicates that DMIS is competitive while relying on **strictly weaker assumptions** about the available supervision. We have incorporated this comparison results into Appendix F.2 in the revised manuscript..
>
> |             |          | Sym-40%  |           | \|   |          | Asym-40%  |           |
> | ----------- | -------- | -------- | --------- | ---- | -------- | --------- | --------- |
> | Metric      | FID      | IS       | Accuracy  | \|   | FID      | IS        | Accuracy  |
> | DMIS (Ours) | **3.47** | **9.80** | **88.63** | \|   | **3.10** | 9.73      | **88.83** |
> | TDSM [4]    | 3.85     | 9.40     | 66.40     | \|   | 3.96     | **10.12** | 72.32     |
>
>
>
> 2. **[Our method is not identical to [1]]**
>
> Our method is **not** identical to [1] for several reasons:
>
> (i) we do not assume known noise-label transition probabilities;
>
> (ii) Our proposed method DMIS supports to handle **partial labels** and **supplementary-unlabeled data** in addition to noisy labels, all within the same likelihood-based framework.
>
> (iii) We augment the generative objective with an explicit classification term and derive a **posterior-guided sampler** and timestep-selection strategy built on our variational formulation.
>
> (Iiii) **TDSM as a special case of DMIS.** When imprecise supervision degenerates to standard label noise *and* the transition matrix is known, our weighted score objective can be instantiated so that it **recovers TDSM [1] as a special case**.
>
>
>
> [2] "Don't drop your samples! Coherence-aware training benefits Conditional diffusion." *Proceedings of the IEEE/CVF Conference on Computer Vision and Pattern Recognition*. 2024.

---

> ### Author Response · Authors · 2025-11-27
> **Response to Reviewer 9nKU (W3)**
>
> > **W3: Limited experimental scope: Recently, zero-shot tasks using text-conditional diffusion models have shown greater potential and received more attention than those based on class-conditional models. It would be more important to address such tasks using text-to-image (T2I) diffusion models.**
>
> We appreciate this valuable comment and agree that text-conditioned diffusion models (e.g. Stable Diffusion,  DALL E-style models) have enabled many impactful zero-shot and generative applications.
>
> 1. **[Why we focus on class-conditional models in this work]**
>
>    Our present work focuses on class-conditional generation because it allows a clean and verifiable theoretical treatment: we can explicitly model a latent clean label and its imprecise counterpart, and systematically study partial-label data, supplementary-unlabeled data, and noisy-label data under controlled conditions.
>
> 2. **[How our framework can in principle extend to text-to-image diffusion]**
>
>    Importantly, however, the proposed framework is **not intrinsically limited to discrete class labels**. A text-conditioned model can be brought into our framework by mapping a text prompt to a soft class-conditional representation via a text classifier that outputs a distribution over classes. For example,
>
>    > On CIFAR-10, a prompt such as “a deer that looks like a horse” could be mapped to a soft label vector like [0, 0, 0 0, 0.7, 0, 0, 0.3, 0, 0], where the “horse” class receives 0.3 confidence and the “deer” class receives 0.7.
>
>    Such a soft label vector is a continuous extension of our imprecise-label variable and can be naturally plugged into our pipeline as the label condition.
>
>    While our current experiments in this paper are class-conditional, to the best of our knowledge there is no prior work that systematically studies diffusion models under multiple forms of *imprecise supervision* (partial-label data, supplementary-unlabeled data, and noisy-label data) and derives a unified objective tailored to this setting. We therefore see our class-conditional study as a foundational step toward handling **imprecise text-conditioned generation**, which is increasingly important as large quantities of low-quality supervision (including AI-generated text and labels) become common.

---

### Official Review · Reviewer_aCun · 2025-10-31

**Soundness:** 1
**Presentation:** 2
**Contribution:** 2
**Rating:** 2
**Confidence:** 4

**Summary:**

This paper studies the robustness of diffusion model training under imprecise supervision. Unlike prior works that focus on specific forms of imprecise supervision, they propose a unified framework for robust diffusion learning in such settings. The approach decomposes the likelihood maximization objective into generative and classification components. The generative component trains the diffusion model, while the classification component handles imprecise label classification. The framework is instantiated from three types of imprecise supervision: partial labels, supplementary-unlabeled data, and noisy labels, and evaluated across three benchmark datasets on image generation, weakly supervised classification, and dataset condensation tasks.

**Strengths:**

* Addressing how to train diffusion models in the presence of incomplete data is an important and meaningful problem.

* Unlike prior works that focus on specific types of imprecise supervision, the paper's proposal of a unified framework represents a valuable contribution.

**Weaknesses:**

* In Section 3.1, Eq. (6) suggests maximizing the joint distribution $p_{\theta} (X,Z)$. It is unclear why the objective is formulated this way rather than maximizing the marginal distribution $p_{\theta} (X)$, or the marginal/conditional distributions involving $Y$, such as $p_{\theta} (X,Y)$ or $p_{\theta}(X|Y)$. I want to know that the reason that the likelihood of imprecise supervision also be increased. I think that it might be more natural to consider maximizing the conditional likelihood over $Y$.

* In Eq. (7), the objective is decomposed into generative and classification components, where the generative objective appears to correspond to $\log p_{\theta} (X|Z)$, as indicated by the subtitle of Section 3.2. However, instead of maximizing this term directly (as in Eq. (8)), the paper proceeds to Eq. (10), which maximizes the conditional likelihood over $Y$. The reasoning behind this transition is unclear. As mentioned above, directly maximizing the conditional likelihood over $Y$ might be conceptually more coherent, so clarification of this logical flow is needed.

* In Section 3.2, Proposition 1 describes the optimal point of the learned score function under the objective in Eq. (10). However, the overall training objective includes an additional classification term, as shown in Eq. (7). Then, it is beneficial to discuss the optimal point of the final objective. A discussion on this one would strengthen the theoretical grounding.

* The methodology in Section 3.2 appears highly similar to that in Na et al. (2024), both in structure and explanation. The mathematical derivations seem almost identical, with only variable substitutions (e.g., replacing $\tilde{Y}$ with $Z$), and the sequence of Remark, Theorem, and Proposition mirrors that paper closely. Several parts of the text also appear to be paraphrased. Proper attribution and a clear explanation of what is newly contributed beyond that paper are necessary.

* In Eq. (13), it is unclear why $f_{\theta}$ is not normalized. Also, the explanation for normalized probability provided below the equation is insufficient. The paper only states that it lies on a simplex, but a concrete description of the normalization process is required.

* The notation for $\tilde{f}_{\phi}$ in Eqs. (13) and (14) appears to refer to different quantities, yet the same symbol is used, which could cause confusion. Distinguishing them more clearly would improve readability.

* The proposed method requires evaluating the diffusion model for each class during every objective computation, which may significantly increase computational complexity. Section 4 discusses efficiency improvements in the classification evaluation stage by reducing the timesteps. Since timesteps are still sampled during training, a discussion on how to mitigate the per-class computational cost during training is needed.

* The experiments are conducted on three benchmark image datasets, each with 10 classes. It would be helpful to analyze whether the proposed method scales to datasets with larger number of classes (or even infinite classes, i.e., continuous labels). This issue is closely related to the aforementioned computational complexity.

* Experimental comparisons with prior diffusion-based methods specifically designed for different types of imprecise supervision are missing. Including such comparisons, both methodological and empirical, would help clarify the novelty and significance of the proposed framework.

**Questions:**

Please discuss the points in the Weaknesses.

---

> ### Author Response · Authors · 2025-11-26
> **Response to Reviewer aCun (W1, W2)**
>
> We thank the reviewer for the careful reading and detailed comments. We are happy that you find both the problem and the idea of a unified framework valuable. Below we address each weaknesse (W) in turn.
>
>
>
> > **W1: Why is the objective in Eq. (6) formulated as maximizing the joint likelihood $p_\theta(X,Z)$ instead of e.g. $p_\theta(X)$ or a conditional likelihood over $Y$?**
>
> We apologize that the motivation here was not clearly explained.
>
> 1. **Why we start from the joint $p_\theta(X,Z)$.**
>    In our setting the **only observed supervision is the imprecise label $Z$**, while the clean label $Y$ is latent. Maximizing the marginal likelihood of observations corresponds to maximizing
>    $$
>    \log p_\theta(X,Z)=\log\sum_{y} p_\theta(X,y,Z),
>    $$
>    which is the standard latent-variable maximum likelihood principle. Starting from $p_\theta(X)$ would discard the available signal in $Z$, and directly maximizing $\log p_\theta(X\mid Y)$ is impossible because $Y$ is unobserved.
>
> 2. **Connection to conditional likelihoods.**
>    Under our factorization $p_\theta(X,Y,Z)=p_\theta(Y\mid X,Z)p_\theta(X\mid Z)p(Z)$, the joint objective is equivalent to
>
>    $$
>    \mathbb{E}\_{p(X,Z)}[\log p\_\theta(X\mid Z)] +
>    \mathbb{E}\_{p(X,Z)}\mathbb{E}\_{p\_\phi(Y\mid X,Z)}[\log p\_\theta(Y\mid X,Z)].
>    $$
>
>    Eq. (7) is precisely a variational lower bound of this form, where $p_\phi(Y\mid X,Z)$ approximates the intractable posterior and yields the decomposition into **generative** and **classification** terms.
>
> 3. **Why not directly maximize $\log p_\theta(X\mid Y)$.**
>    Our later derivation (Eq. (8)–(10)) shows that the generative term in Eq. (7) can be rewritten as an expectation of $\log p_\theta(X\mid Y)$ under the posterior over $Y$ given $(X,Z)$. In other words, **we do effectively maximize a conditional likelihood over $Y$**, but it is necessarily done through the posterior weighting induced by the imprecise supervision.
>
> ------
>
> > **W2: Logical flow from Eq. (7) → Eq. (8) → Eq. (10): why does the generative term, apparently corresponding to $\log p_\theta(X\mid Z)$, lead to an objective based on conditional likelihood over $Y$?**
>
>  Thank you for pointing out that this transition was not clearly explained; this is indeed central.
>
> - **Eq. (7)** decomposes the marginal log-likelihood into a generative term and a classification term using a variational posterior $p_\phi(Y\mid X,Z)$. The generative part corresponds to maximizing $\log p_\theta(X\mid Z)$.
>
> - **Eq. (8)** instantiates this generative term for diffusion models as a **denoising score matching objective w.r.t. the imprecise conditional score** $\nabla_{x_t} \log q_t(x_t\mid z)$. As stated in Remark 1, directly optimizing Eq. (8) causes the learned score to match the imprecise-label conditional, which leads to biased generation because $q_t(x_t\mid z)$ mixes different clean labels.
>
> - **Theorem 1 (Eq. (9))** shows that the imprecise-label conditional score can be expressed **exactly as a convex combination of the clean-label conditional scores**:
>
>   $
>   \nabla_{x_t}\log q_t(x_t\mid z)
>   = \sum_{y} p(y\mid x_t,z)\,\nabla_{x_t}\log q_t(x_t\mid y).
>   $
>
>   That is, correcting the bias amounts to re-weighting clean-label scores by the posterior $p(y\mid x_t,z)$.
>
> - **Eq. (10)** then defines the *modified* generative objective: instead of matching the biased imprecise score (Eq. (8)), we **directly train the network to approximate the weighted aggregation of clean-label scores** from Theorem 1. Proposition 1 shows that, under this objective, the optimal learned score recovers each clean-label conditional score.
>
> Overall,  Eq. (10) is not a unrelated conditional likelihood, but rather an *unbiased reparameterization* of the generative term in Eq. (7) guided by Theorem 1.

---

> ### Author Response · Authors · 2025-11-26
> **Response to Reviewer aCun (W3, W4, W5)**
>
> > **W3. Proposition 1 analyzes the optimum of the generative objective (Eq. (10)) only, whereas the final training objective also contains a classification term from Eq. (7). What is the optimal point of the full objective?**
>
> To clarify, under standard model-correctness assumptions and with sufficient model capacity, the full objective admits an optimum at which both the generative and classification parts are simultaneously optimal. Let $\theta^\star$ be a global maximizer of $\log p_\theta(X,Z)$. If the model family can represent the true joint distribution $q(X,Y,Z)$, then at this optimum we have $p_{\theta^\star}(X,Y,Z) = q(X,Y,Z)$, which immediately implies
> $$
> p_{\theta^\star}(x\mid y) = q(x\mid y), \qquad p_{\theta^\star}(y\mid x,z) = q(y\mid x,z).
> $$
> The first equality, together with the standard diffusion score-matching result and our derivation in Sec. 3.2, yields exactly the statement of Proposition 1:
> $$
> s_{\theta^\star}(x_t,y,t) = \nabla_{x_t} \log q_t(x_t\mid y),
> $$
> i.e., the learned score recovers the clean-label conditional score. The second equality states that the classifier converges to the Bayes posterior under imprecise supervision. Therefore, **the global optimum of the full objective consists of the clean-label score in Proposition 1 plus a Bayes-optimal classifier**, and the classification term does not change the target score—it enforces that the weights $p(y\mid x_t,z)$ used in Eq. (10) converge to the true posterior.
>
> ------
>
> > **W4. Need clearer attribution and explanation of novelty beyond Na et al. (2024).**
>
> We sincerely thank the reviewer for this critical comment. Our work is inspired by theirs, but we targets a more general problem and differs in several key aspects:
>
> 1. **Scope of supervision.**
>    Na et al. consider **only noisy labels with a known, instance-wise and time-dependent transition probability**. Their TDSM objective derives a linear relation between noisy-label and clean-label conditional scores under this strong assumption.
>    In contrast, our framework treats the clean label $Y$ as a latent variable and handles **three distinct forms of imprecise supervision**—partial-label supervision, supplementary-unlabeled supervision, and noisy-label supervision—within a single likelihood-based formulation. We **do not assume knowledge of any transition matrix or noise prior**; instead, we learn the effective weighting via the classifier and the imprecise posterior $p_\phi(Y\mid X,Z)$.
> 2. **Objective derivation and decomposition.**
>    Our derivation starts from the **joint likelihood $p_\theta(X,Z)$** and yields a **variational decomposition into generative and classification terms** (Eq. (7)), which unifies different supervision regimes. Theorem 1 and Eq. (10) then express the imprecise-label score as a mixture of clean-label scores *induced by the posterior $p(y\mid x_t,z)$*, without relying on an externally specified transition model. When the imprecise supervision reduces to simple label corruption with a known transition matrix, our weighted score objective recovers TDSM as a special case—this connection will be explicitly spelled out in the revised paper.
> 3. **Additional components and tasks.**
>    Beyond the robust generative objective, our paper introduces
>    (a) a **classification head and loss** tailored to each imprecise regime (partial, semi-supervised, noisy),
>    (b) a **posterior-guided diffusion classifier** with a new timesteps selection strategy based on Theorem 2 and 3, and
>    (c) the new task of **noisy dataset condensation**, where we use the diffusion model trained from imprecise supervision to synthesize compact yet label-robust datasets.
>
> > **W5: In Eq. (13), why is $f_\theta$ not normalized? The explanation that $\tilde f_\phi$ lies on a simplex is too brief.**
>
> - In our implementation, $f_\theta(\mathbf{x})$ denotes the output of the diffusion classifier defined in Eq. (11). Concretely, the classifier first produces the the conditional likelihood $\log p_\theta(\mathbf{x}|y)$ that is approximated by the conditional evidence lower bound in Eq. (12), and then set
>   $$
>   f_\theta(\mathbf{x})=\text{softmax}(\log p_\theta(\mathbf{x}|y)).
>   $$
>   Thus, $f_\theta(\mathbf{x})$ is a normalized probability vector lying in the simplex $\Delta^{c-1}$.
>
> - The symbol $\tilde f_\phi(\mathbf{x})$ in Eq. (13) denotes the pseudo-target distribution over labels constructed from EMA predictions $f_\phi(\mathbf{x})$, but restricted to the candidate set $\mathcal{S}$. For clarity, we rewrite $\tilde{f}_\phi(\mathbf{x})$ as $\tilde{f}(\mathbf{x};\phi)$, and make the normalization explicit:
>
> $$
> \tilde{f}\_k(x;\phi)=\dfrac{f\_k(x;\phi)}{\sum\_{k' \in \mathcal{S}} f\_{k'}(x;\phi)} \text{ for } k \in \mathcal{S},
> \tilde{f}\_k(x;\phi)=0 \text{ for }  k \notin \mathcal{S},
> $$
>
> so that $\sum_k \tilde{f}_k (\mathbf{x};\phi)=1$.

---

> ### Author Response · Authors · 2025-11-26
> **Response to Reviewer aCun (W6, W7, W8)**
>
> > **W6: The notation $\tilde f\_\phi$ in Eqs. (13) and (14) appears to refer to different quantities.**
>
> Thanks for the valuable comments. $\tilde f_\phi$ in Eqs. (13) and (14)  is a notational oversight.
>
> - In the **partial-label** case (Eq. (13)), $\tilde f_\phi(x)$ is the EMA-smoothed posterior restricted to candidate set $\mathcal{S}$.
> - In the **supplementary-unlabeled** case (Eq. (14)), $\tilde f_\phi(x)$ denotes a pseudo-target distribution that is one-hot for labeled data and the full EMA prediction over all classes for unlabeled data.
>
> We fix this by introducing **two distinct symbols** in the revised manuscript, e.g., $\tilde f_\phi^{\text{PL}}(x)$ for partial labels and $\tilde f_\phi^{\text{SU}}(x)$ for supplementary-unlabeled data, and clearly defining both in the text. This could remove the ambiguity and improve readability.
>
> ------
>
> > **W7: The proposed method requires evaluating the diffusion model for each class during every objective computation, which may significantly increase computational complexity. Section 4 discusses efficiency improvements in the classification evaluation stage by reducing the timesteps. Since timesteps are still sampled during training, a discussion on how to mitigate the per-class computational cost during training is needed.**
>
> Thanks for the valuable suggestion. We agree that, beyond timestep-level efficiency, it is important to also discuss the computational cost along the class level.
>
> In our current formulation, for example under partial-label supervision, the final objective indeed involves network evaluations for all classes, combining Eq. (10) and Eq. (14). Concretely, the generative and classification terms can be written as
>
> $$
> \mathcal{L}(\theta, \mathbf{x})=\mathbb{E}\_t[ w\_t ||  \sum\nolimits\_{y=1}^c p(y \mid \mathbf{x}\_t, z)\mathbf{s}\_\theta(\mathbf{x}\_t,y,t)   - \nabla\_{\mathbf{x}\_t}\log q\_{t\mid0}(\mathbf{x}\_t \mid \mathbf{x}\_0, z)
>    \||\_2^2 ] -\sum\_{y=1}^c \tilde{f}\_\phi^\text{PU}(\mathbf{x})\log f\_\theta(\mathbf{x})\_y,
> $$
>
> A naïve implementation would backpropagate gradients through all $c$ class scores for every training example.
>
> To mitigate this per-class cost, in practice **we only backpropagate through classes whose probability mass is non-negligible**. Specifically, we apply a threshold $\tau_\text{threshold}$ to both the diffusion posterior $p\_\phi(y\mid\mathbf{x}\_t,z)$ and the pseudo-label distribution $\tilde{f}\_\phi^\text{PU}(\mathbf{x})$: entries below $\tau_\text{threshold}$ are zero-masked and do not contribute to the gradient. Equivalently, this can be implemented as keeping only the top-$k$ classes per example and treating the remaining ones as fixed zeros. In this way, the effective complexity scales with the number of *active* classes per example (typically small), rather than with the total number of classes $c$.
>
> Importantly, this masking / top-$k$ strategy provides a straightforward path to scaling the method to datasets with larger classes.  Due to limited computational resources we have not yet included new large-class experiments in this submission, but we will explicitly describe this implementation detail in the revision and discuss how it enables extending our framework to larger-scale class-conditional settings.
>
> ------
>
> > **W8: Scalability to larger (or even infinite/continuous) label spaces; relation to the above complexity concern.**
>
> Thanks for raising this important point. As discussed in our response to W7, the per-class computational cost can be controlled by masking out low-probability classes. Concretely, during training we only backpropagate through classes whose posterior probability or pseudo-label mass exceeds a threshold $\tau_{\text{threshold}}$ (equivalently, we keep only the top-$k$ classes per example). This makes the effective cost depend on the number of *active* classes rather than on the total number of classes $c$, and thus provides a practical path to scaling our framework to datasets with many classes (e.g., hundreds or thousands).
>
> Regarding **continuous or “infinite” label spaces**, our current theory is developed for a discrete latent label $Y$. Conceptually, the sums over labels in our objective (e.g., Eq. (10)) can be generalized to integrals over a continuous semantic variable, and approximated via sampling or low-dimensional embeddings. In particular, many text-conditioned models can be viewed as operating on an effectively continuous label space: a textual prompt is mapped to an embedding or to a soft distribution over base classes. For example, on CIFAR-10, a prompt like “a deer that looks like a horse” could be mapped by a text classifier to a soft label vector $[0,0,0,0,0.7,0,0,0.3,0,0]$, assigning 0.7 confidence to “deer” and 0.3 to “horse”. Such soft label vectors live in a continuous simplex and fit naturally into our imprecise-label formulation as inputs to the same weighted score-matching objective.

---

> ### Author Response · Authors · 2025-11-26
> **Response to Reviewer aCun (W9)**
>
> > **W9: Experimental comparisons with prior diffusion-based methods specifically designed for different types of imprecise supervision are missing. Including such comparisons, both methodological and empirical, would help clarify the novelty and significance of the proposed framework.**
>
> Thank you for emphasizing this. In the following tables (also see new results in Appendix F.2 in the revised manuscript)，we compared other **noise-robust diffusion methods** [1,2] that are designed to handle noisy-label data. Importantly, both two compared methods assume access to additional prior information, which can give them an advantage in this setting. Despite this, our method still achieves the best overall performance under the same backbone and training budget. This suggests that our approach is competitive while relying on strictly weaker assumptions about the available supervision.
>
>
>
> |             |          | Sym-40%  |           | \|   |          | Asym-40%  |           |
> | :-----------: | :--------: | :--------: | :---------: | :----: | :--------: | :---------: | :---------: |
> | Metric      | FID      | IS       | Accuracy  | \|   | FID      | IS        | Accuracy  |
> | DMIS (Ours) | **3.47** | **9.80** | **88.63** | \|   | **3.10** | 9.73      | **88.83** |
> | CAD [1]     | 4.10     | 9.68     | 81.75     | \|   | 3.87     | 9.16      | 82.33     |
> | TDSM [2]    | 3.85     | 9.40     | 66.40     | \|   | 3.96     | **10.12** | 72.32     |
>
>
>
> [1] "Don't drop your samples! Coherence-aware training benefits Conditional diffusion." *Proceedings of the IEEE/CVF Conference on Computer Vision and Pattern Recognition*. 2024.
>
> [2] "Label-Noise Robust Diffusion Models." *The Twelfth International Conference on Learning Representations*. 2024.

---

> > ### Comment · Reviewer_aCun · 2025-11-28
> >
> > Thank you for the detailed response. However, I still have several concerns, outlined below.
> >
> > **W1.**
> >
> > 2. It appears that the factorization used in "our factorization" differs from the factorization used in the "equivalent joint objective". Could you clarify this? Specifically, in "our factorization", is it valid to write $p(X|Y)$ directly without assuming $p(X|Y,Z)=p(X|Y)$? Furthermore, despite this, the "equivalent joint objective" seems to rely on a factorization that is not consistent with the formulation used in "our factorization".
> >
> > 3. Could you explain more precisely where Eq. (7) is being "rewritten" in Eqs. (8)-(10)? My understanding is that Eqs. (8)-(10) constitute a separate derivation. Eqs. (8)-(10) discuss the conditional likelihood over $Y$, whereas Eqs. (6)-(7) discuss $\log p(X,Z)$, and I do not see a clear logical connection between them in the manuscript. I believe this is the essence of my original concern. Although the authors addressed this in W2, as I discuss below, the logical consistency still seems problematic.
> >
> > **W2.**
> > The authors mention that Eq. (10) is a modified generative objecctive. If so, does this not imply that the objective has changed from maximizing $\log p(X,Z)$ (as in Eq. (7)) to something else? In that case, isn't the model no longer maximizing the original target $\log p(X,Z)$?
> >
> > **W3.**
> >
> > * First, $\theta^{*}$ would be the global maximizer of $\mathbb{E}_{q(X,Z)}[ \log p(X,Z)]$, not of $\log p(X,Z)$.
> >
> > * Why must the joint distribution that includes $Y$ also match? This implies that the authors need to justify why consistency in the joint distribution over $(X,Y,Z)$ is required, rather than only over $(X,Z)$, which is the stated objective.
> >
> > **W4.**
> > I understand the authors' point that the methods differ. However, the structure, derivation, and even the associated proofs in Section 3.2 of the original manuscript (now Section 4.2 in the revised version) are nearly consistent to those in the referenced paper. Since this subsection contains no citation or discussion of that work, I raise this issue.
> >
> > **W7.**
> > Regarding the top-k approach you mentioned: is this method actually applied in all current experiments, or are the authors saying it can be applied? This is not mentioned in either the original or the revised manuscript.
> >
> > **W8.**
> >
> > * While top-k certainly helps from a computational perspective, it introduces approximation errors. Therefore, I believe the manuscript should include experiments demonstrating the performance trade-off.
> >
> > * While your comments may hold for CIFAR-10 text prompts, I am more curious about how this would apply to the standard text-to-image diffusion models such as Stable Diffusion. How would the proposed method extend to those architectures? Note that this point is about the potential extensibility of the proposed method; I am simply asking for the authors' thoughts, and it is not a major concern.
> >
> > **W9.**
> > The authors state that the proposed method does not assume knowledge of any transition matrix or noise prior. If this is the case, the approach could perform well under more complex settings such as instance-dependent label noise. It would be beneficial to include the experimental results evaluating the method under such noise conditions.

---

> > > ### Author Response · Authors · 2025-12-02
> > > **Further Response to Reviewer aCun (W1(1))**
> > >
> > > We thank the reviewer for the valuable feedback and for the careful follow-up. We address each of your additional concerns point by point.
> > >
> > > > **W1 (1): It appears that the factorization used in "our factorization" differs from the factorization used in the "equivalent joint objective". Could you clarify this? Specifically, in "our factorization", is it valid to write $p(X|Y)$ directly without assuming $p(X|Y,Z)=p(X|Y)$? Furthermore, despite this, the "equivalent joint objective" seems to rely on a factorization that is not consistent with the formulation used in "our factorization".**
> > >
> > > We apologize for the confusion caused by our earlier wording. The previous description of the factorization was imprecise, and we have corrected it in the revised version.
> > >
> > > **1. [Correct factorization and its role]**
> > >
> > > The factorization actually used in our derivations (see Appendix B.3) is the generic chain rule
> > > $$
> > > p_\theta(X,Y,Z)
> > > = p(Z)\,p_\theta(X\mid Z)\,p_\theta(Y\mid X,Z),
> > > $$
> > > which **does not** require any conditional independence assumption such as  $p(X\mid Y,Z) = p(X\mid Y)$. Our earlier text was a mistake in exposition, not in the derivation itself, and we are grateful to the reviewer for pointing it out.
> > >
> > > With this factorization, our starting point is the marginal likelihood of the observations:
> > > $$
> > > \log p_\theta(X,Z) = \log \sum_{y} p_\theta(X,Y=y,Z).
> > > $$
> > > Following Appendix B.3, for each $(X,Z)$ we introduce an auxiliary distribution $Q(Y)$ over the latent variable $Y$​. Using the standard variational trick, Using the standard variational trick,
> > > $$
> > > \log p\_\theta(X,Z)
> > > = \log \sum\_y Q(Y) \frac{p\_\theta(X,Y,Z)}{Q(Y)}
> > > \ge
> > > \mathbb{E}\_{Q(Y)}[\log p\_\theta(X,Y,Z)] + \mathcal{H}(Q),
> > > $$
> > > where $\mathcal{H}(Q)$ is the entropy of $Q$ and does not depend on $\theta$. Thus, maximizing $\log p_\theta(X,Z)$ w.r.t. $\theta$ is equivalent to maximizing
> > > $$
> > > \mathbb{E}\_{Q(Y)}[\log p\_\theta(X,Y,Z)].
> > > $$
> > > Substituting our factorization
> > > $$
> > > p\_\theta(X,Y,Z)=p(Z)p\_\theta(X\mid Z) p_\theta(Y\mid X,Z)
> > > $$
> > > gives
> > > $$
> > > \mathbb{E}\_{Q(Y)}[\log p\_\theta(X \mid Z)] + \mathbb{E}\_{Q(Y)}[\log p\_\theta(Y \mid X,Z)] + \mathbb{E}\_{Q(Y)}[\log p(Z)].
> > > $$
> > > The last term $\mathbb{E}\_{Q}[\log p(Z)]$ is constant w.r.t. $\theta$ and can be dropped. Moreover, $\log p\_\theta(X\mid Z)$ does not depend on $Y$, so the expectation over $Q$ is trivial. Choosing the variational distribution as our classifier $Q(Y)=p\_\phi(Y\mid X,Z)$​ yields exactly the “equivalent joint objective”:
> > > $$
> > > \mathbb{E}\_{p(X,Z)}[\log p\_\theta(X\mid Z)] + \mathbb{E}\_{p(X,Z)}\mathbb{E}\_{p\_\phi(Y\mid X,Z)}[\log p\_\theta(Y\mid X,Z)],
> > > $$
> > > which is Eq. (7) in the paper. In this precise sense, Eq. (7) is a variational lower bound for maximizing the marginal likelihood $\log p_\theta(X,Z)$, and the factorization used there is fully consistent with the factorization employed in the derivation.

---

> > > ### Author Response · Authors · 2025-12-02
> > > **Further Response to Reviewer aCun (W1(2), W2)**
> > >
> > > > **W1 (2): Could you explain more precisely where Eq. (7) is being "rewritten" in Eqs. (8)-(10)? My understanding is that Eqs. (8)-(10) constitute a separate derivation. Eqs. (8)-(10) discuss the conditional likelihood over $Y$, whereas Eqs. (6)-(7) discuss $\log p(X,Z)$, and I do not see a clear logical connection between them in the manuscript. I believe this is the essence of my original concern. Although the authors addressed this in W2, as I discuss below, the logical consistency still seems problematic.**
> > >
> > > Thank you for raising this comment. We clarify the relationship below.
> > >
> > > 1. **[Only the *generative term* of Eq. (7) is “rewritten” in Eqs. (8)-(10)]**
> > >
> > > Eq. (7) decomposes the original target into a generative part and a classification part:
> > > $$
> > > \underbrace{\log p\_\theta(X\mid Z)}\_{\text{generative term}} + \underbrace{\mathbb{E}\_{p\_\phi(Y\mid X,Z)}[\log p\_\theta(Y\mid X,Z)]}\_{\text{classification term}}
> > > $$
> > > When we mean **the generative part $\log p_\theta(X\mid Z)$ of Eq. (7)** , not the entire Eq. (7). The classification term is treated separately in Sec. 4.3 (Eqs. (13),(14),(15)).
> > >
> > >
> > >
> > > 2. **[The logical connection from Section 4.1 - 4.3]**
> > >
> > > ​	As the logic shown in the subheadings of section 4.1 through 4.3 illustrates,
> > >
> > > - **First**, in Section 4.1 we introduce our unified maximum-likelihood objective in Eq. (6) and its variational lower bound in Eq. (7). This lower bound naturally decomposes into two parts to be optimized: a **generative term** $\arg\max\_\theta \log p\_\theta(X\mid Z)$ and
> > > a **classification term** $\arg\max\_\theta \mathbb{E}\_{p\_\phi(Y\mid X,Z)}[\log p\_\theta(Y\mid X,Z)]$.
> > > - **Second**, Sections 4.2 and 4.3 then derive the concrete loss formulations for these two parts. Specifically, in Section 4.2 we show how the generative term $\log p_\theta(X\mid Z)$ can be reformulated via a variational lower bound and class-conditional assumption. In Section 4.3, we derive the corresponding classification losses for different types of imprecise-label data, given in Eq. (13), Eq. (14), and Eq. (15).
> > > - **Third**, in our experiments we use the sum of these two components as the final training loss, i.e., the generative loss in Eq. (10) plus one of the classification losses (Eq. (13), Eq. (14), or Eq. (15), depending on the supervision regime).
> > >
> > >
> > > > **W2: The authors mention that Eq. (10) is a modified generative objecctive. If so, does this not imply that the objective has changed from maximizing $\log p(X,Z)$ (as in Eq. (7)) to something else? In that case, isn't the model no longer maximizing the original target $\log p(X,Z)$?**
> > >
> > > Thank you for the quesiton.
> > >
> > > 1. As clarified in our response to **W1(2)**, Eq. (10) is a *modified objective only for the generative term* of Eq. (7). It does **not** replace the whole Eq. (7). The classification term in Eq. (7) is instantiated separately by Eqs. (13)/(14)/(15).
> > > 2. Regarding whether we still maximize the original target $\log p(X,Z)$: our derivation preserves the same target step by step:
> > >    - **Eq. (6)** is an equivalent expression of maximizing the original marginal likelihood $\log p_\theta(X,Z)$.
> > >    - **Eq. (7)** is a *variational lower bound* of Eq. (6). Optimizing Eq. (7) w.r.t. $\theta$ coincides with the maximizer of $\log p_\theta(X,Z)$.
> > >    - Within Eq. (7), we separate into the generative term and the classification term. **Eq. (8)** introduces a dvariation lower bound for the generative term; **Eqs. (13)/(14)/(15)** instantiate the classification term for different imprecise-label regimes. They are constructed so that their *global optima in $\theta$* correspond to the same maximum-likelihood solution.
> > >    - **Eq. (10)** is a further rewriting of Eq. (8) under the class-conditional setting,
> > >
> > > In summary, Eq. (10) is a *surrogate objective with the same global optimum in $\theta$* as the generative part of Eq. (7). Together with the classification term, the overall training still targets the maximum-likelihood solution for $\log p_\theta(X,Z)$ under our model.

---

> > > ### Author Response · Authors · 2025-12-02
> > > **Further Response to Reviewer aCun (W3(1), W3(2), W4, W7, W8(1), W8(2))**
> > >
> > > > **W3 (1):  $\theta^*$ would be the global maximizer of $\mathbb{E}\_{q(X,Z)}[\log p(X,Z)]$ , not of $\log p(X,Z)$.**
> > >
> > > Thank you for pointing this out. You are correct that the rigorous target is the population log-likelihood
> > > $$
> > > \mathbb{E}\_{q(X,Z)}[\log p\_\theta(X,Z)],
> > > $$
> > > where $q(X,Z)$ denotes the data distribution over the input variables $(X,Z)$. In the text we used $\log p\_\theta(X,Z)$ as a shorthand for this expected objective. This correction is purely notational and does not affect any of the subsequent arguments.
> > >
> > > > **W3 (2): Why must the joint distribution that includes $Y$ also match? This implies that the authors need to justify why consistency in the joint distribution over $(X,Y,Z)$ is required, rather than only over $(X,Z)$, which is the stated objective.**
> > >
> > > Thank you for raising this point. We agree that our previous wording around the “full optimum” was not precise enough, and we clarify it here:
> > >
> > > Our stated objective is indeed the marginal over $(X,Z)$, via the ELBO in Eq. (7). For the theoretical discussion, we adopt a **standard well-specified assumption** and a parameterization where the score network and the classifier have disjoint heads: we assume that there exists $(\theta^\*,\phi^\*)$ such that the generative part and the classification part of Eq. (7) can each attain their respective optima. Since the full objective is the sum of these two parts, such a pair $(\theta^\*,\phi^\*)$ is also a global maximizer of the full objective. In other words, we do *not* require the entire joint $p\_\theta(X,Y,Z)$ to match $q(X,Y,Z)$ pointwise; we only assume that the model is expressive enough for the generative and classification components to be simultaneously optimal.
> > >
> > >
> > >
> > >
> > >
> > >
> > >
> > > > **W4:  I understand the authors' point that the methods differ. However, the structure, derivation, and even the associated proofs in Section 3.2 of the original manuscript (now Section 4.2 in the revised version) are nearly consistent to those in the referenced paper. Since this subsection contains no citation or discussion of that work, I raise this issue.**
> > >
> > > We thank the reviewer for raising this important point. We will clearly distinguish the relationship with the referenced paper in the revised version and quote this work appropriately.
> > >
> > >
> > >
> > > > **W7: Regarding the top-k approach you mentioned: is this method actually applied in all current experiments, or are the authors saying it can be applied? This is not mentioned in either the original or the revised manuscript.**
> > >
> > > It is proposed as an optional strategy that can be applied within our framework when scaling to larger datasets with many classes; we will make this explicit in the revised manuscript and include corresponding experiments in the next answer.
> > >
> > >
> > >
> > >
> > > > **W8 (1): While top-k certainly helps from a computational perspective, it introduces approximation errors. Therefore, I believe the manuscript should include experiments demonstrating the performance trade-off.**
> > >
> > > Thank you for the suggestion. In the revised manuscript, we introduce the top-$k$ strategy in Appendix F.3 and provide an experiment on Caltech-15 [1] with 40% symmetric noise to quantify the performance when $k=10$.
> > >
> > > |             |      | Generation Metric |     |    |   |  \|  | Classification Metric |
> > > | :-----------: | :----: | :-----------------:   | :-------: | :-------: | :-------: | :--------: | :---------------------: |
> > > | Metric      | FID  | IS                | Density | Coverage |\| | |Accuracy              |
> > > | DMIS (Ours) | 4.25 | 12.39             | 103.83  | 96.20 |\| |  | 78.92                 |
> > >
> > >
> > >
> > > [1] "From trojan horses to castle walls: Unveiling bilateral backdoor effects in diffusion models." NeurIPS 2023 Workshop on Backdoors in Deep Learning-The Good, the Bad, and the Ugly. 2023.
> > >
> > >
> > >
> > > > **W8 (2): While your comments may hold for CIFAR-10 text prompts, I am more curious about how this would apply to the standard text-to-image diffusion models such as Stable Diffusion. How would the proposed method extend to those architectures? Note that this point is about the potential extensibility of the proposed method; I am simply asking for the authors' thoughts, and it is not a major concern.**
> > >
> > > Thank you for this question. One natural way to extend our method to architectures such as Stable Diffusion is to apply the framework **in the latent space** of a latent diffusion model.
> > >
> > > Concretely, instead of taking the pixel-space image $X$ as input, we would use the latent code $\text{Enc}(X)$ produced by the VAE encoder. The rest of our framework can be applied in exactly the same way.

---

> > > ### Author Response · Authors · 2025-12-02
> > > **Further Response to Reviewer aCun (W9)**
> > >
> > > > **W9: The authors state that the proposed method does not assume knowledge of any transition matrix or noise prior. If this is the case, the approach could perform well under more complex settings such as instance-dependent label noise. It would be beneficial to include the experimental results evaluating the method under such noise conditions.**
> > >
> > > Thank you for this suggestion. We agree that instance-dependent noiseare important test cases for methods that do not rely on an explicit noise transition matrix.
> > >
> > > **1. [Instance-dependent label noise on CIFAR-10]**
> > >
> > > We have additionally evaluated DMIS under **instance-dependent label noise** on CIFAR-10 (following the instance-dependent noise protocol in the noisy-label literature). The results below show that DMIS maintains strong generative quality and classification accuracy under such complex noise.
> > >
> > > |             |      | Generation Metric |     |    |       \|   | Classification Metric |
> > > | :-----------: | :----: | :-----------------: | :-------: | :-------: | :--------: | :---------------------: |
> > > | Metric      | FID  | IS                | Density | Coverage | \| |Accuracy              |
> > > | DMIS (Ours) | 4.85 | 9.21              | 102.24  | 82.06    | \| | 81.32                 |
> > >
> > >
> > >
> > > **2. [Real-world noisy-label dataset and partial-label datasets]**
> > >
> > > We also report results on **real noisy-label dataset CIFAR-10N [2]**  and **real partial-label dataset PLCIFAR10 [3]**, where labels are collected from human annotators
> > >
> > > DMIS achieves competitive performance on both datasets, suggesting that the proposed framework is effective not only under synthetic corruption but also under realistic human annotation noise and ambiguity:
> > >
> > > |             |      | CIFAR10N [2] |          | \|   |      | PLCIFAR10 [3] |          |
> > > | ----------- | ---- | ------------ | -------- | ---- | ---- | ------------- | -------- |
> > > | Metric      | FID  | IS           | Accuracy | \|   | FID  | IS            | Accuracy |
> > > | DMIS (Ours) | 3.22 | 9.66         | 93.21    | \|   | 2.95 | 9.82          | 93.65    |
> > >
> > >
> > >
> > > [2] "Learning with noisy labels revisited: A study using real-world human annotations." *The tenth International Conference on Learning Representations*. 2022.
> > >
> > > [3] "Realistic evaluation of deep partial-label learning algorithms." *The thirteenth International Conference on Learning Representations*. 2025.

---

### Official Review · Reviewer_CtmX · 2025-10-31

**Soundness:** 3
**Presentation:** 3
**Contribution:** 2
**Rating:** 4
**Confidence:** 4

**Summary:**

This paper proposes to **disentangle the generative and discriminative aspects of diffusion models** (through implicit classifiers). The authors claim that doing this helps the network to **better learn under label noise conditions**. The authors also introduce the task of **noise-robust dataset condensation**.

**Strengths:**

1. **Problem Relevance:** As highlighted by the authors, **real data is necessarily noisy**. Being **agnostic to the noise level of the data is key** to making the most out of available datasets.

2. **No Prior Knowledge:** The fact that this method **doesn't require prior knowledge on the data's noise structure** is a significant advantage for real-world application.

**Weaknesses:**

1. **Missing Related Work:** The paper is **missing at least two key works** on training with noisy datasets:

   * \[1\] "Don't drop your samples! Coherence-aware training benefits Conditional diffusion" (Dufour et al., CVPR 2024)

   * \[2\] "Ambient Diffusion Omni: Training Good Models with Bad Data" (Daras et al., 2025)
     The **authors should compare to these methods**. While those methods use prior information (which might give them an advantage), a comparison is necessary, especially since both use reasonable and easy-to-obtain priors.

2. **Numerical Results Not Convincing:** The **quantitative results are not convincing**. Some reported numbers show **almost no improvement over the baseline** (e.g., in Table 1, a FID of 3.33 vs. 3.47 for noisy label supervision). Without **confidence intervals**, these small differences are not meaningful. The method only seems to demonstrate clear benefits in the **partial label supervision setup**.

3. **Only Tested on Synthetic Tasks and Small Datasets:** A major issue is that the paper **claims targeting large scale datasets but only tests on small datasets like CIFAR**. For a class-conditional model, one would **at least expect results on ImageNet 256px**. These small datasets are **prone to overfitting**, making them a poor evaluation setup for claims about robustness and scalability.

4. **Problem Formulation:** **Noisy labels are usually not a significant issue for standard class-conditional generation**. This is perhaps reflected in the authors' decision to **work only on synthetic noise structures**. I could see benefits if the authors where to find real noisy datasets, but it's not the case here


5. **Limited Scope (Class-Conditional Only):** The method appears **limited to class-conditional generation**. Currently, **text-conditioned generation is a more prevalent task where noisy labels are a major problem**. The paper **doesn't showcase any real-world use cases** for this, whereas both missing citations \[1\] and \[2\] demonstrate results for T2I, proving it is feasible. I think if the authors want to have the impact they claim in the abstract they should aim for their method to work in this kind of setup

**Questions:**

I advise the authors to try and make their method work where uncertain labels are a reality. A simple setup could be semantic segmentation like in [1] since segmentation masks are much more prone to segmentation errors.
Even finding a real dataset that actually suffers from class conditional noise issues would prove the usefulness of this method. (Maybe very finegrained datasets where images are really similar? It's however not the case of all the datasets in the paper that are "easy" to discriminate

---

> ### Author Response · Authors · 2025-11-26
> **Response to Reviewer CtmX (W1)**
>
> We sincerely thank you for the time and effort you put into reviewing our paper. Below, we respond to each of the weaknesses (W).
>
> > **W1. Missing related work on noise-robust diffusion models [1] [2] and lack of comparison, even though these methods use reasonable and easy-to-obtain priors.**
>
> Thanks for the valuable comment. We will add the related work [1,2] to the main paper and clarify how our setting and objectives differ from them.
>
> 1. Following the reviewer’s suggestion, we have conducted additional experiments comparing our method with the coherence-aware diffusion approach (CAD) approach [1] on both the image generation and the noisy-label learning task. The reason for not having a comparison method [2] is explained in point two.
>
>    **For image generation task**, we compare [1] and our DMIS model on CIFAR-10 with 40% symmetric and asymmetric label noise, reporting FID, IS under the same architecture and training budget. **For noisy-label learning task**, we compare [1] and our DMIS as data generators for downstream classification. We use each model to synthesize the same number of labeled samples and then train a Wide-ResNet-40-10 classifier on top of these synthetic datasets.
>
>    | Dataset     |          | Sym-40%  |           | \|   |          | Asym-40% |           |
>    | :-----------: | :--------: | :--------: | :---------: | :----: | :--------: | :--------: | :---------: |
>    | Metric      | FID      | IS       | Accuracy  | \|   | FID      | IS       | Accuracy  |
>    | CAD [1]     | 4.10     | 9.68     | 81.75     | \|   | 3.87     | 9.16     | 82.33     |
>    | DMIS (Ours) | **3.47** | **9.80** | **88.63** | \|   | **3.10** | **9.73** | **88.83** |
>
>    **Results:**  Empirically, our method is competitive with, and in all cases clearly outperforms [1] on these metrics, even though [1] has access to additional coherence information beyond the noisy labels, whereas DMIS operates solely from noisy label.
>
>    Moreover,  the classifier trained on DMIS-generated data consistently achieves higher accuracy than the classifier trained on data from [1]. We attribute this to our objective explicitly optimizing the classifier component within the diffusion model, whereas [1] primarily focuses on coherence-aware training without an explicit clean-label estimation.
>
> 2. **[The rationale for not including [2] as a baseline]**
>
>    We appreciate the reviewer for drawing our attention to [2]. However, we respectfully clarify that [2] is not a suitable baseline for our experimental comparison, for two main reasons:
>
>    - **Different problem focus.** Daras et al. [2] tackle the problem of training diffusion models with “bad” image data, where the inputs are corrupted by motion blur, sensor artifacts, poor lighting, low resolution, etc. The main goal is to obtain strong conditional or unconditional models despite these image-space degradations. In contrast, *our work does not address image-space corruption; instead, we focus on the case where the supervision signal itself is unreliable*.
>    - **Different assumptions on conditioning information.** In [2], the conditioning information (e.g., labels, captions) is assumed to be reliable. *In our setting, we instead assume clean images but imprecise conditioning information* (partial labels, semi-supervision, noisy labels), which is precisely the problem our method is designed to handle.
>
>    Given these differences in objective and assumptions, we view [2] as addressing a complementary robustness dimension rather than a direct baseline for our setting.
>
> 3. **[On the availability of prior information]**
>
>    Last but not least, although [1] leverages coherence priors obtained from CLIP scores [3] or off-the-shelf confidence estimators, there are many realistic settings where such **priors are not readily available or are expensive to obtain**. For example:
>
>    - **Domain-specific industrial datasets** (e.g., satellite imagery, industrial inspection). In these regimes, off-the-shelf vision–language models such as CLIP are typically poorly calibrated or out-of-distribution. Moreover, in many production pipelines one only has noisy categorical labels from non-expert annotators; computing reliable coherence scores for every sample may be infeasible due to computational cost or data-governance restrictions.
>    - **Medical and scientific imaging.** In these applications, one often only has access to coarse outcome labels (diagnosis, phenotype, experimental condition), while obtaining high-quality auxiliary scores or confidence estimators would require labor-intensive expert annotation.
>
>    In such regimes, assuming that “reasonable and easy-to-obtain” priors are always available may be overly optimistic. Our framework is explicitly designed for these **prior-free scenarios**, where only imprecise labels are available and no additional side information is assumed.

---

> ### Author Response · Authors · 2025-11-26
> **Response to Reviewer CtmX (W2, W3)**
>
> > **W2. Numerical without confidence intervals and gains look small in some generative metrics, except the partial label supervision setup.**
>
> We thank the reviewer for raising this concern. In the revised version, we additionally include the corresponding standard deviations and now report **mean ± standard deviation** for all metrics in Table 10 in Appendix F.1. This makes it clearer that the observed improvements are consistent across runs rather than due to random fluctuations.
>
> 1. **[Regarding the small gains]**
>
>    In the revised Table 1 *we have additionally included an ‘oracle’ column, obtained by training the same diffusion model with clean labels*. This allows us to directly visualize how close each method is to the ideal clean-label performance. Across datasets and supervision regimes, our method consistently moves the performance closer to the oracle than the vanilla diffusion model.
>
>    We also empirically confirm an phenomenon that the reviewer alludes to: the vanilla diffusion model is indeed more robust under noisy-label supervision than under partial-label supervision. Our method nonetheless yields additional improvements in all regimes.
>
> 2. **[The importance of multiple generative metrics beyond FID]**
>
>    While FID is a standard metric, it does not fully capture the quality of image generation. Following prior work on diffusion evaluation [1, 4, 5, 6], we therefore report a battery of metrics, including Density, Coverage, CW-FID, CW-Density, and CW-Coverage. These metrics are designed to assess whether generated samples are not only realistic but also aligned with the intended semantic class.
>
>    In the noisy-label setting, our method brings substantial improvements on these conditional metrics, even when the FID gap alone appears modest. For example, on CIFAR-10 with 40% symmetric noise, we observe improvements such as:
>
>    Density: ***101.39 → 109.75***,  CW-FID: ***29.84 → 13.85***, CW-Density: ***72.98 → 107.23***, CW-Coverage: ***73.39 → 80.11*** over the vanilla diffusion baseline.
>
>    These gains are large in relative terms and clearly indicate that the model trained with our objective produces samples that better match the target classes under imprecise supervision.
>
>
>
> [1] "Don't drop your samples! Coherence-aware training benefits Conditional diffusion." *Proceedings of the IEEE/CVF Conference on Computer Vision and Pattern Recognition*. 2024.
>
> [2] "Ambient Diffusion Omni: Training Good Models with Bad Data."  *Advances in neural information processing systems*. 2025.
>
> [3] "Learning transferable visual models from natural language supervision." *International conference on machine learning*, 2021.
>
> [4] "Label-Noise Robust Diffusion Models." *The Twelfth International Conference on Learning Representations*. 2024.
>
> [5] "Iso-Diffusion: Improving Diffusion Probabilistic Models Using the Isotropy of the Additive Gaussian Noise." *CoRR* (2024).
>
> [6] "Denoising Likelihood Score Matching for Conditional Score-based Data Generation." *The tenth International Conference on Learning Representations*. 2022.
>
>
> > **W3: Only tested on small datasets; concern about scalability and overfitting.**
>
> We thank the reviewer for this concern. Our primary goal in this work is to propose a unified training framework for diffusion models under imprecise supervision, which is **architecturally compatible with larger backbones and larger datasets**.
>
> 1. **[scalability]**
>
>    Due to limited computational resources, we were not able to run ImageNet-256px–scale experiments. Scaling our framework to such large-scale settings is left as important future work.
>
> 2. **[Overfitting]**
>
>    All reported metrics in our experiments are computed on a **held-out clean test set**, not on the noisy training set. As shown by the test accuracy curves in Figure 7 (Appendix), the test performance of our method remains stable as training proceeds, suggesting that the models are not simply memorizing noisy labels.
>
> 3. **[Model size]**
>
>    The base architecture we adopt is EDM with 55.73M parameters, trained from scratch on CIFAR-10. This configuration has been widely used and validated in prior work [1,2,4,5,6] on diffusion models, and is generally not considered excessively overparameterized for CIFAR-10.

---

> ### Author Response · Authors · 2025-11-26
> **Response to Reviewer CtmX (W4, W5)**
>
> > **W4: Noisy labels are usually not a significant issue for standard class-conditional generation.**
>
> We respectfully clarify that label noise is not negligible in conditional generation. Class-conditional generative models are already used in many real-world domains, such as image editing [7], medical imaging [8,9], and manufacturing [10]. In these essential applications,  it is crucial to acknowledge that the problem of noisy labels in datasets continues to pose a significant challenge [11,12,13].
>
> For example, the stable diffusion [14] is trained on large-scale datasets such as LAION dataset [15], where captions and tags are  inevitably contain noise introduced by human annotators or web crawlers during the data collection process. It has been observed that pre-training on such noisy data can negatively impact downstream performance if the noise is not handled properly [16]. In the GAN community, several works explicitly study robustness to label noise in conditional generation [17, 18], which further supports that noisy labels are a relevant issue in practical class-conditional generation scenarios.
>
>
>
> [7] "Invertible conditional gans for image editing." *arXiv preprint arXiv:1611.06355*. 2016.
>
> [8] "Conditional synthetic data generation for robust machine learning applications with limited pandemic data." *Proceedings of the AAAI Conference on Artificial Intelligence*. 2022.
>
> [9] "Brain imaging generation with latent diffusion models." *MICCAI workshop on deep generative models*. 2022.
>
> [10] "A conditional generative model for predicting material microstructures from processing methods." *arXiv preprint arXiv:1910.02133*. 2019.
>
> [11] "The multidimensional wisdom of crowds." *Advances in neural information processing systems*. 2010.
>
> [12] "Learning from massive noisy labeled data for image classification." *Proceedings of the IEEE conference on computer vision and pattern recognition*. 2015.
>
> [13] "Robust learning at noisy labeled medical images: Applied to skin lesion classification." *2019 IEEE 16th International symposium on biomedical imaging (ISBI 2019)*. 2019.
>
> [14] "High-resolution image synthesis with latent diffusion models." *Proceedings of the IEEE/CVF conference on computer vision and pattern recognition*. 2022.
>
> [15] "Laion-400m: Open dataset of clip-filtered 400 million image-text pairs." *arXiv preprint arXiv:2111.02114*. 2021.
>
> [16] "Understanding and Mitigating the Label Noise in Pre-training on Downstream Tasks." *The Twelfth International Conference on Learning Representations*. 2024.
>
> [17] "Robustness of Conditional GANs to Noisy Labels*"Robustness of conditional gans to noisy labels." *Advances in neural information processing systems*. 2018.
>
> [18] "Label-noise robust generative adversarial networks." *Proceedings of the IEEE/CVF conference on computer vision and pattern recognition*. 2019.
>
> ------
>
> > **W5. The method appears limited to class-conditional generation. Currently, text-conditioned generation is a more prevalent task where noisy labels are a major problem**.
>
> We appreciate this feedback and agree that text-conditioned diffusion models such as Stable Diffusion and DALL E-style models  have led to many successful applications.
>
> Our present work focuses on class-conditional generation because it allows a clean theoretical treatment: we can explicitly model a latent clean label and its imprecise counterpart, and systematically study partial-label data, supplementary-unlabeled data, and noisy-label data under controlled conditions. **Importantly, however, the proposed framework is not intrinsically limited to discrete class labels.**
>
> A text-conditioned model can be brought into our framework by mapping a prompt to a soft class-conditional representation via a text classifier that outputs a distribution over classes.  For example,
>
> > On CIFAR-10, a prompt such as “a deer that looks like a horse” could be mapped to a soft label vector like [0, 0, 0 0, 0.7, 0, 0, 0.3, 0, 0], where the “horse” class receives 0.3 confidence and the “deer” class receives 0.7.
>
> Such a soft label vector is a continuous extension of our imprecise-label variable and can be naturally plugged into our pipeline as the label condition.
>
> While our current experiments are class-conditional, to the best of our knowledge there is no prior work that systematically studies diffusion models under multiple forms of *imprecise supervision* (partial-label data, supplementary-unlabeled data, and noisy-label data) and derives a unified objective tailored to this setting. Our work on the class-conditional is a necessary step to solidify the theoretical foundation for handling imprecise-label data and imprecise-text conditioned generation, which we believe that those issues will be further discussed in the future where low-quality data is increasingly generated by AI.

---

### Official Review · Reviewer_MjZ5 · 2025-11-01

**Soundness:** 3
**Presentation:** 3
**Contribution:** 3
**Rating:** 4
**Confidence:** 3

**Summary:**

The paper addresses training diffusion models under noisy supervision by decomposing the likelihood objective into (1) a generative term modeling data confidence and (2) a classification term estimating clean labels. During sampling, the noisy condition is expressed as a linear combination of the clean label and its posterior, which also improves sampling efficiency.

**Strengths:**

- Clear separation between generative modeling and clean-label estimation.
- Theoretical framing is well motivated.
- Sampling process is made more efficient via posterior-guided conditioning.

**Weaknesses:**

- Missing comparisons with existing diffusion or image synthesis works; FID scores remain far from SOTA, making it difficult to assess the benefit.
- Related work is not in the main paper
- For Task 2 appears to require multiple forward passes slower inference.
- Experiments rely mainly on synthetic noisy labels rather than real-world imprecise supervision (e.g., text prompts).

**Questions:**

- The assumption from Bo Han et al. (2021) about overfitting under noise may not hold without explicit regularization is this verified empirically?
- “Noisy dataset condensation” is introduced but undefined; a clear explanation or reference is needed.
- The paper mentions ImageNette but seems to use ImageNet

---

> ### Author Response · Authors · 2025-11-26
> **Response to Reviewer MjZ5 (W1, W2)**
>
> We thank the reviewer for taking the time to carefully evaluate our work and for the constructive, detailed feedback. Below, we present our answers to each weakness (W) and question (Q) point by point.
>
>
>
> > **W1: Missing comparisons with existing diffusion works and FID is far from SOTA.**
>
> Our goal is not to compete with the latest large-scale image synthesis systems, but to study how to train diffusion models under imprecise supervision in a controlled setting. Therefore, our comparisons focus on models trained **under the same imprecise supervision setting**, rather than on SOTA models trained with clean labels and significantly larger architectures or datasets.
>
> That said, we think that the positioning with respect to existing diffusion models can be clarified and strengthened:
>
> 1. **[On SOTA FID vs our setting]**
>
>    SOTA FID scores are typically reported:
>
>    - at higher resolutions (e.g., 256×256 or 512×512),
>
>    * with access to clean labels or rich captions,
>
>    * using substantially larger backbones and training budgets (e.g., Stable Diffusion 1.4 with 860M parameters [1], Imagen with 11B parameters [2]).
>
>    In contrast, we deliberately use moderate-scale backbones (***55.73M parameters***), datasets with resolutions up to ***64×64***, and ***imprecise-label information***. Within this regime, our method consistently improves over all imprecise-supervision baselines (Vanilla) under the same architecture, resolution, and compute budget.
>
> 2. **[Additional comparison is needed]**
>
>    - We have added an “oracle” baseline, where the same backbone is trained with clean labels. The new results shown below (also see Table 1 in the revised manuscript) show how much performance can be achieved with fully clean supervision and make it clear what fraction of this performance is recovered by our method under imprecise supervision.
>
>      |      Metric      | FID  |  IS   | Density | Coverage | CW-FID | CW-Density | CW-Coverage |
>      | :--------------: | :--: | :---: | :-----: | :------: | ------ | :--------: | :---------: |
>      | (CIFAR10) Oracle | 2.05 | 10.61 | 112.59  |  83.27   | 9.83   |   111.70   |    83.91    |
>
>      |       Metric        |  FID  |  IS   | Density | Coverage | CW-FID | CW-Density | CW-Coverage |
>      | :-----------------: | :---: | :---: | :-----: | :------: | ------ | :--------: | :---------: |
>      | (ImageNette) Oracle | 11.52 | 13.81 | 117.23  |  80.12   | 40.20  |   120.35   |    78.48    |
>
>
>
>    - We have also shown comparisons against **noise-robust diffusion methods** [3, 4] that are designed to handle noisy-label data (also see new results in Appendix F.2 in the revised manuscript). Importantly, both two compared methods assume access to additional prior information, which can give them an advantage in this setting. Despite this, our method still achieves the best overall performance under the same backbone and training budget. This suggests that our approach is competitive while relying on strictly weaker assumptions about the available supervision.
>
>      |             |          | Sym-40%  |          | \|   |          | Asym-40%  |           |
>      | :-----------: | :--------: | :--------: | :--------: | :----: | :--------: | :---------: | :---------: |
>      | Metric      | FID      | IS       | Accuracy | \|   | FID      | IS        | Accuracy  |
>      | DMIS (Ours) | **3.47** | **9.80**     | **88.63**    | \|   | **3.10** | 9.73      | **88.83** |
>      | CAD [3]     | 4.10     | 9.68 | 81.75    | \|   | 3.87     | 9.16      | 82.33     |
>      | TDSM [4]    | 3.85     | 9.40     | 66.40    | \|   | 3.96     | **10.12** | 72.32     |
>
> [1]  "High-resolution image synthesis with latent diffusion models." *Proceedings of the IEEE/CVF conference on computer vision and pattern recognition*. 2022.
>
> [2] "Photorealistic text-to-image diffusion models with deep language understanding." *Advances in neural information processing systems*. 2022.
>
> [3] "Don't drop your samples! Coherence-aware training benefits Conditional diffusion." *Proceedings of the IEEE/CVF Conference on Computer Vision and Pattern Recognition*. 2024.
>
> [4] "Label-Noise Robust Diffusion Models." *The Twelfth International Conference on Learning Representations*. 2024.
>
> ---
>
>
>
> > **W2: Related work is not in the main paper.**
>
> We agree that placing the related work only in the appendix makes it harder to see how our method connects to existing literature. This was due to space constraints and can be improved.
>
> In the revised version, we move a concise but complete ``Related Work” section into the main paper. The main text will explicitly cover:
>
> * Robust diffusion models with imperfect conditions,
> * Imprecise label learning in discriminative context, including partial-label learning, semi-supervised learning, and noisy-label learning.
> * Dataset condensation and its main paradigms.

---

> ### Author Response · Authors · 2025-11-26
> **Response to Reviewer MjZ5 (W3, W4)**
>
> > **W3: For Task 2, inference appears to require multiple forward passes and thus slower inference.**
>
> We thank the reviewer for raising this concern. We agree that, in Task 2, estimating the conditional ELBO during inference naively involves $T$ forward passes, since the estimating objective integrates over different timesteps. However, the reviewer’s concern is exactly what our **Time Complexity Reduction** scheme (Section 4) is designed to address.
>
>
>
> Concretely, we compare our inference procedure against the current SOTA diffusion classifier method APNDC [5] as follows. For instance, in the setting where we classify samples at noise level $t = 0.5$, our method requires only ***2 sampling steps per-class*** to achieve *better* classification performance than APNDC with ***64 sampling steps per-class***. Compared to APNDC, ***our method achieves a speed-up of more than 320x in inference***.
>
> |                     |       Ours / APNDC        |    Ours / APNDC     |    Ours / APNDC     |    Ours / APNDC     |    Ours / APNDC     |       Ours / APNDC        |
> | :-----------------: | :-----------------------: | :-----------------: | :-----------------: | :-----------------: | :-----------------: | :-----------------------: |
> | noise level / steps |             2             |          4          |          8          |         16          |         32          |            64             |
> |       $x\_0$        |    74.6±0.8 / 69.9±0.5    | 87.1±1.0 / 85.4±1.1 | 90.7±0.5 / 90.6±0.4 | 93.5±0.3 / 92.0±0.6 | 95.2±0.5 / 93.3±0.6 |    95.6±0.3 / 93.7±0.2    |
> |     $x\_{0.5}$      | ***75.5±1.2*** / 42.8±1.7 | 82.6±0.7 / 61.3±0.6 | 86.5±0.3 / 68.7±1.1 | 86.9±0.3 / 70.7±0.8 | 87.1±0.4 / 71.7±0.7 | 87.9±0.4 / ***72.5±0.4*** |
> |     $x\_{1.0}$      |    60.3±0.3 / 18.9±1.0    | 67.7±1.5 / 29.6±0.9 | 71.4±0.4 / 35.6±0.4 | 72.8±1.1 / 38.6±0.6 | 74.3±0.7 / 39.8±0.5 |    74.6±0.4 / 41.0±0.6    |
>
> We also would like to emphasize that diffusion classifiers are still a relatively new line of work. Among prior methods, [5] is one of the few approaches that can directly classify noisy states $x_t$, whereas works such as [6] and [7] focus on classifying only the clean image $x_0$. Building on this line of work, the proposed time-complexity reduction mechanism further improves inference efficiency while maintaining strong classification accuracy at noisy states.
>
> [5] "Diffusion models are certifiably robust classifiers." *Advances in Neural Information Processing Systems*. 2024.
>
> [6] "Robust classification via a single diffusion model." *Proceedings of the 41st International Conference on Machine Learning*. 2024.
>
> [7] "Your diffusion model is secretly a zero-shot classifier." *Proceedings of the IEEE/CVF International Conference on Computer Vision*. 2023.
>
> ------
>
> > **W4: Experiments rely mainly on synthetic noisy labels rather than real-world imprecise supervision.**
>
> We agree that real-world imprecise supervision is important. We chose synthetic noisy labels primarily for a controlled setting to support the theory and to control the noise rate and type (symmetric, class-dependent, etc.).
>
> We believe that starting from a controlled synthetic-noise setting is a necessary first step to validate the theory and experiments, and adding a real-world noisy-supervision case study further strengthens the empirical evidence. Accordingly, we also report results on the real noisy-label dataset CIFAR-10N [8] and the real partial-label dataset PLCIFAR10 [9], whose labels are provided by human annotators.
>
> |             |      | CIFAR10N [8] |          | \|   |      | PLCIFAR10 [9] |          |
> | :-----------: | :----: | :------------: | :--------: | :----: | :----: | :-------------: | :--------: |
> | Metric      | FID  | IS           | Accuracy | \|   | FID  | IS            | Accuracy |
> | DMIS (Ours) | 3.22 | 9.66         | 93.21    | \|   | 2.95 | 9.82          | 93.65    |
>
> [8] "Learning with noisy labels revisited: A study using real-world human annotations." *The tenth International Conference on Learning Representations*. 2022.
>
> [9] "Realistic evaluation of deep partial-label learning algorithms." *The thirteenth International Conference on Learning Representations*. 2025.

---

> ### Author Response · Authors · 2025-11-26
> **Response to Reviewer MjZ5 (Q1, Q2, Q3)**
>
> > **Q1: The assumption from Bo Han et al. (2021) about overfitting under noise may not hold without explicit regularization. Is this verified empirically?**
>
> In our experiments, we clarify that this phenomenon **does** occur even when the backbone is a diffusion model rather than a purely discriminative network. Qualitatively, this can be seen from Figure 7 in the appendix: for the vanilla conditional diffusion baseline trained directly on noisy-label data and **without any noise-specific regularization**, the test accuracy first increases and then decreases as training proceeds. This “first rise then drop” pattern of test performance is precisely the kind of noisy-label overfitting observed in prior classification work.
>
>
>
>
>
> ------
>
> > **Q2: “Noisy dataset condensation” is introduced but undefined; a clearer explanation or reference is needed.**
>
> We appreciate this comment. Clearly defining the terminology indeed makes the paper more accessible to readers who are not already familiar with dataset condensation.
>
> In our work, dataset condensation (a.k.a. dataset distillation) [10, 11] is defined as follows.
>
> Given a learning algorithm $\Phi$, let $\theta^D$ and $\theta^{\bar{D}}$ be the optimal parameters obtained by directly training $\Phi$ on the original dataset $D$ and on a condensed dataset $\bar{D}$, respectively. The goal of dataset condensation is to learn a small condensed dataset $\bar{D}$, with budget $n = |\bar{D}|$, such that training on $\bar{D}$ closely matches the behavior of training on $D$:
>
> $\arg\min_{\bar{D},n}(\sup_{x \sim \mathcal{X},y \sim \mathcal{Y}}{|l(\Phi_{\theta^D}(x),y)-l(\Phi_{\theta^\bar{D}}(x),y)|})$,
>
> where $\ell$ is a twice-differentiable loss function and $y$ is the desired clean label associated with $x$.
>
> However, most existing methods assume that the observed training dataset $D$ is fully clean. In contrast, we **firstly** consider a more realistic situation where the available training data are noisily labeled. We refer to this setting as **noisy dataset condensation**. In this setting, we aim to construct a small clean condensed dataset $\bar{\hat{D}}$  from $\hat{D}$, such that training on $\bar{\hat{D}}$ faithfully reproduces the behavior of training on the (unobserved) clean dataset $D$:
>
> $\arg\min_{\bar{\hat{D}},n}(\sup_{x \sim \mathcal{X},y \sim \mathcal{Y}}{|l(\Phi_{\theta^D}(x),y)-l(\Phi_{\theta^\bar{\hat{D}}}(x),y)|})$,     s.t. $\bar{\hat{D}}=\mathcal{G}(\hat{D})$,
>
> where $\mathcal{G}$ denotes a condensation procedure that only has access to the noisy dataset $\hat{D}$.
>
> We will add this definition to the revised manuscript to make the notion of “noisy dataset condensation” more precise.
>
> [10] Data Distillation: A Survey. Transactions on Machine Learning Research. 2023.
>
> [11] Dataset distillation: A comprehensive review. IEEE transactions on pattern analysis and machine intelligence. 2023.
>
> ------
>
> > **Q3: The paper mentions ImageNette but seems to use ImageNet.**
>
>
> Thank you for catching this inconsistency. We indeed  use ImageNette, the 10-class subset of ImageNet, in the experiments. We will carefully revise the manuscript to clearly and consistently state this choice throughout the text and tables.
>
> [12] ImageNette: Available at https://github.com/fastai/imagenette

---

### Meta-Review · Area_Chair_voBc · 2026-01-08

**Summary:**

I believe that the paper would benefit from further development and, in its current form, may not yet be ready for publication. My primary concern relates to the experimental validation, particularly the absence of large-scale, real-world experiments. As also noted by **Reviewer MjZ5** and **Reviewer CtmX**, the proposed method has been evaluated mainly on synthetic and small-scale datasets. Without validation on more challenging and realistic benchmarks, it remains difficult to assess the method’s robustness and practical applicability.

In addition, I align with the shared perspective of **Reviewers MjZ5, CtmX, and 9nKU** that the evaluation could be broadened beyond simple class-conditional generation. Demonstrating generalization to more open-ended settings, such as text-to-image generation, would offer stronger evidence of the method’s effectiveness in contemporary generative modeling scenarios.

With respect to empirical performance, the current results do not yet demonstrate a clear advantage over existing approaches, particularly when compared with state-of-the-art baselines, as also observed by **Reviewer MjZ5**. There are also concerns regarding computational efficiency, and I share the reservations expressed by **Reviewers MjZ5 and aCun** about the additional computational cost introduced by the method.

Beyond computational cost, there is a more fundamental concern regarding scalability, as highlighted by **Reviewer aCun**. Restricting the evaluation to a 10-class dataset provides limited insight into whether the method can effectively scale to datasets with substantially higher class complexity. Finally, after careful consideration, I agree with **Reviewer 9nKU** that the technical and theoretical novelty of the proposed framework is not yet sufficiently clear and would benefit from further clarification and justification.

**Reviewer Concerns:**

I believe that only a limited portion of the concerns related to experimental results have been addressed, while most of the reviewers’ comments remain unresolved.

**Reviewer Scores:**

My impression is that most reviewers may not significantly adjust their scores (such as from reject to accept), provided that the evaluation proceeds fairly and without influence from unrelated considerations.

---

### Decision · Program_Chairs · 2026-01-26

Reject